# High-dimensional Analysis of Knowledge Distillation: Weak-to-Strong Generalization and Scaling Laws

**M. Emrullah Ildiz**[*]      **Halil Alperen Gozeten**[*]      **Ege Onur Taga**
University of Michigan, Ann Arbor
`{eildiz, alperen, egetaga}@umich.edu`

**Marco Mondelli**[†]
Institute of Science and Technology Austria
`marco.mondelli@ist.ac.at`

**Samet Oymak**[†]
University of Michigan, Ann Arbor
`oymak@umich.edu`

## Abstract

A growing number of machine learning scenarios rely on knowledge distillation where one uses the output of a surrogate model as labels to supervise the training of a target model. In this work, we provide a sharp characterization of this process for ridgeless, high-dimensional regression, under two settings: *(i)* model shift, where the surrogate model is arbitrary, and *(ii)* distribution shift, where the surrogate model is the solution of empirical risk minimization with out-of-distribution data. In both cases, we characterize the precise risk of the target model through non-asymptotic bounds in terms of sample size and data distribution under mild conditions. As a consequence, we identify the form of the optimal surrogate model, which reveals the benefits and limitations of discarding weak features in a data-dependent fashion. In the context of weak-to-strong (W2S) generalization, this has the interpretation that *(i)* W2S training, with the surrogate as the weak model, can provably outperform training with strong labels under the same data budget, but *(ii)* it is unable to improve the data scaling law. We validate our results on numerical experiments both on ridgeless regression and on neural network architectures.

## 1 Introduction

The increasing number and diversity of machine learning models has motivated the development of techniques that leverage the output of one model to train a different one – a process known as knowledge distillation (Hinton et al., 2015). Variations of this approach include generating synthetic data from powerful language models (Wang et al., 2023; Gunasekar et al., 2023; Abdin et al., 2024), weak-to-strong generalization to obtain stronger models under weak supervision (Burns et al., 2023), and filtering/curating ML datasets via a smaller model to train a larger model (Fang et al., 2023; Lin et al., 2024b). The diversity of these applications motivates a deeper understanding of the statistical properties and limits of the distillation process.

In this work, we focus on the scenario where a target/student model is trained on the labels of a surrogate/teacher model. Let $\mathcal{D}_t$ denote the target distribution, $(\boldsymbol{x}_i, y_i)_{i=1}^n$ be sampled i.i.d. from this distribution, and $p$ denote the dimension of the features $\boldsymbol{x}_i$. Given a surrogate model $s$, we create the synthetic labels $y_i^s = s(\boldsymbol{x}_i)$ and obtain the target model by minimizing the empirical risk, i.e.,

$$\hat{f} = \arg \min_{f \in \mathcal{F}} \frac{1}{n} \sum_{i=1}^n \ell(y_i^s, f(\boldsymbol{x}_i)), \tag{1}$$

where $\mathcal{F}$ denotes the hypothesis class and $\ell$ the loss function. This procedure motivates a few fundamental questions regarding (1): *(i)* What is the excess risk of $\hat{f}$ compared to that of the minimizer of the population

---

[*]Equal Contribution
[†]Equal Advising

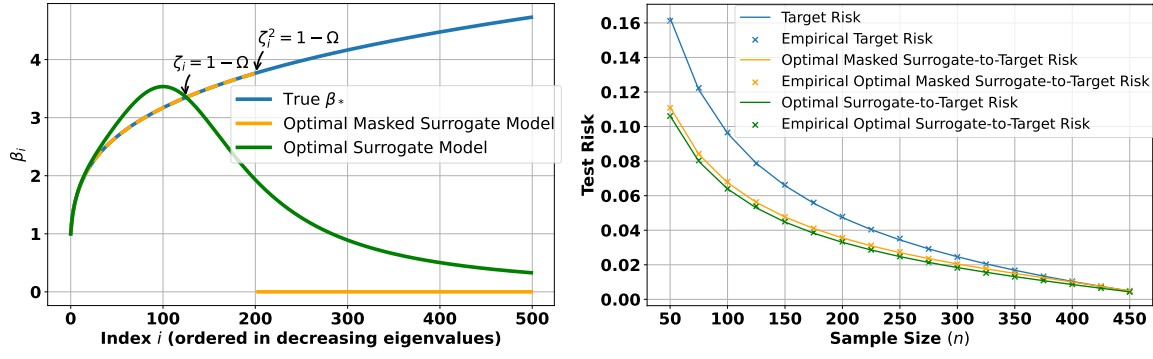

(a) Ground-truth and surrogate model weights      (b) Test risks as a function of sample size

Figure 1: Structure and performance of optimal surrogate models. **(a):** We compare the weights of the optimal surrogate model (green) with the ground-truth (blue). This reveals a transition from amplification to shrinkage as we move from principal to tail eigenvalues. The yellow curve displays the optimal 0-1 masking of the ground-truth where we either keep or discard a feature. **(b):** Associated test risks as a function of sample size. The theoretical bounds (full lines) match the experiments (markers). **Setting:** The feature size is $p = 500$; the sample size is $n = 200$ in (a) and variable in (b); the feature covariance follows the power-law structure $\lambda_i = i^{-2}$, $\lambda_i \beta_i^2 = i^{-1.5}$; $\zeta_i$ is the covariance statistics (see Corollary 1) governing the optimal surrogate's structure.

risk $f_\star = \arg\min_{f \in \mathcal{F}} \mathbb{E}_{(\boldsymbol{x}, y) \sim \mathcal{D}_t}[\ell(y, f(\boldsymbol{x}))]$? *(ii)* What is the optimal surrogate model $s$ that minimizes such excess risk? *(iii)* Can the optimal $s$ strictly outperform using true labels $y_i$ or setting $s = f_\star$ in (1)? Furthermore, in practice, $s$ itself is the outcome of an empirical risk minimization (ERM) procedure. Specifically, let $\mathcal{D}_s$ be the surrogate distribution, $(\tilde{\boldsymbol{x}}_i, \tilde{y}_i)_{i=1}^m$ be sampled i.i.d. from $\mathcal{D}_s$ (having feature dimension $p$), and assume

$$s = \arg\min_{f \in \mathcal{S}} \frac{1}{m} \sum_{i=1}^m \ell(\tilde{y}_i, f(\tilde{\boldsymbol{x}}_i)), \tag{2}$$

where $\mathcal{S}$ denotes the surrogate hypothesis class. Thus, as a final (and more challenging question), one may ask: *(iv)* How do the sample sizes $n, m$ and the data distributions $\mathcal{D}_t, \mathcal{D}_s$ affect the performance of $\hat{f}$?

**Main contributions.** We address these questions in the context of high-dimensional ridgeless regression, where $\mathcal{F}, \mathcal{S}$ are the class of linear models and $\ell$ is the quadratic loss. We provide a sharp non-asymptotic characterization of the test risk of $\hat{f}$ in two core settings:

1. The surrogate $s$ is provided and we solve $\hat{f}$ via (1), which addresses the first three questions above;

2. $\hat{f}$ is obtained via two stages of ERM, i.e., (2) followed by (1), which addresses the last question.

We focus on the regime where the sample sizes $n, m$ and the feature dimension $p$ are all proportional and also allow for a distribution shift between $\mathcal{D}_t$ and $\mathcal{D}_s$, which correspond to the two ERM stages. Our theoretical guarantees in Section 3 precisely characterize the regression coefficients $\boldsymbol{\beta}_s$ of the optimal surrogate model in terms of the corresponding coefficients $\boldsymbol{\beta}_\star$ of the population risk minimizer $f_\star$ and the feature covariance. This is depicted in Figure 1a where $\boldsymbol{\beta}_\star$ and $\boldsymbol{\beta}_s$ are displayed by blue and green curves, respectively. This unveils a remarkable phenomenology in the process of knowledge distillation, that can be described as follows. Define the per-feature 'gain' of the optimal surrogate as $\texttt{gain} = \boldsymbol{\beta}_s / \boldsymbol{\beta}_\star$ where the division is entrywise, which corresponds to the ratio between green and blue curves. We show that the $\texttt{gain}$ vector is entirely controlled by the *covariance statistics*, denoted by $(\zeta_i)_{i=1}^p$, that summarize the role of feature covariance and finite sample size $n$ in the test risk. There is a well-defined transition point, $\zeta_i = 1 - \Omega$ in Fig. 1a, where the $\texttt{gain}$ passes from strict amplification ($\texttt{gain}_i > 1$) to strict shrinkage ($\texttt{gain}_i < 1$) as we move from principal eigendirections to tail. Our theory also clarifies when we are better off discarding the weak features. The yellow curve shows the optimal surrogate when $\boldsymbol{\beta}_s$ is restricted to be a 0-1 mask on the entries of $\boldsymbol{\beta}_\star$ so that the surrogate has the direct interpretation of feature pruning. We show that beyond the transition point $\zeta_i^2 = 1 - \Omega$, truncating the weak features that lie on the tail of the spectrum is strictly beneficial to distillation. This masked surrogate model can be viewed as a weak supervisor as it contains strictly fewer features compared to the target, revealing a success mechanism for weak-to-strong supervision.

Notably, as the sample size $n$ decreases, the optimal surrogate provably becomes sparser (transition points shift to the left) under a power-law decay covariance model. In Section 4, under this power-law model, we also quantify the performance gain that arises from the optimal surrogate and show that while the surrogate can strictly improve the test risk, it does not alter the exponent of the scaling law. This is depicted in Figure 1b where the surrogate risks are smaller but behave similarly to the ground-truth. Finally, in Section 5, we study the more intricate problem of two-stage ERM ((2) followed by (1)) and establish a non-asymptotic risk characterization that precisely captures the influences of the sample sizes $m, n$ and of the surrogate/target covariance matrices.

## 1.1 RELATED WORK

Our work relates to the topics of high-dimensional learning, distribution shift, scaling laws, and distillation.

**High-dimensional risk characterization.** There is a large body of literature dedicated to the study of linear regression in over- and under-parameterized regimes. Via random matrix theory tools, one can precisely study the test risk and various associated phenomena, such as benign overfitting (Bartlett et al., 2020) or double descent (Belkin et al., 2019). Specifically, asymptotic and non-asymptotic risk characterizations for the minimum $\ell_2$-norm interpolator (i.e., ridgeless regression with $p \geq n$) have been provided in a recent line of work (Hastie et al., 2020; Cheng & Montanari, 2024; Han & Xu, 2023; Wu & Xu, 2020; Richards et al., 2021; Loureiro et al., 2022).

While the standard ERM formulation is well studied, the analysis of the distillation problem necessitates a fine-grained characterization of the ERM process. A series of papers (Chang et al., 2021; Montanari et al., 2023; Han & Xu, 2023) utilize Gaussian process theory (Thrampoulidis et al., 2015) to characterize the distribution of ridgeless estimators. Our theory builds on these to precisely characterize the distillation performance by *(i)* accounting for model and covariate shift and *(ii)* tracking the distribution across the two-stage problem where (2) is followed by (1). Our setting strictly subsumes the problem of characterizing the test risk under distribution shift, which relates to the recent papers (Patil et al., 2024; Song et al., 2024; Yang et al., 2023; Mallinar et al., 2024). A distinguishing feature of our work is that we precisely characterize the optimal surrogate model that minimizes the downstream target risk, as highlighted in Figure 1.

**Distillation and weak-to-strong generalization.** Mobahi et al. (2020) provide a theoretical analysis of self-distillation, whereas Menon et al. (2020); Nagarajan et al. (2023); Harutyunyan et al. (2023) study knowledge distillation in a teacher-student setting. A related problem is self-training which relies on progressively generating pseudo-labels for unlabeled data (Frei et al., 2022; Oymak & Gulcu, 2021; Wei et al., 2022b). While these works consider low-dimensional settings or provide loose bounds, our study provides a sharp analysis of ridgeless over-parameterized regression. Closer to us, Jain et al. (2024) investigate the benefit of *surrogate data* by employing both real and surrogate data in a single step of ERM, but the analysis is limited to isotropic covariance. Kolossov et al. (2023) consider the problem of surrogate-based data selection. Finally, Charikar et al. (2024); Lang et al. (2024) aim to demystify weak-to-strong generalization by formalizing the intuition that W2S generalization occurs when the strong model avoids fitting the mistakes of the weak teacher. In contrast, our theory reveals that the strong student can benignly overfit the weak teacher, and in fact, a carefully crafted weak teacher provably outperforms strong labels, see again Figure 1.

**Scaling laws.** The dependence of the performance on the available statistical and computational resources is often empirically well-captured by a power-law (Hestness et al., 2017; Kaplan et al., 2020). This experimental evidence has led to a flurry of theoretical work aimed at characterizing the emergence of scaling laws, mostly focusing on linear regression (Spigler et al., 2020; Simon et al., 2023; Bahri et al., 2024; Paquette et al., 2024; Bordelon et al., 2024b; Lin et al., 2024a; Maloney et al., 2022). Bordelon et al. (2024a) analyze a random feature model trained with gradient descent via dynamical mean field theory. Jain et al. (2024) consider scaling laws with surrogate data, whereas Sorscher et al. (2022) study the benefits of data pruning.

## 2 PROBLEM SETUP

**Notation.** Let $[p]$ denote the set $\{1, \cdots, p\}$ for an integer $p \geq 1$. We use lower-case and upper-case bold letters (e.g., $x, X$) to represent vectors and matrices, respectively; $x_i$ denotes the $i$-th entry of the vector $x$, $X^\dagger$ the pseudo-inverse of the matrix $X$, and $\mathtt{tr}(X)$ the trace of $X$. For further notations, please see Table 1.

We consider a two-stage linear learning problem. In the first stage, pairs of labels and input features from the distribution $\mathcal{D}_s$ are used to produce an estimate of the ground-truth parameter, which is then used to generate labels along with input features from a different distribution $\mathcal{D}_t$. The second stage uses these generated labels to obtain the final estimate of the ground-truth parameter. The models trained in the first and second stages are referred to as surrogate and target models, respectively.

**Stage 1: Surrogate model.** We consider a data distribution $(\tilde{x}, \tilde{y}) \sim \mathcal{D}_s$ following the linear model $\tilde{y} = \tilde{x}^\top \beta_\star + \tilde{z}$, where $\beta_\star \in \mathbb{R}^p$, $\tilde{x} \sim \mathcal{N}(0, \Sigma_s)$ and $\tilde{z} \sim \mathcal{N}(0, \sigma_s^2)$ is independent of $\tilde{x}$. Let $\{(\tilde{x}_i, \tilde{y}_i)_{i=1}^m\}$ be the dataset for the surrogate model drawn i.i.d. from $\mathcal{D}_s$. We analyze both under- and over-parametrized settings: in the former, we estimate $\beta_\star$ by minimizing the quadratic loss; in the latter, we estimate $\beta_\star$ as the minimum norm interpolator. As a result, the estimator of the surrogate model can be written as follows:

$$\beta^s = \text{Est}(\tilde{X}, \tilde{y}) := \begin{cases} \arg\min_\beta \|\tilde{y} - \tilde{X}\beta\|_2^2, & \text{if } m \geq p, \\ \arg\min_\beta \{\|\beta\|_2 : \tilde{X}\beta = \tilde{y}\}, & \text{if } m < p, \end{cases} \tag{3}$$

where $\tilde{X} = [\tilde{x}_1^\top, \ldots, \tilde{x}_m^\top]^\top \in \mathbb{R}^{m \times p}$ and $\tilde{y} = [y_1, \ldots, y_m]^\top \in \mathbb{R}^m$.

**Stage 2: Target model.** Given $\beta^s \in \mathbb{R}^p$, we consider another data distribution $(x, y^s) \sim \mathcal{D}_t(\beta^s)$ following the linear model $y^s = x^\top \beta^s + z$, where $x \sim \mathcal{N}(0, \Sigma_t)$ and $z \sim \mathcal{N}(0, \sigma_t^2)$. Let $\{(x_i, y_i^s)_{i=1}^n\}$ be the dataset for the target model drawn i.i.d. from $\mathcal{D}_t(\beta^s)$. As for the surrogate model, the estimator for the target model is defined as

$$\beta^{s2t} = \text{Est}(X, y^s), \tag{4}$$

where $X = [x_1^\top, \ldots, x_n^\top]^\top \in \mathbb{R}^{n \times p}$ and $y^s = [y_1^s, \ldots, y_n^s]^\top \in \mathbb{R}^n$. Our analysis will generally apply to an arbitrary $\beta^s$ choice and will not require it to be the outcome of (3). Finally, we define the excess (population) risk for a given estimator $\hat{\beta} \in \mathbb{R}^p$ as

$$\mathcal{R}(\hat{\beta}) := \mathbb{E}_{(x,y) \sim \mathcal{D}_t(\beta_\star)}[(y - x^\top \hat{\beta})^2] - \sigma_t^2 = \|\Sigma_t^{1/2}(\hat{\beta} - \beta_\star)\|_2^2. \tag{5}$$

Throughout the paper, we compare the surrogate-to-target model with two different reference models.

**Reference 1: Standard target model.** We study the generalization performance of $\beta^{s2t}$ with respect to the standard target model, which has access to the ground-truth parameter through labeling. Specifically, consider the dataset $\{(x_i, y_i)_{i=1}^n\}$ drawn i.i.d. from $\mathcal{D}_t(\beta_\star)$; then, the estimation is

$$\beta^t := \text{Est}(X, y), \tag{6}$$

where $X = [x_1^\top, \ldots, x_n^\top]^\top \in \mathbb{R}^{n \times p}$ and $y = [y_1, \ldots, y_n]^\top \in \mathbb{R}^n$. We compare the excess risks of the surrogate-to-target model $\mathcal{R}(\beta^{s2t})$ with that of the standard target model $\mathcal{R}(\beta^t)$.

**Reference 2: Covariance shift model** (Mallinar et al., 2024; Patil et al., 2024). Given $\beta_\star \in \mathbb{R}^p$, let $\{(x_i, y_i)_{i=1}^n\}$ be a dataset drawn i.i.d. from $\mathcal{D}_s^{cs}$, where $x_i \sim \mathcal{N}(0, \Sigma_s)$, $z_i \sim \mathcal{N}(0, \sigma_t^2)$, and $y_i = x_i^\top \beta_\star + z_i$; then, using the same notation $X = [x_1^\top, \ldots, x_n^\top]^\top \in \mathbb{R}^{n \times p}$ and $y = [y_1, \ldots, y_n]^\top \in \mathbb{R}^n$, the estimation is $\hat{\beta}^{cs} := \text{Est}(X, y)$. The test risk of $\hat{\beta}^{cs}$ is calculated under covariance shift. Let $(x, y) \sim \mathcal{D}_t$ be a distribution such that $x \sim \mathcal{N}(0, \Sigma_t)$, $z \sim \mathcal{N}(0, \sigma_t^2)$, and $y = x^\top \beta_\star + z$. Then, the excess transfer risk is

$$\mathcal{R}(\hat{\beta}^{cs}) := \mathbb{E}_{(x,y) \sim \mathcal{D}_t}[(y - x^\top \hat{\beta})^2] - \sigma_t^2.$$

We discuss the equivalence between the surrogate-to-target model and the covariance shift model in Section 3.

## 3 ANALYSIS FOR MODEL SHIFT

We start by examining the behavior of the surrogate-to-target model when there is a model shift $\beta^s \neq \beta_\star$. First, we provide a non-asymptotic bound on the risk conditioned on $\beta^s$, and then we optimize this quantity with respect to $\beta^s$ (finding also a closed-form expression of the corresponding optimal value of $\beta^s$). In addition, we build a connection between our surrogate-to-target model and knowledge distillation (Hinton et al., 2015), as well as weak-to-strong generalization (Burns et al., 2023). Finally, we extend our analysis to ridge regression in Appendix A.1.

We note that when $\beta^s = \beta_\star$, the calculations in this section simplify to the non-asymptotic risk characterization of the minimum $\ell_2$-norm interpolator, recently studied by Hastie et al. (2020); Cheng & Montanari (2024); Han & Xu (2023). When $\beta^s \neq \beta_\star$, the problem is instead equivalent to covariance shift in transfer learning, as formalized by the following observation whose proof is deferred to Appendix A.

**Observation 1.** *The model shift in the surrogate-to-target model is equivalent to the covariance shift model (Mallinar et al., 2024). Formally, given any $\beta_\star \in \mathbb{R}^p$ and any jointly diagonalizable covariance matrices $\Sigma_s, \Sigma_t \in \mathbb{R}^{p \times p}$, there exists a unique $\beta^s \in \mathbb{R}^p$ such that the risk of the surrogate-to-target problem $\mathcal{R}(\beta^{s2t})$ with $(\beta_\star, \beta^s, \Sigma_t)$ is equivalent to the risk of the covariance shift model $\mathcal{R}^{cs}(\hat{\beta})$ with $(\beta_\star, \Sigma_s, \Sigma_t)$.*

The joint diagonalizability of $\Sigma_s$ and $\Sigma_t$ is also required by Mallinar et al. (2024) (see their Assumption 2.1), where upper and lower bounds for the bias and variance are provided by adapting results from (Bartlett et al., 2020; Tsigler & Bartlett, 2022). In contrast, our approach utilizes the non-asymptotic characterization of the $\ell_2$-norm interpolator by Han & Xu (2023). This allows us to directly characterize the non-asymptotic risk (instead of giving upper and lower bounds, as in (Mallinar et al., 2024)) and, thus, to obtain the optimal surrogate parameter $\beta^s$. We begin by defining the asymptotic risk of the surrogate-to-target model, given the surrogate parameter $\beta^s$, in the proportional regime where $p, n \to \infty$ and the ratio $\kappa_t = p/n > 1$ is kept fixed.

**Definition 1.** *Let $\kappa_t = p/n > 1$ and $\tau_t \in \mathbb{R}$ be the unique solution of the following equation*

$$\kappa_t^{-1} = \frac{1}{p} tr\left((\Sigma_t + \tau_t I)^{-1} \Sigma_t\right). \tag{7}$$

*Let $\theta_1 := (\Sigma_t + \tau_t I)^{-1} \Sigma_t$ and $\theta_2 := (\Sigma_t + \tau_t I)^{-1} \Sigma_t^{1/2} \frac{g_t}{\sqrt{p}}$ where $g_t \sim \mathcal{N}(0, I_p)$. Now, define the asymptotic characterization of the minimum $\ell_2$-norm interpolator and the function $\gamma_t : \mathbb{R}^p \to \mathbb{R}$ as the following fixed point equation based on the asymptotic risk*

$$X^t_{\kappa_t, \sigma_t^2}(\Sigma_t, \beta^s, g_t) := \theta_1 \beta^s + \gamma_t(\beta^s) \theta_2, \tag{8}$$

$$\gamma_t^2(\beta^s) := \kappa_t \left(\sigma_t^2 + \bar{\mathcal{R}}^{s2t}_{\kappa_t, \sigma_t}(\Sigma_t, \beta^s, \beta^s)\right), \tag{9}$$

*where the asymptotic risk is defined as*

$$\begin{aligned}
\bar{\mathcal{R}}^{s2t}_{\kappa_t, \sigma_t}(\Sigma_t, \beta_\star, \beta^s) &:= \mathbb{E}_{g_t}\left[\|\Sigma_t^{1/2}(X^t_{\kappa_t, \sigma_t^2}(\Sigma_t, \beta^s, g_t) - \beta_\star)\|_2^2\right] \\
&= \underbrace{\|\Sigma_t^{1/2}(\theta_1 \beta^s - \beta_\star)\|_2^2}_{(a)} + \underbrace{\gamma_t^2(\beta^s) \mathbb{E}_{g_t}[\theta_2^\top \Sigma_t \theta_2]}_{(b)} \\
&= (\beta^s - \beta_\star)^\top \theta_1^\top \Sigma_t \theta_1 (\beta^s - \beta_\star) + \gamma_t^2(\beta^s) \mathbb{E}_{g_t}[\theta_2^\top \Sigma_t \theta_2] \\
&\quad + \beta_\star^\top (I - \theta_1)^\top \Sigma_t (I - \theta_1) \beta_\star - 2\beta_\star^\top (I - \theta_1)^\top \Sigma_t \theta_1 (\beta^s - \beta_\star).
\end{aligned} \tag{10}$$

In the asymptotic risk, the term $(a)$ corresponds to a part of the bias risk caused by the model shift $(\beta^s)$, and the implicit regularization term where the eigenvalues of $\theta_1$ are less than 1. The term $(b)$ corresponds to the remaining part of the bias and variance risks. We now state our non-asymptotic characterization of the risk.

**Theorem 1.** *Suppose that, for some constant $M_t > 1$, we have $1/M_t \le \kappa_t, \sigma_t^2 \le M_t$ and $\|\Sigma_t\|_{op}, \left\|\Sigma_t^{-1}\right\|_{op} \le M_t$. Recall from (5) that $\mathcal{R}(\beta^{s2t})$ represents the risk of the surrogate-to-target model given $\beta^s$. Then, there exists a constant $C = C(M_t)$ such that, for any $\varepsilon \in (0, 1/2]$, the following holds with $R + 1 < M_t$:*

$$\sup_{\beta_\star, \beta^s \in B_p(R)} \mathbb{P}\left(\left|\mathcal{R}(\beta^{s2t}) - \bar{\mathcal{R}}^{s2t}_{\kappa_t, \sigma_t}(\Sigma_t, \beta_\star, \beta^s)\right| \ge \varepsilon\right) \le Cpe^{-p\varepsilon^4/C}. \tag{11}$$

The proof of Theorem 1 utilizes the non-asymptotic characterization of the minimum norm interpolator in Han & Xu (2023), which is based on the convex Gaussian min-max theorem (Gordon, 1988; Thrampoulidis et al., 2015). The proof is deferred to Appendix A.

The surrogate parameter $\beta^s$ that minimizes the individual term $(a)$ in (10) is $\theta_1^{-1} \beta_\star$. On the other hand, the surrogate parameter $\beta^s$ that minimizes the individual term $(b)$ in (10) is the zero vector, which follows from (16) in Appendix A. Now, we are going to jointly minimize the asymptotic risk in the next proposition. The optimal surrogate parameter is visualized as the green curve in Figure 1.

**Proposition 1.** *Let $\Omega = \frac{tr(\Sigma_t^2(\Sigma_t + \tau_t I)^{-2})}{n}$. The optimal surrogate $\beta^s$ minimizing the asymptotic risk in (10) is*

$$\beta^{s*} = \left((\Sigma_t + \tau_t I)^{-1} \Sigma_t + \frac{\Omega \tau_t^2}{1 - \Omega} \Sigma_t^{-1}(\Sigma_t + \tau_t I)^{-1}\right)^{-1} \beta_\star.$$

Note that, under the setting of Theorem 1, Proposition 1 can be extended to the non-asymptotic risk by applying (11). The corollary below then offers a direct interpretation of the optimal surrogate parameter $\boldsymbol{\beta}^{s*}$. The proofs of both Corollary 1 and Proposition 1 are in Appendix A.

**Corollary 1.** *Without loss of generality, suppose that $\boldsymbol{\Sigma}_t$ is diagonal.[1] Let $(\lambda_i)_{i=1}^p$ be the eigenvalues of $\boldsymbol{\Sigma}_t$ in non-increasing order and let $\zeta_i = \frac{\tau_t}{\lambda_i + \tau_t}$ for $i \in [p]$. Then, the following results hold:*

*1. $\beta_i^{s*} = (\beta_*)_i \left( (1 - \zeta_i) + \zeta_i \frac{\Omega}{1-\Omega} \frac{\zeta_i}{1-\zeta_i} \right)^{-1}$ for every $i \in [p]$.*

*2. $|\beta_i^{s*}| > |(\beta_*)_i|$ if and only if $1 - \zeta_i > \Omega = \frac{\sum_{j=1}^p (1-\zeta_j)^2}{\sum_{j=1}^p (1-\zeta_j)}$ for every $i \in [p]$.*

*3. $\boldsymbol{\beta}^{s*} = \boldsymbol{\beta}_\star$ if and only if the covariance matrix $\boldsymbol{\Sigma}_t = c\boldsymbol{I}$ for some $c \in \mathbb{R}$.*

The first part shows that the optimal surrogate parameter is fully characterized by $(\zeta_i)_{i=1}^p$, which only depends on the covariance spectrum (via $\lambda_i$) and the sample size $n$ (via $\tau_t$). Note that the spectrum $(\zeta_i)_{i=1}^p$ characterizes the risk in both linear regression and random features regression, shown in Ildiz et al. (2024) and Simon et al. (2023), respectively. As the eigenvalues $(\lambda_i)_{i=1}^p$ are ordered, the $\zeta_i$'s are ordered as well, and the second part of the corollary identifies a threshold behavior: before the transition point $1 - \zeta_i = \Omega$, the entries of the surrogate are amplified w.r.t. the ground-truth parameter $\boldsymbol{\beta}_\star$, while they experience shrinkage after the transition. The threshold corresponds to the ratio of the sample second moment to the sample first moment of the random variable whose realization is given by $(1 - \zeta_i)$, and it arises from the optimization of the trade-off between the bias and variance terms in (10). Finally, the third part of the corollary shows that, unless the eigenvalues of the covariance matrix are constant, there is potential for improvement by tuning the surrogate parameter.

The intuition behind improving the performance of the standard target model by utilizing a surrogate parameter $\boldsymbol{\beta}^s$ different from $\boldsymbol{\beta}_\star$ is associated with the implicit regularization of the minimum norm interpolator in the over-parametrized region ($p > n$). As long as the covariance matrix eigenvalues are not constant, there is a way to mitigate the bias risk caused by the implicit regularization. This implicit regularization term is specific to the over-parametrized region. Indeed, in the next proposition (proved in Appendix A), we will show that the optimal surrogate parameter $\boldsymbol{\beta}^s$ is $\boldsymbol{\beta}_\star$ when the target model is under-parametrized:

**Proposition 2.** *The optimal surrogate parameter $\boldsymbol{\beta}^s$ that minimizes the asymptotic risk in the under-parametrized region ($n > p$) is equivalent to the ground truth parameter $\boldsymbol{\beta}_\star$. In other words, for any $\boldsymbol{\beta}^s$, the surrogate-to-target model cannot outperform the standard target model in the asymptotic risk.*

In other words, the result above shows that the improvement in the surrogate-to-target model compared to the standard target model is special to the over-parameterized region.

### 3.1 WEAK-TO-STRONG GENERALIZATION

To connect with knowledge distillation (Hinton et al., 2015) and weak-to-strong generalization (Burns et al., 2023), we allow the surrogate model to use fewer features, $p_s < p$, by introducing a mask operation $\mathcal{M}(\boldsymbol{x})$, where $\mathcal{M}(\boldsymbol{x}) \in \mathbb{R}^{p_s}$ selects $p_s$ features from the full set of $p$ features in $\boldsymbol{x} \in \mathbb{R}^p$. Alongside this mask, we adjust the distributions for both the surrogate and target models as

$$(\mathcal{M}(\tilde{\boldsymbol{x}}), \tilde{y}) \sim \mathcal{D}_s^{p_s} \text{ follows } \tilde{y} = \mathcal{M}(\tilde{\boldsymbol{x}})^\top \mathcal{M}(\boldsymbol{\beta}_\star) + \tilde{z}, \text{ where } \boldsymbol{\beta}_\star \in \mathbb{R}^p, \tilde{\boldsymbol{x}} \sim \mathcal{N}(\boldsymbol{0}, \boldsymbol{\Sigma}_s), \tilde{z} \sim \mathcal{N}(0, \sigma_s^2),$$

$$(\boldsymbol{x}, y^s) \sim \mathcal{D}_t^{p_s}(\boldsymbol{\beta}^s) \text{ follows } y^s = \mathcal{M}(\boldsymbol{x})^\top \boldsymbol{\beta}^s + z, \text{ where } \boldsymbol{\beta}^s \in \mathbb{R}^{p_s}, \boldsymbol{x} \sim \mathcal{N}(\boldsymbol{0}, \boldsymbol{\Sigma}_t), z \sim \mathcal{N}(0, \sigma_t^2).$$

Then, $\boldsymbol{\beta}^{s2t}$ is estimated based on the samples from $\mathcal{D}_t^{p_s}(\boldsymbol{\beta}^s)$, and the risk $\mathcal{R}(\boldsymbol{\beta}^{s2t})$ is still calculated with respect to the standard target model distribution $\mathcal{D}_t(\boldsymbol{\beta}_\star)$ as defined in (5). As we focus on analyzing the model shift case, we assume that the covariance matrices $\boldsymbol{\Sigma}_s$ and $\boldsymbol{\Sigma}_t$ are identical.

In this formulation, the surrogate model is considered *weak* because it has access to fewer features, while the target model is the *strong* model. We now address the following question: Can the surrogate-to-target model outperform the standard target model, provided in (6), in the absence of model shift ($\mathcal{M}(\boldsymbol{\beta}_\star) = \boldsymbol{\beta}^s$)?

---

[1]If not, there exists an orthogonal matrix $\boldsymbol{U} \in \mathbb{R}^{p \times p}$ s.t. $\boldsymbol{U}\boldsymbol{\Sigma}_t\boldsymbol{U}^\top$ is diagonal. Then, we can consider the covariance matrix as $\boldsymbol{U}\boldsymbol{\Sigma}_t\boldsymbol{U}^\top$ and the ground truth parameter as $\boldsymbol{U}\boldsymbol{\beta}_\star$, which behaves the same as the original parameters, see Observation 2.

The absence of model shift corresponds to the case where the surrogate model has infinitely many data. The next proposition provides a sufficient condition to answer the question above in the affirmative, and it derives the optimal selection of features.

**Proposition 3.** *Consider the target model in* (6), *assume that $\Sigma_t$ is diagonal, and recall the definitions of $\zeta_i$ and $\Omega$. Then, the following results hold:*

1. *If the mask operation $\mathcal{M}$ selects all the features that satisfy $1 - \zeta_i^2 > \Omega$, then the surrogate-to-target model outperforms the standard target model in the asymptotic risk in* (10).

2. *Let $\boldsymbol{M}$ represent the set of all possible $\mathcal{M}$, where $|\boldsymbol{M}| = 2^p$. The optimal $\mathcal{M}^*$ for the asymptotic risk in* (10) *within $\boldsymbol{M}$ is the one that selects all features satisfying $1 - \zeta_i^2 > \Omega$.*

The proof of Proposition 3 is provided in Appendix A, and the result can be extended to the non-asymptotic risk by applying Theorem 1. Similarly to Corollary 1, the result above identifies a threshold behavior: the entries of the surrogate are masked (i.e., set to 0) after the transition point $1 - \zeta_i^2 = \Omega$, while they coincide with the ground-truth parameter $\boldsymbol{\beta}_\star$ otherwise. The transition point changes with respect to Corollary 1 and, as $1 - \zeta_i^2 > 1 - \zeta_i$, it is shifted to the right: the optimal mask includes not only features whose magnitude increases, but also features whose magnitude decreases while selecting the optimal surrogate $\boldsymbol{\beta}^{s*}$.

In Figure 1a, we illustrate the optimal $\mathcal{M}$ and $\boldsymbol{\beta}^s$, showing that the threshold associated with the optimal $\mathcal{M}$ is larger than the threshold associated with the transition from amplification to shrinkage in the optimal $\boldsymbol{\beta}^s$. In addition, we note that the ratio between the green curve and the blue curve in Figure 1a is not monotone with respect to $\lambda_i$. In Figure 1b, we also present a comparison of their associated risks.

In Figure 2a, we examine the surrogate-to-target model in the context of image classification. Specifically, we fine-tune a pretrained ResNet-50 model (He et al., 2015) using both ground-truth labels and predictions from a surrogate (weak) model on the CIFAR-10 dataset (Krizhevsky & Hinton, 2009). The surrogate models are shallow, 3-layer convolutional neural networks with varying parameter sizes. In all cases, surrogate-to-target models consistently outperform surrogate models across different model sizes. However, in this setting, surrogate-to-target models do not outperform the standard target (strong) model. This is in agreement with the weak-to-strong results in Burns et al. (2023), where the GPT-4 model trained with GPT-2 labels performs comparably to GPT-3.5. The reason why the surrogate-to-target model underperforms the standard target model is the surrogate model is not able to follow the feature selection mechanism characterized in Proposition 3. This suggests that the feature selection mechanism is crucial for surpassing the performance of the standard target model. We provide further experimental details in Appendix A.2.

## 4 Fundamental limits and scaling laws

We now study the fundamental limits of the surrogate-to-target model with the optimal surrogate parameter $\boldsymbol{\beta}^{s*}$ (see Proposition 1) and the optimal mask operator $\mathcal{M}^*$ (see Proposition 3). Our analysis shows that, when eigenvalues ($\lambda_i$) and signal coefficients ($\lambda_i \beta_i^2$) follow a power law, the risk of the surrogate-to-target model under the optimal selection of the parameters $\boldsymbol{\beta}^{s*}$ and $\mathcal{M}^*$ scales the same as that of the target model (even though there is a strict improvement in the risk, as per Corollary 1). By Observation 1, this also indicates that the gain obtained by the covariance shift model, as outlined in Mallinar et al. (2024), does not change the scaling law. We start our analysis with the definition of the omniscient test risk estimate.

**Definition 2** (Omniscient test risk estimate). *Fix $p > n \geq 1$. Given a covariance $\Sigma = \boldsymbol{U} \operatorname{diag}(\lambda) \boldsymbol{U}^\top$, $\boldsymbol{\beta}_\star$, and the noise term $\sigma$, set $\bar{\boldsymbol{\beta}} = \boldsymbol{U}^\top \boldsymbol{\beta}_\star$ and define $\tau \in \mathbb{R}$ as the unique non-negative solution of $n = \sum_{i=1}^p \frac{\lambda_i}{\lambda_i + \tau}$. Then, the omniscient excess test risk estimate is the following:*

$$\mathcal{R}_{om}(\hat{\boldsymbol{\beta}}) \approx \mathbb{E}_{\hat{\boldsymbol{\beta}} \sim D(\boldsymbol{\beta}_\star)} \left[ (y - \boldsymbol{x}^\top \hat{\boldsymbol{\beta}})^2 \right] - \sigma^2 = \frac{\sigma^2 \Omega + \mathcal{B}(\bar{\boldsymbol{\beta}})}{1 - \Omega}, \tag{12}$$

$$\text{where} \quad \zeta_i = \frac{\tau}{\lambda_i + \tau}, \quad \Omega = \frac{1}{n} \sum_{i=1}^p (1 - \zeta_i)^2, \quad \mathcal{B}(\bar{\boldsymbol{\beta}}) = \sum_{i=1}^p \lambda_i \zeta_i^2 \bar{\beta}_i^2.$$

The above test risk estimate yields exact results (and, hence, $\approx$ in (12) becomes $=$) in the proportional limit via the analysis of Section 3. In other words, the omniscient test risk estimate is identical to $\bar{\mathcal{R}}_{\kappa,\sigma}^{s2t}(\Sigma, \boldsymbol{\beta}_\star, \boldsymbol{\beta}_\star)$, which can be derived by substituting $\boldsymbol{\beta}^s = \boldsymbol{\beta}_\star$ into Equation 10. Specifically, suppose that the empirical distributions

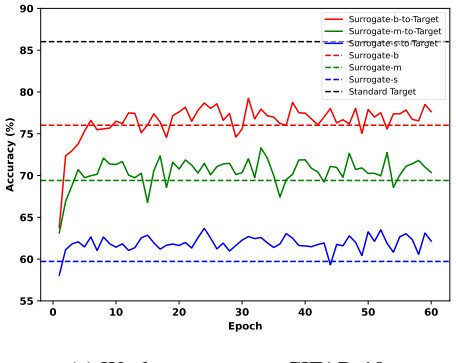
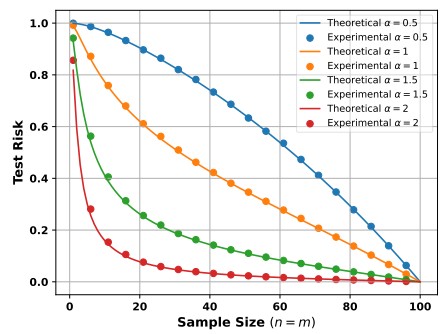

(a) Weak-to-strong on CIFAR-10      (b) Comparison of theoretical and experimental risks

Figure 2: **(a):** On the CIFAR-10 dataset, we fine-tune a ResNet50 model using the ground-truth labels (target) and the predictions of three weak convolutional models (surrogate) with different capacities: big (b), medium (m), and small (s). We observe that surrogate-to-target models consistently outperform surrogate models' accuracies, even though they are trained on the surrogate models' predictions. **(b):** We compare the experimental two-stage risk with our estimated theoretical risk. In the experimental setup, $p = 100$, and we vary $n = m$ from 1 to 100. Both feature covariances follow the power-law structure $\lambda_i = i^{-\alpha}$ for $\alpha = 0.5, 1, 1.5$ and 2; the ground truth parameter $\boldsymbol{\beta}_\star$ is specified as $\beta_i = 1$.

of $\bar{\boldsymbol{\beta}}$ and $\boldsymbol{\lambda}$ converge as $p \to \infty$ having fixed the ratio $p/n = \kappa$. Then, the risk obtained in Theorem 1 converges to the omniscient risk estimate given in (12), as proved in Appendix B. We will use this omniscient risk estimate in the limit of $p \to \infty$, as considered in several papers (Cui et al., 2022; Simon et al., 2024; Wei et al., 2022a). Yet, our empirical validations in Figure 1 demonstrate that this framework yields consistent results even when applied to scenarios with moderately sized $p$ and $n$.

Throughout the section, we analyze the case where the surrogate parameter $\boldsymbol{\beta}^s$ is given, therefore we need to take into account only the target covariance matrix $\boldsymbol{\Sigma}_t$. Without loss of generality, we assume that the covariance matrix $\boldsymbol{\Sigma}_t$ is diagonal by Observation 2. From now on, we will consider the particular case of power-law eigenstructure, that is $\boldsymbol{\Sigma}_{i,i} = \lambda_i = i^{-\alpha}$. The omniscient risk under this structure still depends on the parameters $\tau_t$ and $\Omega$, and the next proposition analyzes them asymptotically. Its proof is in Appendix B.

**Proposition 4** (Asymptotic analysis of $\tau_t$ and $\Omega$). *Let the covariance matrix $\boldsymbol{\Sigma} \in \mathbb{R}^{p \times p}$ be diagonal and $\boldsymbol{\Sigma}_{i,i} = \lambda_i = i^{-\alpha}$ for $1 < \alpha$. Recall from Definition 2 that, as $p \to \infty$, $\tau_t$ and $\Omega$ are given by the equations*

$$\sum_{i=1}^{\infty} \frac{\lambda_i}{\lambda_i + \tau_t} = n, \qquad n\Omega = \sum_{i=1}^{\infty} \left( \frac{i^{-\alpha}}{i^{-\alpha} + \tau_t} \right)^2 .$$

*Then, the following results hold*

$$\tau_t = cn^{-\alpha} \left( 1 + O(n^{-1}) \right), \qquad for \ c = \left( \frac{\pi}{\alpha \sin(\pi/\alpha)} \right)^\alpha ,$$

$$\Omega = \frac{\alpha - 1}{\alpha} - O(n^{-1}). \tag{13}$$

Recall from Corollary 1 and Proposition 3 that the cut-off indices for the optimal surrogate parameter $\boldsymbol{\beta}^{s*}$ and the optimal mask operation $\mathcal{M}^*$ are respectively $\zeta_i < 1 - \Omega$ and $\zeta_i^2 < 1 - \Omega$, which depend on $\tau_t, \Omega$. Armed with the asymptotic expressions in (13), we now identify the cut-off indices as a function of the sample size $n$.

**Proposition 5.** *Set the constants $C_1 := \dfrac{\alpha \sin(\pi/\alpha)}{\pi(\alpha - 1)^{1/\alpha}}$ and $C_2 := \dfrac{\alpha \sin(\pi/\alpha)}{\pi(\sqrt{\alpha} - 1)^{1/\alpha}}$ and assume the power-law eigenstructure $\boldsymbol{\Sigma}_{i,i} = \lambda_i = i^{-\alpha}$ for $1 < \alpha$. Let $\tau_t$ and $\Omega$ be the solutions given by Proposition 4 and define $\zeta_i = \frac{\tau_t}{\lambda_i + \tau_t}$. Then, the indices $i$ for which $\zeta_i < 1 - \Omega$ are $i < nC_1 + O(1)$; while the indices $i$ for which is $\zeta_i^2 < 1 - \Omega$ are $i < nC_2 + O(1)$.*

The result above is proved in Appendix B, and it shows that, as the sample size $n$ decreases, the cut-off indices of both the optimal surrogate parameter $\boldsymbol{\beta}^{s*}$ and the optimal mask $\mathcal{M}^*$ shift to the left linearly in $n$. This also implies that, with less data, optimal surrogate models tend to be more sparse.

Next, we address the question of how the excess test risk scales with respect to the sample size $n$, when the surrogate parameter is the optimal $\boldsymbol{\beta}^{s*}$. Specifically, Proposition 6 below shows that, under a power-law decay of both the eigenvalues ($\lambda_i$) and the signal coefficients ($\lambda_i \beta_i^2$), the excess test risk of the optimal surrogate-to-target model scales the same as the standard target model.

Before stating the result, we make a comment on the noise assumption needed to ensure that the scaling law of the excess test risk remains unaffected by the introduction of noise, which allows us to analyze the model's inherent error. Specifically, we choose the variance of the noise term $\sigma_t^2$ to be at most of the order of the scaling law of the excess test risks when $\sigma_t^2 = 0$. This corresponds to $\sigma_t^2 = O(n^{-\gamma})$, where $\gamma$ is the exponent of the scaling law in the noiseless setting. Conversely, a fixed noise variance $\sigma_t^2 = \Theta(1)$ that does not decay with $n$ would cause the noise to overshadow the uncaptured part of the signal, which scales down with $n$. In this unintended scenario, the noise would dominate our observations.

**Proposition 6** (Scaling law). *Let the covariance matrix $\Sigma_t$ be diagonal with eigenvalues $\lambda_i$, and let the ground-truth parameter $\boldsymbol{\beta}_\star$ have components $\beta_i$ corresponding to each feature. Assume that both eigenvalues $\lambda_i$ and signal coefficients $\lambda_i \beta_i^2$ follow a power-law decay, i.e., $\lambda_i \beta_i^2 = i^{-\beta}$ and $\lambda_i = i^{-\alpha}$ for $\alpha, \beta > 1$. Let the optimal surrogate parameter $\boldsymbol{\beta}^{s*}$ be given by Proposition 1 and define the minimum surrogate-to-target risk attained by $\boldsymbol{\beta}^{s*}$ as $\mathcal{R}_{om}^*(\boldsymbol{\beta}^{s2t}) = \min \mathcal{R}_{om}(\boldsymbol{\beta}^{s2t})$, where $\mathcal{R}_{om}(\boldsymbol{\beta}^{s2t})$ is described in Definition 2. Then, in the limit of $p \to \infty$, the excess test risk of the surrogate-to-target model with an optimal surrogate parameter scales the same as that of the standard target model. Specifically, we have*

$$\mathcal{R}_{om}^*(\boldsymbol{\beta}^{s2t}) = \Theta(n^{-(\beta-1)}) = \mathcal{R}_{om}(\boldsymbol{\beta}^t), \qquad \text{if } \beta < 2\alpha + 1,$$
$$\mathcal{R}_{om}^*(\boldsymbol{\beta}^{s2t}) = \Theta(n^{-2\alpha}) = \mathcal{R}_{om}(\boldsymbol{\beta}^t), \qquad \text{if } \beta > 2\alpha + 1.$$

Since $\mathcal{R}_{om}^*(\boldsymbol{\beta}^{s2t})$ is a lower bound on $\mathcal{R}_{om}(\boldsymbol{\beta}^{s2t})$, we have that the scaling law of the excess test risk of the surrogate-to-target model cannot be improved beyond that of the standard target model, even with the freedom to choose $\boldsymbol{\beta}^s$. This also indicates that the optimal selection of the mask does not improve the scaling law, see Proposition 7 in Appendix B for details.

The proof of Proposition 6 is deferred to Appendix B. Here, we note that we utilize the expression $\frac{\Omega}{1-\Omega}\left(\sum_{i=1}^p \lambda_i (\beta_i^s)^2 \zeta_i^2\right)$ as a lower bound for the risk, while proving the scaling law for the optimal surrogate-to-target model. Although this expression alone is insufficient to determine an asymptotic lower bound for an arbitrary $\boldsymbol{\beta}^s$, it becomes particularly useful when considering the optimal surrogate parameter $\boldsymbol{\beta}^{s*}$ provided in Proposition 1. We then leverage the fact that the optimal surrogate parameter yields the minimum test risk to characterize its asymptotic behavior.

Finally, we provide in Appendix B also a non-asymptotic analysis of $\tau_t$ and $\Omega$ (see Propositions 8 and 9, respectively), which complements the asymptotic one in Proposition 4 above. This allows us to characterize a region with finite $n$ and $p$ where the surrogate-to-target model strictly outperforms the standard target model, see Proposition 10.

## 5 RISK CHARACTERIZATION FOR THE TWO-STAGE MODEL

Until now, we have examined the behavior of the surrogate-to-target model when $\boldsymbol{\beta}^s$ is given. In this section, we characterize the non-asymptotic risk of the surrogate-to-target model when $\boldsymbol{\beta}^s$ is the solution of the surrogate problem (3) where $\kappa_s = p/m > 1$. Our analysis includes two cases: *(i)* the target model has infinitely many data ($n = \infty$), and *(ii)* the target model is overparametrized, i.e., $\kappa_t = p/n > 1$.

When the target model has infinitely many data, the estimate of the surrogate-to-target model $\boldsymbol{\beta}^{s2t}$ is equal to the estimate of the surrogate model $\boldsymbol{\beta}^s$. This means that the correct ground-truth parameter $\boldsymbol{\beta}_\star$ is estimated under a distribution $\mathcal{D}_s$ and tested under another distribution $\mathcal{D}_t(\boldsymbol{\beta}_\star)$, which is equivalent to the covariance shift model by definition. By Observation 1, the model shift in the surrogate-to-target model is equivalent to the covariance shift model and, hence, the analysis in Sections 3 and 4 is valid for this case.

Finally, we consider the case where the target model is overparametrized and begin with the following asymptotic risk definition.

**Definition 3.** *Recall the definition of $\tau_t$ and $\gamma_t$ in Theorem 1. Let $\kappa_s = p/m > 1$ and define $\tau_s \in \mathbb{R}$ similarly to $\tau_t$. We define the random variable $X^s_{\kappa_s, \sigma^2_s}$ based on $\boldsymbol{g}_s \sim \mathcal{N}(0, \boldsymbol{I})$ and the function $\gamma_s : \mathbb{R}^p \to \mathbb{R}$ as follows:*

$$X^s_{\kappa_s, \sigma^2_s}(\boldsymbol{\Sigma}_s, \boldsymbol{\beta}_\star, \boldsymbol{g}_s) := (\boldsymbol{\Sigma}_s + \tau_s \boldsymbol{I})^{-1} \boldsymbol{\Sigma}_s \left[ \boldsymbol{\beta}_\star + \frac{\boldsymbol{\Sigma}_s^{-1/2} \gamma_s(\boldsymbol{\beta}_\star) \boldsymbol{g}_s}{\sqrt{p}} \right]$$

$$\gamma^2_s(\boldsymbol{\beta}_\star) := \kappa_s \left( \sigma^2_s + \mathbb{E}_{\boldsymbol{g}_s}[\|\boldsymbol{\Sigma}_s^{1/2}(X^s_{\kappa_s, \sigma^2_s}(\boldsymbol{\Sigma}_s, \boldsymbol{\beta}_\star, \boldsymbol{g}_s) - \boldsymbol{\beta}_\star)\|^2_2] \right).$$

*Let $\dot{k} = (\kappa_s, \kappa_t)$, $\dot{\boldsymbol{\Sigma}} = (\boldsymbol{\Sigma}_s, \boldsymbol{\Sigma}_t)$, and $\dot{\sigma} = (\sigma^2_s, \sigma^2_t)$. Then, we define the asymptotic risk estimate as*

$$\bar{\mathcal{R}}_{\dot{k}, \dot{\sigma}}(\dot{\boldsymbol{\Sigma}}, \boldsymbol{\beta}_\star) = \|\boldsymbol{\Sigma}_t^{1/2} \left( \boldsymbol{I} - (\boldsymbol{\Sigma}_t + \tau_t \boldsymbol{I})^{-1} \boldsymbol{\Sigma}_t (\boldsymbol{\Sigma}_s + \tau_s \boldsymbol{I})^{-1} \boldsymbol{\Sigma}_s \right) \boldsymbol{\beta}_\star\|^2_2 + \frac{\mathbb{E}_{\boldsymbol{\beta}^s \sim X^s_{\kappa_s, \sigma^2_s}}[\gamma^2_t(\boldsymbol{\beta}^s)]}{p} \operatorname{tr}\left( \boldsymbol{\Sigma}_t^2 (\boldsymbol{\Sigma}_t + \tau_t \boldsymbol{I})^{-2} \right)$$

$$+ \frac{\gamma^2_s(\boldsymbol{\beta}_\star)}{p} \operatorname{tr}\left( \boldsymbol{\Sigma}_s^{1/2} (\boldsymbol{\Sigma}_s + \tau_s \boldsymbol{I})^{-1} \boldsymbol{\Sigma}_t (\boldsymbol{\Sigma}_t + \tau_t \boldsymbol{I})^{-1} \boldsymbol{\Sigma}_t (\boldsymbol{\Sigma}_t + \tau_t \boldsymbol{I})^{-1} \boldsymbol{\Sigma}_t (\boldsymbol{\Sigma}_s + \tau_s \boldsymbol{I})^{-1} \boldsymbol{\Sigma}_s^{1/2} \right).$$

The non-asymptotic characterization of the risk is stated below and proved in Appendix C, which also contains a closed-form expression for $\mathbb{E}_{\boldsymbol{\beta}^s \sim X^s_{\kappa_s, \sigma^2_s}}[\gamma^2_t(\boldsymbol{\beta}^s)]$ (see Lemma 1).

**Theorem 2.** *Suppose that, for some constant $M_t > 1$, we have $1/M_t \leq \kappa_s, \sigma^2_s, \kappa_t, \sigma^2_t \leq M_t$ and $\|\boldsymbol{\Sigma}_s\|_{op}, \|\boldsymbol{\Sigma}_s^{-1}\|_{op}, \|\boldsymbol{\Sigma}_t\|_{op}, \|\boldsymbol{\Sigma}_s^{-1}\|_{op} \leq M_t$. Consider the surrogate-to-target model defined in Section 2, and let $\mathcal{R}(\boldsymbol{\beta}^{s2t})$ represent its risk when $\boldsymbol{\beta}_\star$ is given. Recall the definition of $\dot{\boldsymbol{\Sigma}}, \dot{k}, \dot{\sigma}$ and $\bar{\mathcal{R}}_{\dot{k}, \dot{\sigma}}$ in Definition 3. Then, there exists a constant $C = C(M_t)$ such that for any $\varepsilon \in (0, 1/2]$, the following holds when $R + 1 < M_t$:*

$$\sup_{\boldsymbol{\beta}_\star \in \boldsymbol{B}_p(R)} \mathbb{P}\left( \left| \mathcal{R}(\boldsymbol{\beta}^{s2t}) - \bar{\mathcal{R}}_{\dot{k}, \dot{\sigma}}(\dot{\boldsymbol{\Sigma}}, \boldsymbol{\beta}_\star) \right| \geq \varepsilon \right) \leq C p e^{-p\varepsilon^4/C}.$$

In the proof of Theorem 2, we apply the distributional characterization of the minimum norm interpolator (Han & Xu, 2023) twice. The main technical difficulty is to satisfy the Lipschitz condition of the distributional characterization in the second step, which is handled by bounding the intermediate step's interpolator in a ball.

We implement the surrogate-to-target model with a synthetic dataset and demonstrate that our risk characterization agrees well with the experimental two-stage linear regression in Figure 2b. Furthermore, we extend our analysis of the two-stage model to the under-parametrized region in Appendix C.1. This extension enables us to analyze Section 3.1 thoroughly because this analysis allows the surrogate model to have finite data.

## 6 CONCLUDING REMARKS

We have provided a sharp characterization of knowledge distillation for high-dimensional linear regression when labels are generated by a surrogate model and, additionally, characterized the risk of the two-stage process where the surrogate model is the outcome of an initial ERM. These results shed light on the form of the optimal surrogate model, reveal an amplify-to-shrink phase transition as a function of the eigenspectrum, and draw connections to weak-to-strong generalization. Specifically, we have shown that the labels coming from the optimal surrogate model strictly allow for improving the performance of the target model, unless the covariance is a multiple of the identity. However, even though there is a strict improvement in the risk, the scaling behavior of the two-stage process with labels coming from the optimal surrogate model remains unchanged compared to the standard target model that utilizes ground-truth labels.

We outline three interesting directions for future research. The first is to extend the two-stage process to multiple stages, establishing whether this further improves the risk. The second is to apply the two-stage learning to data pruning, using the surrogate model to decide whether to keep or discard each $(\boldsymbol{x}, y)$ pair during the training of the target model. The third is to go beyond linear regression towards neural network models. In this regard, the precise asymptotics of the test error of the ERM solution were provided by (Mei & Montanari, 2022) for the random features model and by (Montanari & Zhong, 2022) for two-layer neural networks in the NTK regime. However, a non-asymptotic characterization (similar to that given by (Han & Xu, 2023) for linear regression) remains an open problem. The resolution of this open problem, as well as the analysis of the phenomena of knowledge distillation and weak-to-strong generalization, represent exciting future directions.

ACKNOWLEDGEMENTS

M.E.I., H.A.G., E.O.T., S.O. are supported by the NSF grants CCF-2046816, CCF-2403075, the Office of Naval Research grant N000142412289, an OpenAI Agentic AI Systems grant, and gifts by Open Philanthropy and Google Research. M. M. is funded by the European Union (ERC, INF$^2$, project number 101161364). Views and opinions expressed are however those of the author(s) only and do not necessarily reflect those of the European Union or the European Research Council Executive Agency. Neither the European Union nor the granting authority can be held responsible for them.

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

## A    Proofs for Section 3

Below, we summarize the notations used throughout this paper:

| Category | Symbol | Description |
|---|---|---|
| Risk | $\mathcal{R}(\cdot)$ | Excess population test risk. |
| | $\bar{\mathcal{R}}(\cdot)$ | Asymptotic excess test risk. |
| | $\mathcal{R}_{om}(\cdot)$ | Asymptotic omniscient excess test risk, which is equivalent to the two-stage risk when $\beta^s = \beta_\star$. |
| Parameters | $\beta^s$ | Estimator of the surrogate model after the stage 1. |
| | $\beta^{s2t}$ | Estimator of the target model after the stage 2. |
| | $\beta^t$ | Estimator of the standard target model. |
| | $\beta_\star$ | Ground truth parameter. |
| Covariances | $\Sigma_s$ | Covariance matrix of the data distribution used in surrogate model. |
| | $\Sigma_t$ | Covariance matrix of the data distribution used in target and standard target models. |
| Variables | $\lambda_i$ | $i$'th eigenvalue of the covariance matrix $\Sigma_t$ in decreasing order. |
| | $\tau_t$ | The unique solution to the fixed point equation: $\kappa_t^{-1} = \frac{1}{p}\mathrm{tr}\left((\Sigma_t + \tau_t I)^{-1}\Sigma_t\right)$. |
| | $\tau_s$ | The unique solution to the fixed point equation: $\kappa_s^{-1} = \frac{1}{p}\mathrm{tr}\left((\Sigma_s + \tau_s I)^{-1}\Sigma_s\right)$. |
| | $\zeta_i$ | Covariance statistics given by $\zeta_i = \frac{\tau_t}{\lambda_i + \tau_t}$ . |
| | $\Omega$ | $\Omega = \frac{\mathrm{tr}(\Sigma_t^2(\Sigma_t+\tau_t I)^{-2})}{n} = \frac{\sum_{j=1}^p (1-\zeta_j)^2}{\sum_{j=1}^p (1-\zeta_j)}$ . |
| | $X^s_{\kappa_s, \sigma_s^2}(\Sigma_s, \beta_\star, g_s)$ | Asymptotic characterization of the minimum $\ell_2$ interpolator obtained after training surrogate model. |
| | $X^t_{\kappa_t, \sigma_t^2}(\Sigma_t, \beta^s, g_t)$ | Asymptotic characterization of the minimum $\ell_2$ interpolator obtained after training target model. |
| | $\gamma_s^2(\beta_\star)$ | $\gamma_s^2(\beta_\star) = \kappa_s\left(\sigma_s^2 + \mathbb{E}_{g_s}[\|\Sigma_s^{1/2}(X^s_{\kappa_s, \sigma_s^2}(\Sigma_s, \beta_\star, g_s) - \beta_\star)\|_2^2]\right)$. |
| | $\gamma_t^2(\beta^s)$ | $\gamma_t^2(\beta^s) = \kappa_t\left(\sigma_t^2 + \bar{\mathcal{R}}^{s2t}_{\kappa_t, \sigma_t}(\Sigma_t, \beta^s, \beta^s)\right)$. |

Table 1: Summary of notations used in the paper.

**Observation 1.** *The model shift in the surrogate-to-target model is equivalent to the covariance shift model (Mallinar et al., 2024). Formally, given any $\beta_\star \in \mathbb{R}^p$ and any jointly diagonalizable covariance matrices $\Sigma_s, \Sigma_t \in \mathbb{R}^{p \times p}$, there exists a unique $\beta^s \in \mathbb{R}^p$ such that the risk of the surrogate-to-target problem $\mathcal{R}(\beta^{s2t})$ with $(\beta_\star, \beta^s, \Sigma_t)$ is equivalent to the risk of the covariance shift model $\mathcal{R}^{cs}(\hat{\beta})$ with $(\beta_\star, \Sigma_s, \Sigma_t)$.*

*Proof.* By Observation 2, we assume that $\Sigma_t$ and $\Sigma_s$ are diagonal matrices. As $\Sigma_t$ and $\Sigma_s$ are jointly diagonalizable, there exists a unique diagonal matrix $A \in \mathbb{R}^{p \times p}$ such that

$$\Sigma_s = A^\top \Sigma_t A.$$

Then, consider the model shift discussed in Section 3. Take the case where $\beta^s = A\beta_\star$ and labels are generated as $y = x^\top \beta^s + z$, where $x \sim \mathcal{N}(0, \Sigma_t)$ and $z \sim \mathcal{N}(0, \sigma_t^2)$. This is equivalent to the case where $y = (x^\top A)\beta_\star + z = \bar{x}^\top \beta_\star + z$ such that $x \sim \mathcal{N}(0, \Sigma_s)$ and $z \sim \mathcal{N}(0, \sigma_t^2)$. Note that *(i)* the transformed inputs and the labels are identical in both scenarios, and *(ii)* the estimators are computed in the same way. Thus, it follows that the risks $\mathcal{R}(\beta^{s2t})$ and $\mathcal{R}^{cs}(\hat{\beta})$ are equivalent. The other way follows from an almost identical argument.    □

**Observation 2.** *For any covariance matrix $\Sigma \in \mathbb{R}^{p \times p}$, there exists an orthonormal matrix $U \in \mathbb{R}^{p \times p}$ such that the transformation of $x \to U^\top x$ and $\beta \to U^\top \beta$ does not affect the labels $y$ but ensures that the covariance matrix is diagonal.*

*Proof.* Since the covariance matrix $\Sigma$ is PSD, its unit-norm eigenvectors are orthogonal. Consider the matrix $U$ whose columns are the eigenvectors of $\Sigma$. Then, $\Sigma$ can be expressed as $\Sigma = U\Lambda U^\top$, where $\Lambda$ is the diagonal

matrix containing the eigenvalues of $\boldsymbol{\Sigma}$. Consider now the transformation

$$z = \boldsymbol{U}^\top \boldsymbol{x} \implies \mathbb{E}\left[\boldsymbol{z}\boldsymbol{z}^\top\right] = \mathbb{E}\left[\boldsymbol{U}^\top \boldsymbol{x}\boldsymbol{x}^\top \boldsymbol{U}\right] = \boldsymbol{U}^\top \mathbb{E}\left[\boldsymbol{x}\boldsymbol{x}^\top\right]\boldsymbol{U} = \boldsymbol{U}^\top \boldsymbol{U}\boldsymbol{\Lambda}\boldsymbol{U}^\top \boldsymbol{U} = \boldsymbol{\Lambda}.$$

In this way, the covariance matrix is diagonalized. Thus, the transformation $(\boldsymbol{x}, \boldsymbol{\beta}_\star) \to (\boldsymbol{U}^\top \boldsymbol{x}, \boldsymbol{U}^\top \boldsymbol{\beta}_\star)$ works as intended since the labels are preserved. $\qquad\square$

**Definition 1.** *Let $\kappa_t = p/n > 1$ and $\tau_t \in \mathbb{R}$ be the unique solution of the following equation*

$$\kappa_t^{-1} = \frac{1}{p}\mathrm{tr}\left((\boldsymbol{\Sigma}_t + \tau_t \boldsymbol{I})^{-1}\boldsymbol{\Sigma}_t\right). \tag{7}$$

*Let $\boldsymbol{\theta}_1 := (\boldsymbol{\Sigma}_t + \tau_t \boldsymbol{I})^{-1}\boldsymbol{\Sigma}_t$ and $\boldsymbol{\theta}_2 := (\boldsymbol{\Sigma}_t + \tau_t \boldsymbol{I})^{-1}\boldsymbol{\Sigma}_t^{1/2}\frac{\boldsymbol{g}_t}{\sqrt{p}}$ where $\boldsymbol{g}_t \sim \mathcal{N}(\boldsymbol{0}, \boldsymbol{I}_p)$. Now, define the asymptotic characterization of the minimum $\ell_2$-norm interpolator and the function $\gamma_t : \mathbb{R}^p \to \mathbb{R}$ as the following fixed point equation based on the asymptotic risk*

$$X^t_{\kappa_t, \sigma_t^2}(\boldsymbol{\Sigma}_t, \boldsymbol{\beta}^s, \boldsymbol{g}_t) := \boldsymbol{\theta}_1 \boldsymbol{\beta}^s + \gamma_t(\boldsymbol{\beta}^s)\boldsymbol{\theta}_2, \tag{8}$$

$$\gamma_t^2(\boldsymbol{\beta}^s) := \kappa_t\left(\sigma_t^2 + \bar{\mathcal{R}}^{s2t}_{\kappa_t, \sigma_t}(\boldsymbol{\Sigma}_t, \boldsymbol{\beta}^s, \boldsymbol{\beta}^s)\right), \tag{9}$$

*where the asymptotic risk is defined as*

$$\begin{aligned}
\bar{\mathcal{R}}^{s2t}_{\kappa_t, \sigma_t}(\boldsymbol{\Sigma}_t, \boldsymbol{\beta}_\star, \boldsymbol{\beta}^s) &:= \mathbb{E}_{\boldsymbol{g}_t}\left[\|\boldsymbol{\Sigma}_t^{1/2}(X^t_{\kappa_t, \sigma_t^2}(\boldsymbol{\Sigma}_t, \boldsymbol{\beta}^s, \boldsymbol{g}_t) - \boldsymbol{\beta}_\star)\|_2^2\right] \\
&= \underbrace{\|\boldsymbol{\Sigma}_t^{1/2}(\boldsymbol{\theta}_1\boldsymbol{\beta}^s - \boldsymbol{\beta}_\star)\|_2^2}_{(a)} + \underbrace{\gamma_t^2(\boldsymbol{\beta}^s)\mathbb{E}_{\boldsymbol{g}_t}[\boldsymbol{\theta}_2^\top \boldsymbol{\Sigma}_t \boldsymbol{\theta}_2]}_{(b)} \\
&= (\boldsymbol{\beta}^s - \boldsymbol{\beta}_\star)^\top \boldsymbol{\theta}_1^\top \boldsymbol{\Sigma}_t \boldsymbol{\theta}_1(\boldsymbol{\beta}^s - \boldsymbol{\beta}_\star) + \gamma_t^2(\boldsymbol{\beta}^s)\mathbb{E}_{\boldsymbol{g}_t}[\boldsymbol{\theta}_2^\top \boldsymbol{\Sigma}_t \boldsymbol{\theta}_2] \\
&\quad + \boldsymbol{\beta}_\star^\top(\boldsymbol{I} - \boldsymbol{\theta}_1)^\top \boldsymbol{\Sigma}_t(\boldsymbol{I} - \boldsymbol{\theta}_1)\boldsymbol{\beta}_\star - 2\boldsymbol{\beta}_\star^\top(\boldsymbol{I} - \boldsymbol{\theta}_1)^\top \boldsymbol{\Sigma}_t \boldsymbol{\theta}_1(\boldsymbol{\beta}^s - \boldsymbol{\beta}_\star).
\end{aligned} \tag{10}$$

**Theorem 1.** *Suppose that, for some constant $M_t > 1$, we have $1/M_t \le \kappa_t, \sigma_t^2 \le M_t$ and $\|\boldsymbol{\Sigma}_t\|_{op}, \left\|\boldsymbol{\Sigma}_t^{-1}\right\|_{op} \le M_t$. Recall from (5) that $\mathcal{R}(\boldsymbol{\beta}^{s2t})$ represents the risk of the surrogate-to-target model given $\boldsymbol{\beta}^s$. Then, there exists a constant $C = C(M_t)$ such that, for any $\varepsilon \in (0, 1/2]$, the following holds with $R + 1 < M_t$:*

$$\sup_{\boldsymbol{\beta}_\star, \boldsymbol{\beta}^s \in \boldsymbol{B}_p(R)} \mathbb{P}(|\mathcal{R}(\boldsymbol{\beta}^{s2t}) - \bar{\mathcal{R}}^{s2t}_{\kappa_t, \sigma_t}(\boldsymbol{\Sigma}_t, \boldsymbol{\beta}_\star, \boldsymbol{\beta}^s)| \ge \varepsilon) \le Cp e^{-p\varepsilon^4/C}. \tag{11}$$

*Proof.* Even though the claim readily follows from Theorem 2, we give a proof for the sake of completeness.

Define a function $f_1 : \mathbb{R}^p \to \mathbb{R}$ as $f_1(\boldsymbol{x}) = \|\boldsymbol{\Sigma}_t^{1/2}(\boldsymbol{x} - \boldsymbol{\beta}_\star)\|_2^2$. The gradient of this function is

$$\|\nabla f_1(\boldsymbol{x})\|_2 = \|2\boldsymbol{\Sigma}_t(\boldsymbol{x} - \boldsymbol{\beta}_\star)\|_2 \le 2\|\boldsymbol{\Sigma}_t\|_{op}\|\boldsymbol{x} - \boldsymbol{\beta}_\star\|_2.$$

Using Corollary 2, there exists an event $E$ with $\mathbb{P}(E^c) \le C_t e^{-p/C_t}$ where $C_t = C_t(M_t, \frac{M_t - R}{2})$ (with the definition of $M_t$ in Corollary 2), such that $f_1(\boldsymbol{\beta}^{s2t})$ is $2M_t^2$-Lipschitz if $\boldsymbol{\beta}_\star, \boldsymbol{\beta}^s \in \boldsymbol{B}_p(R)$. Applying Theorem 4 on the target model, there exists a constant $\bar{C}_s = \bar{C}_s(M_t)$ such that for any $\varepsilon \in (0, 1/2]$, we obtain

$$\sup_{\boldsymbol{\beta}^s \in \boldsymbol{B}(\frac{M_t + R}{2})} \mathbb{P}\left(\left|f(\boldsymbol{\beta}^{s2t}) - \mathbb{E}_{\boldsymbol{g}_t}[f(X^t_{\kappa_t, \sigma_t^2}(\boldsymbol{\Sigma}_t, \boldsymbol{\beta}^s, \boldsymbol{g}_t))]\right| \ge \varepsilon\right) \le Cp e^{-p\varepsilon^4/C}, \tag{14}$$

where $f(\boldsymbol{\beta}^{s2t}) = \mathcal{R}(\boldsymbol{\beta}^{s2t})$ and

$$X^t_{\kappa_t, \sigma_t^2}(\boldsymbol{\Sigma}_t, \boldsymbol{\beta}^s, \boldsymbol{g}_t) = (\boldsymbol{\Sigma}_t + \tau_t \boldsymbol{I})^{-1}\boldsymbol{\Sigma}_t\left[\boldsymbol{\beta}^s + \frac{\boldsymbol{\Sigma}_t^{-1/2}\gamma_t(\boldsymbol{\beta}^s)\boldsymbol{g}_t}{\sqrt{p}}\right].$$

Furthermore,

$$\begin{aligned}
\mathbb{E}_{\boldsymbol{g}_t}\left[f(X^s_{\kappa_t, \sigma_t^2}(\boldsymbol{\Sigma}_t, \boldsymbol{\beta}^s, \boldsymbol{g}_t))\right] &= \mathbb{E}_{\boldsymbol{g}_t}\left[\|\boldsymbol{\Sigma}_t^{1/2}(\boldsymbol{\theta}_1(\boldsymbol{\beta}^s - \boldsymbol{\beta}_\star) - (\boldsymbol{I} - \boldsymbol{\theta}_1)\boldsymbol{\beta}_\star + \boldsymbol{\theta}_2\gamma_t(\boldsymbol{\beta}^s))\|_2^2\right] \\
&= (\boldsymbol{\beta}^s - \boldsymbol{\beta}_\star)^\top \boldsymbol{\theta}_1^\top \boldsymbol{\Sigma}_t \boldsymbol{\theta}_1(\boldsymbol{\beta}^s - \boldsymbol{\beta}_\star) + \gamma_t^2(\boldsymbol{\beta}^s)\mathbb{E}_{\boldsymbol{g}_t}[\boldsymbol{\theta}_2^\top \boldsymbol{\Sigma}_t \boldsymbol{\theta}_2] \\
&\quad + \boldsymbol{\beta}_\star^\top(\boldsymbol{I} - \boldsymbol{\theta}_1)^\top \boldsymbol{\Sigma}_t(\boldsymbol{I} - \boldsymbol{\theta}_1)\boldsymbol{\beta}_\star - 2\boldsymbol{\beta}_\star^\top(\boldsymbol{I} - \boldsymbol{\theta}_1)^\top \boldsymbol{\Sigma}_t \boldsymbol{\theta}_1(\boldsymbol{\beta}^s - \boldsymbol{\beta}_\star),
\end{aligned} \tag{15}$$

where $\boldsymbol{\theta}_1 := (\boldsymbol{\Sigma}_t + \tau_t \boldsymbol{I})^{-1}\boldsymbol{\Sigma}_t$ and $\boldsymbol{\theta}_2 := (\boldsymbol{\Sigma}_t + \tau_t \boldsymbol{I})^{-1}\boldsymbol{\Sigma}_t^{1/2}\frac{\boldsymbol{g}_t}{\sqrt{p}}$. This completes the proof. $\qquad\square$

**Proposition 1.** *Let* $\Omega = \frac{\text{tr}(\Sigma_t^2(\Sigma_t + \tau_t I)^{-2})}{n}$. *The optimal surrogate* $\beta^s$ *minimizing the asymptotic risk in* (10) *is*

$$\beta^{s*} = \left((\Sigma_t + \tau_t I)^{-1}\Sigma_t + \frac{\Omega\tau_t^2}{1 - \Omega}\Sigma_t^{-1}(\Sigma_t + \tau_t I)^{-1}\right)^{-1}\beta_\star.$$

*Proof.* We have that

$$\mathbb{E}_{g_t}\left[f(X^t_{\kappa_t,\sigma_t^2}(\Sigma_t, \beta^s, g_t))\right] = \mathbb{E}_{g_t}\left[\|\Sigma_t^{1/2}(\theta_1(\beta^s - \beta_\star) - (I - \theta_1)\beta_\star + \theta_2\gamma_t(\beta^s))\|_2^2\right]$$

$$= (\beta^s - \beta_\star)^\top\theta_1^\top\Sigma_t\theta_1(\beta^s - \beta_\star) + \gamma_t^2(\beta^s)\mathbb{E}_{g_t}[\theta_2^\top\Sigma_t\theta_2]$$
$$+ \beta_\star^\top(I - \theta_1)^\top\Sigma_t(I - \theta_1)\beta_\star - 2\beta_\star^\top(I - \theta_1)^\top\Sigma_t\theta_1(\beta^s - \beta_\star),$$

where $\theta_1 := (\Sigma_t + \tau_t I)^{-1}\Sigma_t$, $\theta_2 := (\Sigma_t + \tau_t I)^{-1}\Sigma_t^{1/2}\frac{g_t}{\sqrt{p}}$. Recall from (9) that

$$\gamma_t^2(\beta^s) = \kappa_t\left(\sigma_t^2 + \bar{\mathcal{R}}^{s2t}_{\kappa_t,\sigma_t}(\Sigma_t, \beta^s, \beta^s)\right)$$
$$= \kappa_t\left(\sigma_t^2 + \gamma_t^2(\beta^s)\mathbb{E}_{g_t}[\theta_2^\top\Sigma_t\theta_2] + (\beta^s)^\top(I - \theta_1)^\top\Sigma_t(I - \theta_1)\beta^s\right).$$

This implies that

$$\gamma_t^2(\beta^s) = \kappa_t\frac{\sigma_t^2 + (\beta^s)^\top(I - \theta_1)^\top\Sigma_t(I - \theta_1)\beta^s}{1 - \kappa_t\mathbb{E}_{g_t}[\theta_2^\top\Sigma_t\theta_2]} \tag{16}$$

$$\overset{(a)}{=} \frac{\sigma_t^2 + \tau_t^2\|\Sigma_t^{1/2}(\Sigma_t + \tau_t I)^{-1}\beta^s\|_2^2}{1 - \frac{1}{n}\text{tr}\left((\Sigma_t + \tau_t I)^{-2}\Sigma_t^2\right)},$$

where (*a*) follows from the fact that $I - \theta_1 = I - (\Sigma_t + \tau_t I)^{-1}\Sigma_t = \tau_t(\Sigma_t + \tau_t I)^{-1}$ and $\kappa_t\mathbb{E}_{g_t}[\theta_2^\top\Sigma_t\theta_2] = \frac{1}{n}\text{tr}\left((\Sigma_t + \tau_t I)^{-2}\Sigma_t^2\right)$.

In order to optimize this with respect to $\beta^s$, let's take the derivative:

$$\frac{\partial}{\partial\beta^s}\mathbb{E}_{g_t}\left[f(X^t_{\kappa_t,\sigma_t^2}(\Sigma_t, \beta^s, g_t))\right]$$

$$= 2\theta_1^\top\Sigma_t\theta_1(\beta^s - \beta_\star) - 2\theta_1^\top\Sigma_t(I - \theta_1)\beta_\star + 2\frac{\kappa_t\tau_t^2}{1 - \Omega}\Sigma_t(\Sigma_t + \tau_t I)^{-2}\beta^s\frac{\text{tr}\left(\Sigma_t^2(\Sigma_t + \tau_t I)^{-2}\right)}{p}$$

$$= 2\theta_1^\top\Sigma_t\theta_1\beta^s - 2\Sigma_t\theta_1\beta_\star + 2\frac{\kappa_t\tau_t^2}{1 - \Omega}\Sigma_t(\Sigma_t + \tau_t I)^{-2}\beta^s\frac{\text{tr}\left(\Sigma_t^2(\Sigma_t + \tau_t I)^{-2}\right)}{p}$$

$$= 2\theta_1^\top\Sigma_t\theta_1\beta^s - 2\Sigma_t\theta_1\beta_\star + 2\frac{\kappa_t\tau_t^2}{1 - \Omega}\Sigma_t(\Sigma_t + \tau_t I)^{-2}\beta^s\frac{n\Omega}{p}$$

$$\implies \theta_1^\top\Sigma_t\theta_1\beta^{s*} - \Sigma_t\theta_1\beta_\star + \frac{\Omega\tau_t^2}{1 - \Omega}\Sigma_t(\Sigma_t + \tau_t I)^{-2}\beta^{s*} = 0$$

$$\implies (\theta_1^\top\Sigma_t\theta_1 + \frac{\Omega\tau_t^2}{1 - \Omega}\theta_1(\Sigma_t + \tau_t I)^{-1})\beta^{s*} = \Sigma_t\theta_1\beta_\star$$

Hence, the claimed result follows. $\square$

**Corollary 1.** *Without loss of generality, suppose that* $\Sigma_t$ *is diagonal.*[2] *Let* $(\lambda_i)_{i=1}^p$ *be the eigenvalues of* $\Sigma_t$ *in non-increasing order and let* $\zeta_i = \frac{\tau_t}{\lambda_i + \tau_t}$ *for* $i \in [p]$. *Then, the following results hold:*

1. $\beta_i^{s*} = (\beta_*)_i\left((1 - \zeta_i) + \zeta_i\frac{\Omega}{1 - \Omega}\frac{\zeta_i}{1 - \zeta_i}\right)^{-1}$ *for every* $i \in [p]$.

---

[2]If not, there exists an orthogonal matrix $U \in \mathbb{R}^{p\times p}$ s.t. $U\Sigma_t U^\top$ is diagonal. Then, we can consider the covariance matrix as $U\Sigma_t U^\top$ and the ground truth parameter as $U\beta_\star$, which behaves the same as the original parameters, see Observation 2.

2. $|\beta_i^{s*}| > |(\beta_*)_i|$ if and only if $1 - \zeta_i > \Omega = \frac{\sum_{j=1}^p (1-\zeta_j)^2}{\sum_{j=1}^p (1-\zeta_j)}$ for every $i \in [p]$.

3. $\boldsymbol{\beta}^{s*} = \boldsymbol{\beta}_\star$ if and only if the covariance matrix $\boldsymbol{\Sigma}_t = c\boldsymbol{I}$ for some $c \in \mathbb{R}$.

*Proof.* When the definition of $\zeta_i$ and $\Omega$ is plugged in Proposition 1, the first claim is obtained. Using the diagonalization assumption on $\boldsymbol{\Sigma}_t$, let's analyze only the $i$-th component of the optimal surrogate given in Proposition 1:

$$\beta_i^{s*} = \frac{1}{\frac{\lambda_i}{\lambda_i + \tau_t} + \frac{\Omega}{1-\Omega}\frac{\tau_t^2}{\lambda_i(\lambda_i + \tau_t)}}(\beta_*)_i$$

$$\iff \beta_i^{s*} = \frac{\frac{\lambda_i}{\lambda_i + \tau_t}}{\left(\frac{\lambda_i}{\lambda_i + \tau_t}\right)^2 + \frac{\Omega}{1-\Omega}\left(\frac{\tau_t}{\lambda_i + \tau_t}\right)^2}(\beta_*)_i$$

$$\iff \beta_i^{s*} = (\beta_*)_i \frac{(1-\zeta_i)}{(1-\zeta_i)^2 + \frac{\Omega}{1-\Omega}\zeta_i^2}$$

$$\iff \beta_i^{s*} = (\beta_*)_i \frac{1}{(1-\zeta_i) + \frac{\Omega}{1-\Omega}\frac{\zeta_i}{1-\zeta_i}\zeta_i}.$$

It's now clear that $\zeta_i > 1 - \Omega$ if and only if $|\beta_i^{s*}| < |(\beta_*)_i|$.

Let's now check when the ratio between them is 1. Algebraic manipulations give:

$$\frac{(1-\zeta_i)}{(1-\zeta_i)^2 + \frac{\Omega}{1-\Omega}\zeta_i^2} = 1$$

$$\iff (1-\zeta_i) - (1-\zeta_i)^2 = \frac{\Omega}{1-\Omega}\zeta_i^2$$

$$\iff \zeta_i = 1 - \Omega \iff 1 - \zeta_i = \Omega \text{ where } \Omega = \frac{\sum_{i=1}^p (1-\zeta_i)^2}{\sum_{i=1}^p (1-\zeta_i)}.$$

This gives that $\boldsymbol{\beta}^{s*} = \boldsymbol{\beta}_\star$ if all $\zeta_i$'s are equal, which implies that all $\lambda_i$'s are equal. Concluding, the covariance matrix is a multiple of the identity if and only if $\boldsymbol{\beta}^{s*} = \boldsymbol{\beta}_\star$. $\qquad\square$

**Proposition 3.** *Consider the target model in (6), assume that $\boldsymbol{\Sigma}_t$ is diagonal, and recall the definitions of $\zeta_i$ and $\Omega$. Then, the following results hold:*

1. *If the mask operation $\mathcal{M}$ selects all the features that satisfy $1 - \zeta_i^2 > \Omega$, then the surrogate-to-target model outperforms the standard target model in the asymptotic risk in (10).*

2. *Let $\boldsymbol{M}$ represent the set of all possible $\mathcal{M}$, where $|\boldsymbol{M}| = 2^p$. The optimal $\mathcal{M}^*$ for the asymptotic risk in (10) within $\boldsymbol{M}$ is the one that selects all features satisfying $1 - \zeta_i^2 > \Omega$.*

*Proof.* The proof for Proposition 3 was correct, yet we made changes to it to improve clarity by explicitly stating all intermediate steps. For the purposes of analysis, we assume, without loss of generality, that the first $p_s$ dimensions are selected from $\boldsymbol{\beta}_\star$ in $\mathcal{M}(\boldsymbol{\beta}_\star) = \boldsymbol{\beta}^s \in \mathbb{R}^{p_s}$. Based on this, we no longer need to have the decreasing order for the corresponding $\lambda_i$'s. Let $\bar{\mathcal{R}}(\boldsymbol{\beta}^t)$ and $\bar{\mathcal{R}}(\boldsymbol{\beta}_\star)$ represent the asymptotic risk of the target and the surrogate-to-target models, respectively. From the excess test risk formula in Definition 1, we have that

$$\bar{\mathcal{R}}(\boldsymbol{\beta}^t) = \mathbb{E}\left[\left(y - \boldsymbol{x}^\top \boldsymbol{\beta}^t\right)^2\right] - \sigma_t^2 = \frac{\mathcal{B}(\boldsymbol{\beta}_\star) + \sigma_t^2 \Omega}{1 - \Omega}, \tag{17}$$

where $\mathcal{B}(\boldsymbol{\beta}_\star) = \sum_{i=1}^p \lambda_i \zeta_i^2 \beta_i^2$. Now, consider the zero-padded vector $\bar{\boldsymbol{\beta}}^s = \begin{bmatrix} \boldsymbol{\beta}^s \\ \boldsymbol{0}_{p-p_s} \end{bmatrix} \in \mathbb{R}^p$. This way, we consider the labels in the second training phase as $y^s = \boldsymbol{x}^\top \bar{\boldsymbol{\beta}}^s + z$, where $z \sim \mathcal{N}(0, \sigma_t^2)$. Next, using the asymptotic risk

estimate in Equation (10), we write the excess test risk formula for the surrogate-to-target model with respect to the original ground truth labels:

$$\bar{\mathcal{R}}^{s2t}_{\kappa_t,\sigma_t}(\Sigma_t,\beta_\star,\bar{\beta}^s) := (\bar{\beta}^s - \beta_\star)^\top \theta_1^\top \Sigma_t \theta_1 (\bar{\beta}^s - \beta_\star) + \gamma_t^2(\bar{\beta}^s)\, \mathbb{E}_{g_t}[\theta_2^\top \Sigma_t \theta_2]$$
$$+ \beta_\star^\top (I - \theta_1)^\top \Sigma_t (I - \theta_1)\beta_\star - 2\beta_\star^\top (I - \theta_1)^\top \Sigma_t \theta_1 (\bar{\beta}^s - \beta_\star),$$

where $\theta_1 := (\Sigma_t + \tau_t I)^{-1} \Sigma_t$, $\theta_2 := (\Sigma_t + \tau_t I)^{-1} \Sigma_t^{1/2} \frac{g_t}{\sqrt{p}}$, and $g_t \sim \mathcal{N}(0, I_p)$. Algebraic manipulations give:

$$\bar{\mathcal{R}}(\beta^{s2t}) = \mathbb{E}\left[\left(y - x^\top \beta^{s2t}\right)^2\right] - \sigma_t^2$$

$$\approx \bar{\mathcal{R}}^{s2t}_{\kappa_t,\sigma_t}(\Sigma_t,\beta_\star,\bar{\beta}^s)$$

$$= (\bar{\beta}^s - \beta_\star)^\top \theta_1^\top \Sigma_t \theta_1 (\bar{\beta}^s - \beta_\star) + \kappa_t \frac{\sigma_t^2 + \tau_t^2 (\bar{\beta}^s)^\top \Sigma_t (\Sigma_t + \tau_t I)^{-2} \bar{\beta}^s}{1 - \Omega} \frac{\mathrm{tr}\left(\Sigma_t^2 (\Sigma_t + \tau_t I)^{-2}\right)}{p}$$

$$+ \beta_\star^\top (I - \theta_1)^\top \Sigma_t (I - \theta_1)\beta_\star - 2\beta_\star^\top (I - \theta_1)^\top \Sigma_t \theta_1 (\bar{\beta}^s - \beta_\star)$$

$$= \sum_{i=p_s+1}^{p} \lambda_i \beta_i^2 \left(\frac{\lambda_i}{\lambda_i + \tau_t}\right)^2 + \Omega \frac{\sigma_t^2 + \sum_{i=1}^{p_s} \lambda_i \beta_i^2 \left(\frac{\tau_t}{\lambda_i + \tau_t}\right)^2}{1 - \Omega}$$

$$+ \sum_{i=1}^{p} \lambda_i \beta_i^2 \left(\frac{\tau_t}{\lambda_i + \tau_t}\right)^2 + \sum_{i=p_s+1}^{p} \lambda_i \beta_i^2 \frac{2\tau_t \lambda_i}{(\lambda_i + \tau_t)^2}$$

$$= \frac{\sigma_t^2 \Omega + \sum_{i=1}^{p_s} \lambda_i \beta_i^2 \zeta_i^2}{1 - \Omega} + \sum_{i=p_s+1}^{p} \lambda_i \beta_i^2$$

$$= \frac{\mathcal{B}(\bar{\beta}^s) + \sigma_t^2 \Omega}{1 - \Omega} + \sum_{i=p_s+1}^{p} \lambda_i \beta_i^2, \tag{18}$$

Thus, the risk difference between the target and surrogate-to-target models is

$$\bar{\mathcal{R}}(\beta^t) - \bar{\mathcal{R}}(\beta^{s2t}) = \frac{\mathcal{B}(\beta_\star) - \mathcal{B}(\bar{\beta}^s)}{1 - \Omega} - \sum_{i=p_s+1}^{p} \lambda_i \beta_i^2$$

$$= \frac{\sum_{i=p_s+1}^{p} \lambda_i \zeta_i^2 \beta_i^2}{1 - \Omega} - \sum_{i=p_s+1}^{p} \lambda_i \beta_i^2.$$

We observe that each dimension's contribution to the excess test risk can be analyzed individually. Therefore, if

$$\zeta_i^2 > 1 - \Omega, \tag{19}$$

excluding feature $i$ in the feature selection reduces the overall risk $\bar{\mathcal{R}}(\beta^{s2t})$. Along the same lines, the projection $\mathcal{M}$ that selects all the features $i$ that satisfy $\zeta_i^2 < 1 - \Omega$ minimizes the asymptotic excess test risk. $\qquad\square$

**Proposition 2.** *The optimal surrogate parameter $\beta^s$ that minimizes the asymptotic risk in the under-parametrized region ($n > p$) is equivalent to the ground truth parameter $\beta_\star$. In other words, for any $\beta^s$, the surrogate-to-target model cannot outperform the standard target model in the asymptotic risk.*

*Proof.* According to Theorem 3 of Chang et al. (2021), when the second stage is in the under-parameterized regime, the estimator $\beta^{s2t}$ can be expressed asymptotically as:

$$\beta^{s2t} = \beta^s + \sigma_t \frac{\Sigma_t^{-1/2} g_t}{\sqrt{p\left(\kappa_t^{-1} - 1\right)}},$$

where $\boldsymbol{g}_t \sim \mathcal{N}(\mathbf{0}, \boldsymbol{I}_p)$. Then, the excess test risk estimate of this estimate is:

$$
\mathbb{E}_{(x,y)\sim\mathcal{D}_t(\boldsymbol{\beta}_\star),\boldsymbol{g}_t}[(y - \boldsymbol{x}^\top\boldsymbol{\beta}^{s2t})^2] - \sigma_t^2 = \mathbb{E}_{\boldsymbol{g}_t}\left[\|\boldsymbol{\Sigma}_t^{1/2}(\boldsymbol{\beta}^{s2t} - \boldsymbol{\beta}_\star)\|_2^2\right]
$$

$$
= \mathbb{E}_{\boldsymbol{g}_t}\left[\left(\boldsymbol{\beta}^s + \sigma_t \frac{\boldsymbol{\Sigma}_t^{-1/2}\boldsymbol{g}_t}{\sqrt{p\left(\kappa_t^{-1} - 1\right)}} - \boldsymbol{\beta}_\star\right)^\top \boldsymbol{\Sigma}_t\left(\boldsymbol{\beta}^s + \sigma_t \frac{\boldsymbol{\Sigma}_t^{-1/2}\boldsymbol{g}_t}{\sqrt{p\left(\kappa_t^{-1} - 1\right)}} - \boldsymbol{\beta}_\star\right)\right]
$$

$$
= (\boldsymbol{\beta}^s - \boldsymbol{\beta}_\star)^\top \boldsymbol{\Sigma}_t (\boldsymbol{\beta}^s - \boldsymbol{\beta}_\star) + \frac{\sigma_t^2}{\kappa_t^{-1} - 1}
$$

As $\boldsymbol{\Sigma}_t$ is positive semi-definite, the expression $(\boldsymbol{\beta}^s - \boldsymbol{\beta}_\star)^\top \boldsymbol{\Sigma}_t (\boldsymbol{\beta}^s - \boldsymbol{\beta}_\star)$ is non-negative and takes its minimum value of 0 at $\boldsymbol{\beta}^s = \boldsymbol{\beta}_\star$. Note that at $\boldsymbol{\beta}^s = \boldsymbol{\beta}_\star$ the expression corresponds to the risk of the standard target model. Hence, we conclude that it is not possible to surpass the performance of the standard target model with the surrogate-to-target model when the target model is under-parameterized ($n > p$). □

## A.1 ANALYSIS FOR MODEL SHIFT UNDER RIDGE REGRESSION

In this subsection, we analyze the behavior of the surrogate-to-target model under ridge regression. First, we redefine the surrogate and target models below.

**Stage 1: Surrogate model.** We consider a data distribution $(\tilde{\boldsymbol{x}}, \tilde{y}) \sim \mathcal{D}_s$ following the linear model $\tilde{y} = \tilde{\boldsymbol{x}}^\top\boldsymbol{\beta}_\star + \tilde{z}$, where $\boldsymbol{\beta}_\star \in \mathbb{R}^p$, $\tilde{\boldsymbol{x}} \sim \mathcal{N}(\mathbf{0}, \boldsymbol{\Sigma}_s)$ and $\tilde{z} \sim \mathcal{N}(0, \sigma_s^2)$ is independent of $\tilde{\boldsymbol{x}}$. Let $\{(\tilde{\boldsymbol{x}}_i, \tilde{y}_i)_{i=1}^m\}$ be the dataset for the surrogate model drawn i.i.d. from $\mathcal{D}_s$. The estimator of the surrogate model can be written as follows:

$$
\boldsymbol{\beta}_r^s := \arg\min_{\boldsymbol{\beta}\in\mathbb{R}^p}\left\{\frac{1}{2p}\|\tilde{\boldsymbol{y}} - \tilde{X}\boldsymbol{\beta}\|_2^2 + \frac{\lambda_s}{2}\|\boldsymbol{\beta}\|_2^2\right\} = \frac{1}{p}\left(\frac{1}{p}\tilde{X}^\top\tilde{X} + \lambda_s \boldsymbol{I}\right)^{-1}\tilde{X}^\top\tilde{\boldsymbol{y}}, \tag{20}
$$

where $\tilde{X} = [\tilde{\boldsymbol{x}}_1^\top, \ldots, \tilde{\boldsymbol{x}}_m^\top]^\top \in \mathbb{R}^{m\times p}$ and $\tilde{\boldsymbol{y}} = [y_1, \ldots, y_m]^\top \in \mathbb{R}^m$.

**Stage 2: Target model.** Given $\boldsymbol{\beta}_r^s \in \mathbb{R}^p$, we consider another data distribution $(\boldsymbol{x}, y^s) \sim \mathcal{D}_t(\boldsymbol{\beta}^s)$ following the linear model $y^s = \boldsymbol{x}^\top\boldsymbol{\beta}^s + z$, where $\boldsymbol{x} \sim \mathcal{N}(\mathbf{0}, \boldsymbol{\Sigma}_t)$ and $z \sim \mathcal{N}(0, \sigma_t^2)$. Let $\{(\boldsymbol{x}_i, y_i^s)_{i=1}^n\}$ be the dataset for the target model drawn i.i.d. from $\mathcal{D}_t(\boldsymbol{\beta}^s)$. As for the surrogate model, the estimator for the target model is defined as

$$
\boldsymbol{\beta}_r^{s2t} := \arg\min_{\boldsymbol{\beta}\in\mathbb{R}^p}\left\{\frac{1}{2p}\|\boldsymbol{y}^s - X\boldsymbol{\beta}\|_2^2 + \frac{\lambda_t}{2}\|\boldsymbol{\beta}\|_2^2\right\} = \frac{1}{p}\left(\frac{1}{p}\tilde{X}^\top\tilde{X} + \lambda_t \boldsymbol{I}\right)^{-1}\tilde{X}^\top\boldsymbol{y}^s, \tag{21}
$$

where $X = [\boldsymbol{x}_1^\top, \ldots, \boldsymbol{x}_n^\top]^\top \in \mathbb{R}^{n\times p}$ and $\boldsymbol{y}^s = [y_1^s, \ldots, y_n^s]^\top \in \mathbb{R}^n$. Our analysis will generally apply to an arbitrary $\boldsymbol{\beta}_r^s$ choice and will not require it to be the outcome of (20). Finally, we define the excess (population) risk for a given estimator $\hat{\boldsymbol{\beta}} \in \mathbb{R}^p$ as defined in (5). Throughout the section, we compare the surrogate-to-target model with the following reference model.

**Reference Model: Standard target model.** We study the generalization performance of $\boldsymbol{\beta}_r^{s2t}$ with respect to the standard target model, which has access to the ground-truth parameter through labeling. Specifically, consider the dataset $\{(\boldsymbol{x}_i, y_i)_{i=1}^n\}$ drawn i.i.d. from $\mathcal{D}_t(\boldsymbol{\beta}_\star)$; then, the estimation is

$$
\boldsymbol{\beta}_r^t := \arg\min_{\boldsymbol{\beta}\in\mathbb{R}^p}\left\{\frac{1}{2p}\|\boldsymbol{y} - X\boldsymbol{\beta}\|_2^2 + \frac{\lambda_t}{2}\|\boldsymbol{\beta}\|_2^2\right\} = \frac{1}{p}\left(\frac{1}{p}\tilde{X}^\top\tilde{X} + \lambda_t \boldsymbol{I}\right)^{-1}\tilde{X}^\top\boldsymbol{y}, \tag{22}
$$

where $X = [\boldsymbol{x}_1^\top, \ldots, \boldsymbol{x}_n^\top]^\top \in \mathbb{R}^{n\times p}$ and $\boldsymbol{y} = [y_1, \ldots, y_n]^\top \in \mathbb{R}^n$. We compare the excess risks of the surrogate-to-target model $\mathcal{R}(\boldsymbol{\beta}_r^{s2t})$ with that of the standard target model $\mathcal{R}(\boldsymbol{\beta}_r^t)$.

Now, we will obtain a result similar to Theorem 1 under ridge regression. For this purpose, we have the following definition:

**Definition 4.** *Let $\kappa_t = p/n > 1$ and $\tau_{t,r} \in \mathbb{R}$ be the unique solution of the following equation*

$$
\kappa_t^{-1} - \frac{\lambda_t}{\tau_{t,r}} = \frac{1}{p}\mathrm{tr}\left((\boldsymbol{\Sigma}_t + \tau_{t,r}\boldsymbol{I})^{-1}\boldsymbol{\Sigma}_t\right). \tag{23}
$$

*Then, define $X_{\kappa_t,\sigma_t^2}^t(\boldsymbol{\Sigma}_t, \boldsymbol{\beta}^s, \boldsymbol{g}_t)$, the function $\gamma_t$, and the asymptotic risk $\bar{\mathcal{R}}_{\kappa_t,\sigma_t}^{s2t}(\boldsymbol{\Sigma}_t, \boldsymbol{\beta}_\star, \boldsymbol{\beta}^s)$ as in Definition 1 based on $\tau_{t,r}$.*

**Theorem 3.** *Suppose that, for some constant $M_t > 1$, we have $1/M_t \leq \kappa_t, \sigma_t^2 \leq M_t$ and $\|\Sigma_t\|_{op}, \|\Sigma_t^{-1}\|_{op} \leq M_t$. Then, there exists a constant $C = C(M_t)$ such that, for any $\varepsilon \in (0, 1/2]$, the following holds with $R + 1 < M_t$:*

$$\sup_{\boldsymbol{\beta}_\star, \boldsymbol{\beta}^s \in \boldsymbol{B}_p(R)} \mathbb{P}(\sup_{\lambda_t \in [0, M_t]} \left| \mathcal{R}(\boldsymbol{\beta}^{s2t}) - \bar{\mathcal{R}}_{\kappa_t, \sigma_t}^{s2t}(\Sigma_t, \boldsymbol{\beta}_\star, \boldsymbol{\beta}^s) \right| \geq \varepsilon) \leq Cp e^{-p\varepsilon^4/C}. \tag{24}$$

*Proof.* The proof directly follows from the proof of Theorem 1 with only one modification in (14):

$$\sup_{\boldsymbol{\beta}^s \in \boldsymbol{B}(\frac{M_t+R}{2})} \mathbb{P}\left( \sup_{\lambda_t \in [0, M_t]} \left| f(\boldsymbol{\beta}^{s2t}) - \mathbb{E}_{\boldsymbol{g}_t}[f(X_{\kappa_t, \sigma_t^2}^t(\Sigma_t, \boldsymbol{\beta}^s, \boldsymbol{g}_t))] \right| \geq \varepsilon \right) \leq Cp e^{-p\varepsilon^4/C}. \tag{25}$$

□

In Propositions 1, 3 and Corollary 1, we treat $\tau_t$ as a variable, which is a fixed point equation based on the covariance matrix and ratio between numbers of features and samples under ridgeless regression. However, under ridge regression, we modify the parameter $\tau_{t,r}$ and define the asymptotic risk the same as we define under ridgeless regression based on this parameter. This means that we can directly apply the results of Propositions 1, 3 and Corollary 1 to ridge regression by only modifying the parameter $\tau$. Similarly, Theorem 2 can also be extended to ridge regression.

## A.2 Experimental Details

In the CIFAR-10 experiment, we initially trained the surrogate models on the training portion of the CIFAR-10 dataset. While training the surrogate-to-target models, we employed the predictions from the surrogate models. Conversely, when training the standard target model, we utilized the ground truth labels. During testing, all models were evaluated using the test portion of the CIFAR-10 dataset. It is important to note that the distributions for both surrogate and target are identical since we trained and tested the models on the same training and testing sets.

We employ three distinct surrogate model sizes: big, medium, and small. The big model contains 127,094 parameters, the medium model 58,342 parameters, and the small model 28,286 parameters. All three models are shallow, three-layer convolutional networks that follow the same architectural specifications. For the small model, $x = 4$; for the medium model, $x = 8$; and for the big model, $x = 16$.

| Layer Type | Input Channels | Output Channels | Kernel/Stride/Pad |
|---|---|---|---|
| Conv2d | 3 | $x$ | $3 \times 3/1/1$ |
| ReLU | − | − | − |
| MaxPool2d | − | − | $2 \times 2/2$ |
| Conv2d | $x$ | $2x$ | $3 \times 3/1/1$ |
| ReLU | − | − | − |
| MaxPool2d | − | − | $2 \times 2/2/0$ |
| Conv2d | $2x$ | $4x$ | $3 \times 3/1/1$ |
| ReLU | − | − | − |
| MaxPool2d | − | − | $2 \times 2/2/0$ |
| Flatten | − | − | − |
| Linear | $64x$ | 100 | − |
| ReLU | − | − | − |
| Linear | 100 | 10 | − |

We initialize the optimizer for our model using stochastic gradient descent (SGD) provided by the optim module of PyTorch. The optimizer is configured with the following parameters: learning rate set to 0.01, momentum to 0.9, and weight decay to $5 \times 10^{-4}$. Additionally, we define a learning rate scheduler, specifically a cosine annealing scheduler, which adjusts the learning rate using a cosine function over 200 iterations, denoted T_max. We use a batch size of 32 and trained all models over 60 epochs.

## B   Proofs for Section 4

**Definition 2** (Omniscient test risk estimate). *Fix $p > n \geq 1$. Given a covariance $\Sigma = U \operatorname{diag}(\lambda)U^{\top}$, $\beta_{\star}$, and the noise term $\sigma$, set $\bar{\beta} = U^{\top}\beta_{\star}$ and define $\tau \in \mathbb{R}$ as the unique non-negative solution of $n = \sum_{i=1}^{p} \frac{\lambda_i}{\lambda_i + \tau}$. Then, the omniscient excess test risk estimate is the following:*

$$\mathcal{R}_{om}(\hat{\beta}) \approx \mathbb{E}_{\hat{\beta} \sim D(\beta_{\star})}\left[(y - x^{\top}\hat{\beta})^2\right] - \sigma^2 = \frac{\sigma^2 \Omega + \mathcal{B}(\bar{\beta})}{1 - \Omega}, \tag{12}$$

*where* $\quad \zeta_i = \frac{\tau}{\lambda_i + \tau}, \quad \Omega = \frac{1}{n}\sum_{i=1}^{p}(1 - \zeta_i)^2, \quad \mathcal{B}(\bar{\beta}) = \sum_{i=1}^{p} \lambda_i \zeta_i^2 \bar{\beta}_i^2.$

In the following proof, we suppose that the empirical distributions of $\bar{\beta}$ and $\lambda$ converge as $p \to \infty$ having fixed the ratio $p/n = \kappa$. Then, we will prove that the omniscient risk converges to the asymptotic risk defined in (10).

*Proof for the proportional asymptotic case.* Using Theorem 2.3 of Han & Xu (2023), we can estimate $\hat{\beta}$ as follows:

$$\hat{\beta} = (\Sigma + \tau I)^{-1}\Sigma\left(\beta_{\star} + \frac{\Sigma^{-1/2}\gamma_s(\beta_{\star})g}{\sqrt{p}}\right),$$

where

$$g \sim \mathcal{N}(0, I_p), \quad \gamma_s^2(\beta_{\star}) = \kappa \frac{\sigma + \tau^2 \|(\Sigma + \tau I)^{-1}\Sigma^{1/2}\beta_{\star}\|_2^2}{1 - \frac{1}{n}\operatorname{tr}\left((\Sigma + \tau I)^{-2}\Sigma^2\right)}, \quad \tau \text{ is the solution to } n = \sum_{i=1}^{p} \frac{\lambda_i}{\lambda_i + \tau}.$$

Let

$$X_1 = (\Sigma + \tau I)^{-1}\Sigma \quad , \quad X_2 = \frac{(\Sigma + \tau I)^{-1}\Sigma^{1/2}\gamma_s(\beta_{\star})}{\sqrt{p}}.$$

Using this estimate, we can calculate the excess test risk as

$$\begin{aligned}
\mathcal{R}_{om}(\hat{\beta}) &= \mathbb{E}\left[((X_1 - I)\beta_{\star} + X_2 g)^{\top}\Sigma((X_1 - I)\beta_{\star} + X_2 g)\right] \\
&= \beta_{\star}^{\top}(X_1 - I)^{\top}\Sigma(X_1 - I)\beta_{\star} + \mathbb{E}\left[g^{\top}X_2^{\top}\Sigma X_2 g\right] \\
&= \beta_{\star}^{\top}(X_1 - I)^{\top}\Sigma(X_1 - I)\beta_{\star} + \operatorname{tr}\left(X_2^{\top}\Sigma X_2\right).
\end{aligned} \tag{26}$$

Then by recalling the eigendecomposition for the covariance matrix $\Sigma = U\Lambda U^{\top}$, we have

$$\begin{aligned}
X_1 &= (U\Lambda U^{\top} + \tau U U^{\top})^{-1}U\Lambda U^{\top} \\
&= U(\Lambda + \tau I)^{-1}U^{\top}U\Lambda U^{\top} \\
&= U \operatorname{diag}\left(\frac{\lambda}{\lambda + \tau}\right)U^{\top}.
\end{aligned}$$

Using the diagonalization of $I$, $X_1 - I$ can now be computed as

$$X_1 - I = U \operatorname{diag}\left(\frac{-\tau}{\lambda + \tau}\right)U^{\top}.$$

Let's now compute

$$\begin{aligned}
\beta_{\star}^{\top}(X_1 - I)^{\top}\Sigma(X_1 - I)\beta_{\star} &= \beta_{\star}^{\top}U \operatorname{diag}\left(\frac{-\tau}{\lambda + \tau}\right)U^{\top}U\Lambda U^{\top}U \operatorname{diag}\left(\frac{-\tau}{\lambda + \tau}\right)U^{\top}\beta_{\star} \\
&= \beta_{\star}^{\top}U \operatorname{diag}\left(\frac{\lambda\tau^2}{(\lambda + \tau)^2}\right)U^{\top}\beta_{\star}.
\end{aligned}$$

As $\bar{\beta} = U^{\top}\beta_{\star}$, we obtain that the RHS of the previous expression equals

$$\sum_{i=1}^{p} \frac{\lambda_i \tau^2 \bar{\beta}_i^2}{(\lambda_i + \tau)^2} = \mathcal{B}(\bar{\beta}).$$

Next, we write more compactly the terms $\mathrm{tr}\left(X_2^\top \Sigma X_2\right)$ and $\gamma_s^2(\boldsymbol{\beta}_\star)$. By defining the short-hand notation $\Omega = \frac{1}{n}\mathrm{tr}\left((\Sigma + \tau I)^{-2}\Sigma^2\right) = \frac{1}{n}\sum_{i=1}^p (1 - \zeta_i)^2$, we have

$$\mathrm{tr}\left(X_2^\top \Sigma X_2\right) = \frac{\gamma_s^2(\boldsymbol{\beta}_\star)}{p}\sum_{i=1}^p\left(\frac{\lambda_i}{\lambda_i + \tau}\right)^2 = \frac{\gamma_s^2(\boldsymbol{\beta}_\star)n\Omega}{p},$$

$$\gamma_s^2(\boldsymbol{\beta}_\star) = \kappa\frac{\sigma^2 + \tau^2\|(\Sigma + \tau I)^{-1}\Sigma^{1/2}\boldsymbol{\beta}_\star\|_2^2}{1 - \Omega} = \kappa\frac{\sigma^2 + \sum_{i=1}^p\frac{\lambda_i\tau^2\bar{\beta}_i^2}{(\lambda_i+\tau)^2}}{1 - \Omega} = \kappa\frac{\sigma^2 + \mathcal{B}(\bar{\boldsymbol{\beta}})}{1 - \Omega},$$

where $\kappa = \frac{p}{n}$. Hence, putting it all together in (26) gives the desired result. $\square$

**Proposition 4** (Asymptotic analysis of $\tau_t$ and $\Omega$). *Let the covariance matrix $\Sigma \in \mathbb{R}^{p\times p}$ be diagonal and $\Sigma_{i,i} = \lambda_i = i^{-\alpha}$ for $1 < \alpha$. Recall from Definition 2 that, as $p \to \infty$, $\tau_t$ and $\Omega$ are given by the equations*

$$\sum_{i=1}^\infty \frac{\lambda_i}{\lambda_i + \tau_t} = n, \qquad n\Omega = \sum_{i=1}^\infty\left(\frac{i^{-\alpha}}{i^{-\alpha} + \tau_t}\right)^2.$$

*Then, the following results hold*

$$\tau_t = cn^{-\alpha}\left(1 + O(n^{-1})\right), \qquad for\ c = \left(\frac{\pi}{\alpha\sin(\pi/\alpha)}\right)^\alpha,$$

$$\Omega = \frac{\alpha - 1}{\alpha} - O(n^{-1}). \tag{13}$$

*Proof.* We start with the asymptotic analysis of $\tau_t$. Along the same lines as Simon et al. (2024), since $\frac{i^{-\alpha}}{i^{-\alpha}+\tau_t}$ is a monotonically decreasing function, we have:

$$n = \sum_{i=1}^\infty \frac{i^{-\alpha}}{i^{-\alpha} + \tau_t} \le \int_0^\infty \frac{x^{-\alpha}}{x^{-\alpha} + \tau_t}\,dx = \frac{\pi}{\alpha\sin(\pi/\alpha)}\tau_t^{-1/\alpha}.$$

Furthermore,

$$\frac{\pi}{\alpha\sin(\pi/\alpha)}\tau_t^{-1/\alpha} - 1 = \int_0^\infty \frac{x^{-\alpha}}{x^{-\alpha} + \tau_t}\,dx - 1 \le \int_1^\infty \frac{x^{-\alpha}}{x^{-\alpha} + \tau_t}\,dx \le \sum_{i=1}^\infty \frac{i^{-\alpha}}{i^{-\alpha} + \tau_t} = n.$$

Hence, combining these two facts gives

$$\frac{\pi}{\alpha\sin(\pi/\alpha)}\tau_t^{-1/\alpha} - 1 \le n \le \frac{\pi}{\alpha\sin(\pi/\alpha)}\tau_t^{-1/\alpha}$$

$$\iff \left(\frac{(n+1)\alpha\sin(\pi/\alpha)}{\pi}\right)^{-\alpha} \le \tau_t \le \left(\frac{n\alpha\sin(\pi/\alpha)}{\pi}\right)^{-\alpha},$$

which leads to the desired result.

Next, we move to the asymptotic analysis of $\Omega$. We have that

$$n\Omega = \sum_{i=1}^\infty\left(\frac{i^{-\alpha}}{i^{-\alpha} + \tau_t}\right)^2 \le \int_0^\infty\left(\frac{x^{-\alpha}}{x^{-\alpha} + \tau_t}\right)^2\,dx = \frac{\pi(\alpha-1)}{\alpha^2\sin(\pi/\alpha)}\tau_t^{-1/\alpha}.$$

Besides, since the summand is monotonically decreasing, we also have

$$\frac{\pi(\alpha-1)}{\alpha^2\sin(\pi/\alpha)}\tau_t^{-1/\alpha} - 1 \le \int_0^\infty\left(\frac{x^{-\alpha}}{x^{-\alpha} + \tau_t}\right)^2\,dx - 1 \le \int_1^\infty\left(\frac{x^{-\alpha}}{x^{-\alpha} + \tau_t}\right)^2\,dx \le \sum_{i=1}^\infty\left(\frac{i^{-\alpha}}{i^{-\alpha} + \tau_t}\right)^2 = n\Omega.$$

Hence,

$$\frac{\pi(\alpha-1)}{\alpha^2\sin(\pi/\alpha)}\tau_t^{-1/\alpha} - 1 \le n\Omega \le \frac{\pi(\alpha-1)}{\alpha^2\sin(\pi/\alpha)}\tau_t^{-1/\alpha}. \tag{27}$$

By the hypothesis on $\tau_t$, we have that

$$\tau_t^{-1/\alpha} = n\frac{\alpha\sin(\pi/\alpha)}{\pi}\left(1 - O\left(n^{-1}\right)\right), \tag{28}$$

and plugging this in (27) gives the desired result. $\square$

**Proposition 5.** *Set the constants $C_1 := \frac{\alpha \sin(\pi/\alpha)}{\pi(\alpha - 1)^{1/\alpha}}$ and $C_2 := \frac{\alpha \sin(\pi/\alpha)}{\pi(\sqrt{\alpha} - 1)^{1/\alpha}}$ and assume the power-law eigenstructure $\Sigma_{i,i} = \lambda_i = i^{-\alpha}$ for $1 < \alpha$. Let $\tau_t$ and $\Omega$ be the solutions given by Proposition 4 and define $\zeta_i = \frac{\tau_t}{\lambda_i + \tau_t}$. Then, the indices $i$ for which $\zeta_i < 1 - \Omega$ are $i < nC_1 + O(1)$; while the indices $i$ for which is $\zeta_i^2 < 1 - \Omega$ are $i < nC_2 + O(1)$.*

*Proof.* Recall from Proposition 3 that we should identify indices $i$ which satisfy the condition $\zeta_i^2 > 1 - \Omega$ to decide if we're better off not selecting this dimension $i$ in the surrogate model. Furthermore, Proposition 4 gives that $\Omega = \frac{\alpha - 1}{\alpha} - O(n^{-1})$. Putting these together, we have

$$\zeta_i^2 > 1 - \Omega$$

$$\iff \zeta_i^2 > c' \quad \text{where } c' = \frac{1}{\alpha} + O(n^{-1})$$

$$\iff \frac{\tau_t^2}{(\tau_t + i^{-\alpha})^2} = \frac{\tau_t^2 i^{2\alpha}}{(\tau_t i^\alpha + 1)^2} > c'$$

$$\iff (1 - c')\tau_t^2 i^{2\alpha} > 2c'\tau_t i^\alpha + c'$$

$$\iff \left( \sqrt{1 - c'}\tau_t i^\alpha - \frac{c'}{\sqrt{1 - c'}} \right)^2 > \frac{c'}{1 - c'}$$

$$\iff i^\alpha > \frac{\sqrt{c'}}{\tau_t(1 - \sqrt{c'})}$$

$$\iff i > \tau_t^{-1/\alpha} \left( \frac{\sqrt{c'}}{1 - \sqrt{c'}} \right)^{1/\alpha}$$

As $c' = \frac{1}{\alpha} + O(n^{-1})$, we get $\left( \frac{\sqrt{c'}}{1 - \sqrt{c'}} \right)^{1/\alpha} = \frac{1}{(\sqrt{\alpha} - 1)^{1/\alpha}}(1 + O(n^{-1}))$. Incorporating (28), we achieve that

$$\tau_t^{-1/\alpha} \left( \frac{\sqrt{c'}}{1 - \sqrt{c'}} \right)^{1/\alpha} = n \frac{\alpha \sin(\pi/\alpha)}{\pi(\sqrt{\alpha} - 1)^{1/\alpha}} \left( 1 + O(n^{-1}) \right) = nC_2 + O(1).$$

Similarly, by following the same procedure with the initial inequality $\zeta_i > 1 - \Omega$, we get

$$\zeta_i > 1 - \Omega \iff i > nC_1 + O(1), \quad \text{where} \quad C_1 = \frac{\alpha \sin(\pi/\alpha)}{\pi(\alpha - 1)^{1/\alpha}}.$$

$\square$

In Figure 3, we compare the empirical results with theoretical predictions for the number of features that meet the selection criteria in the optimal mask $\mathcal{M}^*$ ($\zeta_i^2 < 1 - \Omega$). The theoretical value, calculated as $n \frac{\alpha \sin(\pi/\alpha)}{\pi(\sqrt{\alpha} - 1)^{1/\alpha}}$ ignoring the $O(1)$ term, aligns well with the experimental data and the accuracy in estimation increases with $\alpha$. Notably, our theoretical estimate also closely matches the empirical results when the sample size $n$ is small relative to the feature size $p$.

**Proposition 7** (Scaling law for masked surrogate-to-target model). *Together with the eigenvalues, also assume now power-law form for $\lambda_i \beta_i^2$, that is $\lambda_i \beta_i^2 = i^{-\beta}$ for $\beta > 1$. Then, in the limit of $p \to \infty$, the excess test risk for the masked surrogate-to-target model with the optimal dimensionality has the same scaling law as the reference (target) model:*

$$\mathcal{R}_{om}(\boldsymbol{\beta}^{s2t}) = \Theta(n^{-(\beta-1)}) \quad \text{if } \beta < 2\alpha + 1,$$

*and*

$$\mathcal{R}_{om}(\boldsymbol{\beta}^{s2t}) = \Theta(n^{-2\alpha}) \quad \text{if } \beta > 2\alpha + 1.$$

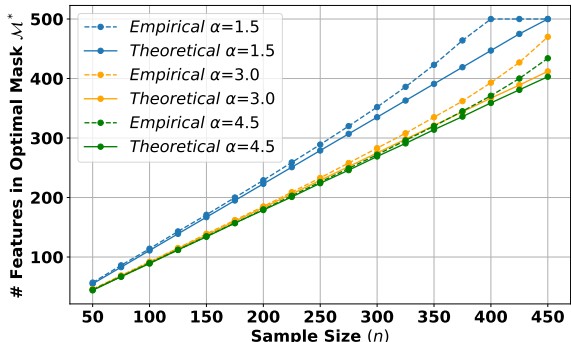

Figure 3: Comparison of the empirical and theoretical number of features satisfying the feature selection condition in the optimal mask $\mathcal{M}^*$ ($\zeta_i^2 < 1 - \Omega$). The theoretical value is calculated as $n\dfrac{\alpha \sin(\pi/\alpha)}{\pi(\sqrt{\alpha} - 1)^{1/\alpha}}$, ignoring the $O(1)$ in Proposition 5. **Setting:** The feature size is $p = 500$, and the feature covariance follows the power-law structure $\lambda_i = i^{-\alpha}$ for $\alpha = 1.5, 3.0,$ and $4.5$.

*Proof.* As discussed in Section 4, in order to analyze the model's inherent error, we need to set $\sigma_t^2 = O(n^{-\gamma})$ where $\gamma$ is the exponent characterizing the scaling law of the test risk in the noiseless setting. We will work on this proof in two cases depending on $\beta$ and $2\alpha + 1$.

**Case 1:** $\beta < 2\alpha + 1$. In this case, it is previously stated by Cui et al. (2022); Simon et al. (2024) that the test risk of ridgeless overparameterized linear regression can be described in the scaling sense as $\mathbf{err} = \Theta(n^{-\beta+1})$ when $\beta < 2\alpha + 1$. Consider the optimal mask operation $\mathcal{M}$ mentioned in Proposition 3 that selects all features satisfying $1 - \zeta_i^2 > \Omega$. Let $p_s$ be the number of selected features. We can then decompose the risk estimate in Definition 2 as follows:

$$\frac{\mathcal{B}(\bar{\beta}_\star) + \sigma_t^2 \Omega}{1 - \Omega} = \frac{\sum_{i=1}^{p_s} \lambda_i \zeta_i^2 \beta_i^2 + \sum_{i=p_s+1}^{p} \lambda_i \zeta_i^2 \beta_i^2 + \sigma_t^2 \Omega}{1 - \Omega} = \frac{\mathbf{err1} + \mathbf{err2} + \sigma_t^2 \Omega}{1 - \Omega},$$

where $\mathbf{err1}$ and $\mathbf{err2}$ are the contributions to the total risk of the target model from dimensions selected and omitted in the surrogate model, respectively. Therefore, we express the total error as:

$$\frac{\mathbf{err1} + \mathbf{err2} + \sigma_t^2 \Omega}{1 - \Omega} = \mathbf{err} = \Theta(n^{-\beta+1}).$$

Going back to Proposition 5, we know that, as $p \to \infty$, the criterion for selecting a feature $i$ in the optimal masked surrogate model is given by

$$i < nC_2 + O(1), \quad \text{where} \quad C_2 = \frac{\alpha \sin(\pi/\alpha)}{\pi(\sqrt{\alpha} - 1)^{1/\alpha}}.$$

Define now $\omega_n = nC_2 + O(1)$. The equation (18) tells us that after the optimal mask operation $\mathcal{M}$, $\dfrac{\mathbf{err2}}{1 - \Omega}$ is replaced by $\mathbf{err2}'$, which is calculated as follows

$$\mathbf{err2}' = \sum_{i=\omega_n+1}^{p} \lambda_i \beta_i^2 = \sum_{i=\omega_n+1}^{p} i^{-\beta}.$$

Since $x^{-\beta}$ is a monotonically decreasing function, we can bound the summation by the following two integrals:

$$\int_{\omega_n+1}^{p+1} x^{-\beta} dx \leq \sum_{\omega_n+1}^{p} i^{-\beta} \leq \int_{\omega_n}^{p} x^{-\beta} dx$$

$$\frac{(\omega_n + 1)^{-\beta+1} - (p+1)^{-\beta+1}}{\beta - 1} \leq \sum_{\omega_n+1}^{p} i^{-\beta} \leq \frac{(\omega_n)^{-\beta+1} - p^{-\beta+1}}{\beta - 1}.$$

In the limit of $p \to \infty$, we obtain

$$\mathbf{err2'} = \Theta(n^{-\beta+1}).$$

Thus, we have tightly estimated $\mathbf{err2'}$. Using the fact from Proposition 4 that $\Omega = \Theta(1)$, and our assumption on the noise variance $\sigma_t^2 = O(n^{-\beta+1})$, we conclude that the scaling law doesn't change for the surrogate-to-target model as

$$\mathcal{R}_{om}(\boldsymbol{\beta}^{s2t}) = \frac{\mathbf{err1} + \sigma_t^2 \Omega}{1 - \Omega} + \mathbf{err2'} = \Theta(n^{-\beta+1}).$$

**Case 2:** $\beta > 2\alpha + 1$. In this case, we show that the scaling law is determined by $\mathbf{err1}$, hence changing $\mathbf{err2}$ to $\mathbf{err2'}$ has no effect in the scaling sense. From Proposition 4, we have the asymptotic expression $\tau_t = cn^{-\alpha}\left(1 + O(n^{-1})\right)$, for $c = \left(\dfrac{\pi}{\alpha \sin(\pi/\alpha)}\right)^{\alpha}$. We can argue that there exists positive constants $c_1 < \dfrac{1}{c} < c_2$, such that $c_1 n^\alpha \leq \dfrac{1}{\tau_t} \leq c_2 n^\alpha$. We have that

$$\mathbf{err1} = \sum_{i=1}^{\omega_n} \frac{i^{-\beta}}{(1 + \frac{1}{\tau_t} i^{-\alpha})^2} \leq \sum_{i=1}^{\omega_n} \frac{i^{-\beta}}{(1 + c_1 n^\alpha i^{-\alpha})^2}$$

$$= \sum_{i=1}^{\omega_n} \frac{i^{2\alpha-\beta}}{(i^\alpha + c_1 n^\alpha)^2} \leq \sum_{i=1}^{\omega_n} \frac{i^{2\alpha-\beta}}{c_1^2 n^{2\alpha}}.$$

This implies $\mathbf{err1} = O(n^{-2\alpha})$. At the same time,

$$\mathbf{err1} = \sum_{i=1}^{\omega_n} \frac{i^{-\beta}}{(1 + \frac{1}{\tau_t} i^{-\alpha})^2} \geq \sum_{i=1}^{\omega_n} \frac{i^{2\alpha-\beta}}{(i^\alpha + c_2 n^\alpha)^2}$$

$$\geq \sum_{i=1}^{\omega_n} \frac{i^{2\alpha-\beta}}{((\omega_n)^\alpha + c_2 n^\alpha)^2} = \sum_{i=1}^{\omega_n} \frac{i^{2\alpha-\beta}}{n^{2\alpha}((\omega_n/n)^\alpha + c_2)^2}.$$

Using $\omega_n/n = \Theta(1)$ gives $\mathbf{err1} = \Omega(n^{-2\alpha})$ and we can conclude that $\mathbf{err1} = \Theta(n^{-2\alpha})$. From Cui et al. (2022), we already know that $\mathbf{err} = \Theta(n^{-2\alpha})$ when $\beta > 2\alpha + 1$. Using $\Omega = \Theta(1)$, and our assumption on the noise variance $\sigma_t^2 = O(n^{-2\alpha})$ allows us to conclude that the scaling is dominated by $\mathbf{err1}$, and thus, the scaling law remains unchanged. $\square$

**Proposition 6** (Scaling law). *Let the covariance matrix $\Sigma_t$ be diagonal with eigenvalues $\lambda_i$, and let the ground-truth parameter $\boldsymbol{\beta}_\star$ have components $\beta_i$ corresponding to each feature. Assume that both eigenvalues $\lambda_i$ and signal coefficients $\lambda_i \beta_i^2$ follow a power-law decay, i.e., $\lambda_i \beta_i^2 = i^{-\beta}$ and $\lambda_i = i^{-\alpha}$ for $\alpha, \beta > 1$. Let the optimal surrogate parameter $\boldsymbol{\beta}^{s*}$ be given by Proposition 1 and define the minimum surrogate-to-target risk attained by $\boldsymbol{\beta}^{s*}$ as $\mathcal{R}_{om}^*(\boldsymbol{\beta}^{s2t}) = \min \mathcal{R}_{om}(\boldsymbol{\beta}^{s2t})$, where $\mathcal{R}_{om}(\boldsymbol{\beta}^{s2t})$ is described in Definition 2. Then, in the limit of $p \to \infty$, the excess test risk of the surrogate-to-target model with an optimal surrogate parameter scales the same as that of the standard target model. Specifically, we have*

$$\mathcal{R}_{om}^*(\boldsymbol{\beta}^{s2t}) = \Theta(n^{-(\beta-1)}) = \mathcal{R}_{om}(\boldsymbol{\beta}^t), \qquad \text{if } \beta < 2\alpha + 1,$$
$$\mathcal{R}_{om}^*(\boldsymbol{\beta}^{s2t}) = \Theta(n^{-2\alpha}) = \mathcal{R}_{om}(\boldsymbol{\beta}^t), \qquad \text{if } \beta > 2\alpha + 1.$$

*Proof.* From asymptotic risk decomposition in (34), we can write

$$\mathbb{E}_{\boldsymbol{g}_t}\left[f(X_{\kappa_t, \sigma_t^2}^t(\Sigma_t, \boldsymbol{\beta}^s, \boldsymbol{g}_t))\right] = (\boldsymbol{\beta}^s - \boldsymbol{\beta}_\star)^\top \boldsymbol{\theta}_1^\top \Sigma_t \boldsymbol{\theta}_1 (\boldsymbol{\beta}^s - \boldsymbol{\beta}_\star) + \gamma_t^2(\boldsymbol{\beta}^s) \mathbb{E}_{\boldsymbol{g}_t}[\boldsymbol{\theta}_2^\top \Sigma_t \boldsymbol{\theta}_2]$$

$$+ \boldsymbol{\beta}_\star^\top (I - \boldsymbol{\theta}_1)^\top \Sigma_t (I - \boldsymbol{\theta}_1) \boldsymbol{\beta}_\star - 2\boldsymbol{\beta}_\star^\top (I - \boldsymbol{\theta}_1)^\top \Sigma_t \boldsymbol{\theta}_1 (\boldsymbol{\beta}^s - \boldsymbol{\beta}_\star)$$

$$\geq \gamma_t^2(\boldsymbol{\beta}^s) \mathbb{E}_{\boldsymbol{g}_t}[\boldsymbol{\theta}_2^\top \Sigma_t \boldsymbol{\theta}_2],$$

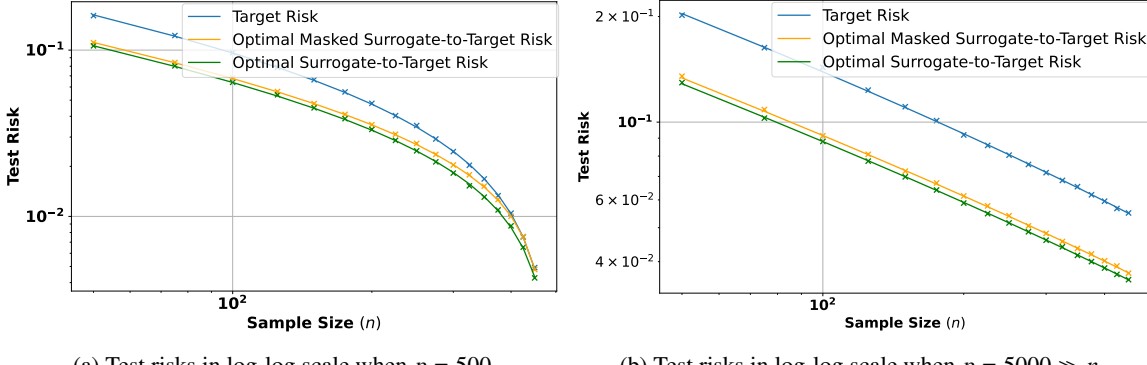

(a) Test risks in log-log scale when $p = 500$      (b) Test risks in log-log scale when $p = 5000 \gg n$

Figure 4: Scaling law behavior of the test risks of optimal surrogate models. **(a):** Associated test risks as a function of sample size in log-log scale. **Setting:** The feature size is $p = 500$; the sample size $n$ changes from 50 to 450 with increments of 50; the feature covariance follows the power-law structure $\lambda_i = i^{-2}$, $\lambda_i \beta_i^2 = i^{-1.5}$ **(b):** Associated test risks as a function of sample size in log-log scale when $p \gg n$. **Setting:** Same as in (a) except that $p = 5000$.

since we can put in the form of $(a - b)^2 + c^2 \geq c^2$. At the same time, we know that

$$\gamma_t^2(\boldsymbol{\beta}^s)\, \mathbb{E}_{g_i}[\boldsymbol{\theta}_2^\top \boldsymbol{\Sigma}_t \boldsymbol{\theta}_2] = \kappa_t \frac{\sigma_t^2 + \tau_t^2 \|(\boldsymbol{\Sigma}_t + \tau_t \boldsymbol{I})^{-1}\boldsymbol{\Sigma}_t^{1/2}\boldsymbol{\beta}_\star\|_2^2}{1 - \frac{1}{n}\operatorname{tr}\left((\boldsymbol{\Sigma}_t + \tau_t \boldsymbol{I})^{-2}\boldsymbol{\Sigma}_t^2\right)} \frac{\operatorname{tr}\left(\boldsymbol{\Sigma}_t^2(\boldsymbol{\Sigma}_t + \tau_t \boldsymbol{I})^{-2}\right)}{p}$$

$$= \kappa_t \frac{\sigma_t^2 + \tau_t^2 \|(\boldsymbol{\Sigma}_t + \tau_t \boldsymbol{I})^{-1}\boldsymbol{\Sigma}_t^{1/2}\boldsymbol{\beta}^s\|_2^2}{1 - \Omega} \frac{n\Omega}{p}$$

$$= \frac{\Omega}{1 - \Omega}\left(\sigma_t^2 + \sum_{i=1}^{p} \lambda_i \beta_i^{s2} \zeta_i^2\right).$$

Recall the optimal surrogate vector discussed in Proposition 1 and the corresponding minimal surrogate-to-target risk $\mathcal{R}_{om}^*(\boldsymbol{\beta}^{s2t})$. In this case, we can write

$$\sum_{i=1}^{p} \lambda_i \beta_i^{s*2}\zeta_i^2 = \sum_{i=1}^{p} \lambda_i \beta_i^2 \frac{(1 - \zeta_i)^2 \zeta_i^2}{\left((1 - \zeta_i)^2 + \frac{\Omega}{1-\Omega}\zeta_i^2\right)^2}.$$

Similar to the previous proposition and as discussed in Section 4, to analyze the model's inherent error, we set $\sigma_t^2 = O(n^{-\gamma})$ where $\gamma$ is the exponent characterizing the scaling law of the test risk in the noiseless setting. It is previously stated by Cui et al. (2022); Simon et al. (2024) that the test risk of ridgeless overparameterized linear regression can be described in the scaling sense as $\mathbf{err} = \Theta(n^{-\beta+1})$ when $\beta < 2\alpha + 1$. We will proceed by considering two cases based on the relationship between $\beta$ and $2\alpha + 1$.

**Case 1:** $\beta < 2\alpha + 1$

Consider the interval of $i$'s satisfying $\zeta_i > 1 - \Omega$ and $\zeta_i^2 < 1 - \Omega$. By Proposition 5, we have

$$\zeta_i > 1 - \Omega \iff i > nC_1 + O(1), \quad \text{where} \quad C_1 = \frac{\alpha \sin(\pi/\alpha)}{\pi(\alpha - 1)^{1/\alpha}}.$$

$$\zeta_i^2 > 1 - \Omega \iff i > nC_2 + O(1), \quad \text{where} \quad C_2 = \frac{\alpha \sin(\pi/\alpha)}{\pi(\sqrt{\alpha} - 1)^{1/\alpha}}.$$

Let $\omega_n$ be defined as in the previous proposition and define $\phi_n = nC_1 + O(1)$. Then, the interval of interest corresponds to the set of indices $i$ such that $\phi_n < i < \omega_n$. Within this interval, we observe

$$(1 - \zeta_i)^2 \zeta_i^2 \geq \left(1 - \sqrt{(1 - \Omega)}\right)^2 (1 - \Omega)^2 = k_1$$

$$(1 - \zeta_i)^2 + \frac{\Omega}{1 - \Omega}\zeta_i^2 \leq 1 + \frac{\Omega}{1 - \Omega} = k_2$$

Using the fact from Proposition 4 that $\Omega = \dfrac{\alpha - 1}{\alpha} - O(n^{-1})$ tells us $k_1 = \Theta(1)$ and $k_2 = \Theta(1)$. Utilizing these bounds, we obtain

$$
\begin{aligned}
\mathcal{R}_{om}^*(\boldsymbol{\beta}^{s2t}) &\geq \frac{\Omega}{1 - \Omega} \sum_{i=1}^{p} \lambda_i \beta_i^2 \frac{(1 - \zeta_i)^2 \zeta_i^2}{\left((1 - \zeta_i)^2 + \frac{\Omega}{1-\Omega}\zeta_i^2\right)^2} \geq \frac{\Omega}{1 - \Omega} \sum_{i=\phi_n}^{\omega_n} i^{-\beta} \frac{(1 - \zeta_i)^2 \zeta_i^2}{\left((1 - \zeta_i)^2 + \frac{\Omega}{1-\Omega}\zeta_i^2\right)^2} \\
&\geq \frac{\Omega}{1 - \Omega} \sum_{i=\phi_n}^{\omega_n} i^{-\beta} \frac{k_1}{k_2} \\
&\geq n^{-\beta+1} \frac{k_1}{k_2} \frac{\Omega}{1 - \Omega} \left( \frac{(\phi_n/n)^{-\beta+1} - (\omega_n/n)^{-\beta+1}}{\beta - 1} \right) \\
&\overset{(a)}{=} \Theta(n^{-\beta+1}),
\end{aligned}
$$

where (a) follows from the fact $\omega_n/n = \Theta(1)$, $\phi_n/n = \Theta(1)$, and $\Omega = \Theta(1)$. This implies that
$$
\mathcal{R}_{om}^*(\boldsymbol{\beta}^{s2t}) = \Omega(n^{-\beta+1}).
$$

Recall that the optimal surrogate-to-target improves over the risk of the standard target model, thus $\mathcal{R}_{om}^*(\boldsymbol{\beta}^{s2t}) = O(n^{-\beta+1})$. We therefore conclude $\mathcal{R}_{om}^*(\boldsymbol{\beta}^{s2t}) = \Theta(n^{-\beta+1})$ for this case.

**Case 2:** $\beta > 2\alpha + 1$

In this case, we have

$$
\begin{aligned}
\mathcal{R}_{om}^*(\boldsymbol{\beta}^{s2t}) &\geq \frac{\Omega}{1 - \Omega} \sum_{i=1}^{p} \lambda_i \beta_i^2 \frac{(1 - \zeta_i)^2 \zeta_i^2}{\left((1 - \zeta_i)^2 + \frac{\Omega}{1-\Omega}\zeta_i^2\right)^2} = \frac{\Omega}{1 - \Omega} \sum_{i=1}^{p} \lambda_i \beta_i^2 \zeta_i^2 \frac{(1 - \zeta_i)^2}{\left((1 - \zeta_i)^2 + \frac{\Omega}{1-\Omega}\zeta_i^2\right)^2} \\
&\geq \sum_{i:\zeta_i < 1 - \Omega} \lambda_i \zeta_i^2 \beta_i^2 \frac{\Omega^3}{\left(1 + \frac{\Omega}{1-\Omega}\right)^2 (1 - \Omega)} = \sum_{i=1}^{\phi_n} \frac{i^{-\beta}}{(1 + \frac{1}{\tau_t} i^{-\alpha})^2} k_3,
\end{aligned}
$$

where $k_3 = \dfrac{\Omega^3}{\left(1 + \frac{\Omega}{1-\Omega}\right)^2 (1 - \Omega)} = \Theta(1)$. From Case 2 in Proposition 7, we already know that the same summation – with upper bound $\omega_n$ rather than $\phi_n$ – scales as $\Theta(n^{-2\alpha})$. Yet, since $\phi_n$ and $\omega_n$ have the same order $\Theta(n)$, the result remains. This gives $\mathcal{R}_{om}^*(\boldsymbol{\beta}^{s2t}) = \Omega(n^{-2\alpha})$, which eventually yields

$$
\mathcal{R}_{om}^*(\boldsymbol{\beta}^{s2t}) \leq \mathcal{R}_{om}(\boldsymbol{\beta}^t) = O(n^{-2\alpha}) \implies \mathcal{R}_{om}^*(\boldsymbol{\beta}^{s2t}) = \Theta(n^{-2\alpha}).
$$

Hence, this allows us to say that the scaling law doesn't improve even with the freedom to choose any $\boldsymbol{\beta}^s$. $\quad\square$

In Figures 4 and 5, we illustrate the scaling-law behavior of the test risk. In Figure 4a, the sample size $n$ varies from small values up to values close to $p$. Even though a linear trend is observable for smaller values of $n$ when $p = 500$ in the log-log scale, as $n$ approaches $p$, it is less apparent due to finite-sample effects of $p$. We note that the asymptotic approximations in Propositions 4 and 5 rely on $p$ being significantly large compared to $n$. Thus, Figure 4b considers the same experiment with a larger dimension ($p = 5000$) and the same range of $n$ values to satisfy $n \ll p$. In this scenario, we observe a clear linear behavior in the log-log plot, which is consistent with the scaling-law results presented in Proposition 6.

In Figure 5, we analyze the scaling-law behavior of the test risk under a two-stage framework where the sample sizes $m = n$ vary relative to a fixed dimension $p = 1000$. In this scenario, we observe a similar linear behavior in the log-log scale when $m = n$ is sufficiently small compared to $p = 1000$. Whereas, as the gap between $m = n$ and $p$ diminishes, the linearity in the figure disappears as the asymptotic approximations in Propositions 4 and 5 require $p$ to be significantly larger compared to $n$. These observations align well with our expectations and results in Section 4.

**Proposition 8** (Non-asymptotic analysis of $\tau$). *Suppose that $\Sigma \in \mathbb{R}^{p \times p}$ is diagonal and $\Sigma_{i,i} = \lambda_i = i^{-\alpha}$ for $1 < \alpha$. Assume that $n < pk$ for $k = \dfrac{3 + 2^{-\alpha}}{4 + 2^{-(\alpha-2)}}$. If $\tau_t$ satisfies*

$$
\sum_{i=1}^{p} \frac{\lambda_i}{\lambda_i + \tau_t} = n,
$$

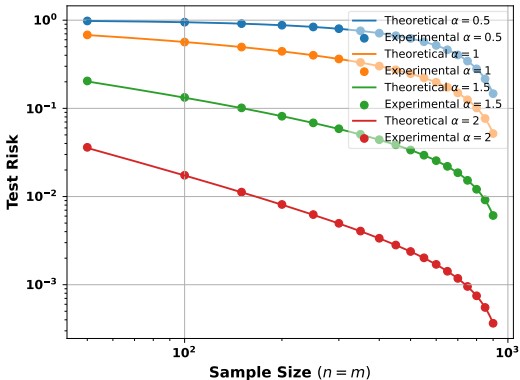

Figure 5: We compare the experimental two-stage risk with our estimated theoretical risk in log-log scale to demonstrate the scaling law. **Setting:** The feature size is $p = 1000$; the sample sizes $m = n$ change from 50 to 900 with increments of 50; both feature covariances follow the power-law structure $\lambda_i = i^{-\alpha}$ for $\alpha = 0.5, 1, 1.5$ and 2; the ground truth parameter $\boldsymbol{\beta}_\star$ is specified as $\beta_i = 1$.

then $cn^\alpha \leq \tau_t^{-1} \leq c\left(n + 1 + \frac{p+1}{\alpha-1}\right)^\alpha$ for $c = \left(\frac{\alpha \sin(\pi/\alpha)}{\pi}\right)^\alpha$.

*Proof.* In a similar vein to Simon et al. (2024), we have:

$$n = \sum_{i=1}^{p} \frac{i^{-\alpha}}{i^{-\alpha} + \tau_t} \leq \int_0^p \frac{x^{-\alpha}}{x^{-\alpha} + \tau_t} \, dx \leq \int_0^\infty \frac{x^{-\alpha}}{x^{-\alpha} + \tau_t} \, dx = \frac{\pi}{\alpha \sin(\pi/\alpha)} \tau^{-1/\alpha}.$$

Since the summand is decreasing, we can bound the Riemann sum by an integral, thus:

$$n = \sum_{i=1}^{p} \frac{i^{-\alpha}}{i^{-\alpha} + \tau_t} \geq \int_1^{p+1} \frac{x^{-\alpha}}{x^{-\alpha} + \tau_t} \, dx$$

$$= \int_0^\infty \frac{x^{-\alpha}}{x^{-\alpha} + \tau_t} \, dx - \int_0^1 \frac{x^{-\alpha}}{x^{-\alpha} + \tau_t} \, dx - \int_{p+1}^\infty \frac{x^{-\alpha}}{x^{-\alpha} + \tau_t} \, dx$$

$$= \frac{\pi}{\alpha \sin(\pi/\alpha)} \tau_t^{-1/\alpha} - \int_0^1 \frac{x^{-\alpha}}{x^{-\alpha} + \tau_t} \, dx - \int_{p+1}^\infty \frac{x^{-\alpha}}{x^{-\alpha} + \tau_t} \, dx$$

$$\geq \frac{\pi}{\alpha \sin(\pi/\alpha)} \tau_t^{-1/\alpha} - 1 - \int_{p+1}^\infty \frac{1}{1 + \tau_t x^\alpha} \, dx$$

$$\geq \frac{\pi}{\alpha \sin(\pi/\alpha)} \tau_t^{-1/\alpha} - 1 - \int_{p+1}^\infty \frac{1}{\tau_t x^\alpha} \, dx$$

$$= \frac{\pi}{\alpha \sin(\pi/\alpha)} \tau_t^{-1/\alpha} - 1 - \left[\frac{\tau_t^{-1} x^{-\alpha+1}}{-\alpha + 1}\right]_{p+1}^\infty$$

$$= \frac{\pi}{\alpha \sin(\pi/\alpha)} \tau_t^{-1/\alpha} - \frac{\tau_t^{-1}(p+1)^{-\alpha+1}}{\alpha - 1} - 1.$$

Recalling that $\alpha > 1$ and assuming $\tau_t^{-1} < p^\alpha$, we derive:

$$\frac{\pi}{\alpha \sin(\pi/\alpha)} \tau_t^{-1/\alpha} - 1 - \frac{p+1}{\alpha-1} \leq n \leq \frac{\pi}{\alpha \sin(\pi/\alpha)} \tau_t^{-1/\alpha}$$

$$\iff \left(\frac{n\alpha \sin(\pi/\alpha)}{\pi}\right)^\alpha \leq \tau_t^{-1} \leq \left(\frac{\left(n + 1 + \frac{p+1}{\alpha-1}\right)\alpha \sin(\pi/\alpha)}{\pi}\right)^\alpha.$$

We conclude by proving that $\tau_t^{-1} < p^\alpha$. For the sake of contradiction, assume that $\tau_t^{-1} \geq p^\alpha$. Then,

$$n = \sum_{i=1}^{p} \frac{1}{1 + i^\alpha \tau_t} = \sum_{i=1}^{p/2} \frac{1}{1 + i^\alpha \tau_t} + \sum_{i=p/2+1}^{p} \frac{1}{1 + i^\alpha \tau_t}$$

$$\geq \sum_{i=1}^{p/2} \frac{1}{1 + \frac{1}{2^\alpha}} + \sum_{i=p/2+1}^{p} \frac{1}{1 + 1}$$

$$= p\left(\frac{3 + \frac{1}{2^\alpha}}{4 + \frac{1}{2^{\alpha-2}}}\right),$$

which contradicts our assumption that $n < pk$. $\qquad\square$

**Proposition 9** (Non-asymptotic analysis of $\Omega$). *Suppose that $\Sigma \in \mathbb{R}^{p \times p}$ is diagonal and $\Sigma_{i,i} = \lambda_i = i^{-\alpha}$ for $1 < \alpha$. Let $\tau_t$ be defined as in Proposition 8 and assume that $pk_1 + \frac{\alpha^2}{(\alpha-1)^2} < n < pk_2$ for $k_1 = \frac{\alpha}{(\alpha-1)^2}$ and $k_2 = \frac{3 + 2^{-\alpha}}{4 + 2^{-(\alpha-2)}}$. Let $\Omega$ be the solution to*

$$n\Omega = \sum_{i=1}^{p} \left(\frac{\lambda_i}{\lambda_i + \tau_t}\right)^2.$$

*Then,*

$$\Omega > \frac{\alpha - 1}{\alpha} - \frac{1}{\alpha}\left(\frac{n + 1 + \frac{p+1}{\alpha-1}}{p+1}\right)^{2\alpha-1} - \frac{1}{n}.$$

*Proof.* Since the summand is monotonically decreasing, we can lower bound the sum by the following integral and manipulate:

$$n\Omega = \sum_{i=1}^{p} \left(\frac{i^{-\alpha}}{i^{-\alpha} + \tau_t}\right)^2 \geq \int_1^{p+1} \left(\frac{x^{-\alpha}}{x^{-\alpha} + \tau_t}\right)^2 dx$$

$$= \int_0^\infty \left(\frac{x^{-\alpha}}{x^{-\alpha} + \tau_t}\right)^2 dx - \int_0^1 \left(\frac{x^{-\alpha}}{x^{-\alpha} + \tau_t}\right)^2 dx - \int_{p+1}^\infty \left(\frac{x^{-\alpha}}{x^{-\alpha} + \tau_t}\right)^2 dx$$

$$= \frac{\pi(\alpha - 1)}{\alpha^2 \sin(\pi/\alpha)}\tau_t^{-1/\alpha} - \int_0^1 \left(\frac{x^{-\alpha}}{x^{-\alpha} + \tau_t}\right)^2 dx - \int_{p+1}^\infty \left(\frac{x^{-\alpha}}{x^{-\alpha} + \tau_t}\right)^2 dx$$

$$\geq \frac{\pi(\alpha - 1)}{\alpha^2 \sin(\pi/\alpha)}\tau_t^{-1/\alpha} - 1 - \int_{p+1}^\infty \left(\frac{1}{1 + \tau_t x^\alpha}\right)^2 dx$$

$$\geq \frac{\pi(\alpha - 1)}{\alpha^2 \sin(\pi/\alpha)}\tau_t^{-1/\alpha} - 1 - \int_{p+1}^\infty \frac{1}{(\tau_t x^\alpha)^2} dx$$

$$= \frac{\pi(\alpha - 1)}{\alpha^2 \sin(\pi/\alpha)}\tau_t^{-1/\alpha} - 1 - \left[\frac{\tau_t^{-2} x^{-2\alpha+1}}{-2\alpha + 1}\right]_{p+1}^\infty$$

$$= \frac{\pi(\alpha - 1)}{\alpha^2 \sin(\pi/\alpha)}\tau_t^{-1/\alpha} - \frac{\tau_t^{-2}(p+1)^{-2\alpha+1}}{2\alpha - 1} - 1.$$

Let's now utilize the upper and lower bounds for $\tau_t^{-1}$ from Proposition 8. Substituting, we have

$$
\begin{aligned}
n\Omega &\geq \frac{\pi(\alpha-1)}{\alpha^2 \sin(\pi/\alpha)} \frac{n\alpha \sin(\pi/\alpha)}{\pi} - \frac{\tau_t^{-2}(p+1)^{-2\alpha+1}}{2\alpha-1} - 1 \\
&= \frac{n(\alpha-1)}{\alpha} - \frac{\tau_t^{-2}(p+1)^{-2\alpha+1}}{2\alpha-1} - 1 \\
&\geq \frac{n(\alpha-1)}{\alpha} - \left(\frac{\left(n+1+\frac{p+1}{\alpha-1}\right)\alpha \sin(\pi/\alpha)}{\pi(p+1)}\right)^{2\alpha} \frac{p+1}{2\alpha-1} - 1
\end{aligned}
$$

Dividing both left and right-hand side by $n$ gives:

$$
\begin{aligned}
\implies \Omega &> \frac{\alpha-1}{\alpha} - \left(\frac{\left(n+1+\frac{p+1}{\alpha-1}\right)\alpha \sin(\pi/\alpha)}{\pi(p+1)}\right)^{2\alpha} \frac{p+1}{n(2\alpha-1)} - \frac{1}{n} \\
&> \frac{\alpha-1}{\alpha} - \frac{1}{2\alpha-1} \frac{n+1+\frac{p+1}{\alpha-1}}{n}\left(\frac{n+1+\frac{p+1}{\alpha-1}}{p+1}\right)^{2\alpha-1} - \frac{1}{n}, \\
&> \frac{\alpha-1}{\alpha} - \frac{1}{\alpha}\left(\frac{n+1+\frac{p+1}{\alpha-1}}{p+1}\right)^{2\alpha-1} - \frac{1}{n},
\end{aligned}
$$

since $\frac{\alpha \sin(\pi/\alpha)}{\pi} < 1$ for $\alpha > 1$ and $n+1+\frac{p+1}{\alpha-1} < n^{\frac{2\alpha-1}{\alpha}}$ by our assumption on $n$. $\qquad\square$

**Proposition 10.** *Under the assumption that*

$$
\max\left(2\alpha, p\frac{\alpha}{(\alpha-1)^2} + \frac{\alpha^2}{(\alpha-1)^2}\right) < n < \min\left((p+1)\frac{\alpha-2}{\alpha}, p\left(\frac{3+2^{-\alpha}}{4+2^{-(\alpha-2)}}\right), p\frac{\pi\left(\sqrt{\frac{2\alpha}{5}}-1\right)^{1/\alpha}}{\alpha \sin(\pi/\alpha)} - \frac{p+1}{\alpha-1}\right) - 1
$$

*and $4 < \alpha$, we can find a masked surrogate-to-target setting that improves over the risk of the standard target model by selecting all features $i$ such that $\zeta_i^2 > 1 - \Omega$.*

*Proof.* From Proposition 9, we have

$$
\Omega > \frac{\alpha-1}{\alpha} - \frac{1}{\alpha}\left(\frac{n+1+\frac{p+1}{\alpha-1}}{p+1}\right)^{2\alpha-1} - \frac{1}{n}.
$$

It's then enough to show that we can find a set of $i$'s such that

$$
\zeta_i^2 > \frac{1}{\alpha} + \frac{1}{\alpha}\left(\frac{n+1+\frac{p+1}{\alpha-1}}{p+1}\right)^{2\alpha-1} + \frac{1}{n}.
$$

From the proof of Proposition 5, we know that

$$
\zeta_i^2 > c' \iff i > \tau_t^{-1/\alpha}\left(\frac{\sqrt{c'}}{1-\sqrt{c'}}\right)^{1/\alpha}.
$$

Hence, using the bound on $\tau_t^{-1}$ from Proposition 8, it's enough to find indices $i$ such that

$$
i > \frac{\alpha \sin(\pi/\alpha)}{\pi}\left(n+1+\frac{p+1}{\alpha-1}\right)\left(\frac{\sqrt{c'}}{1-\sqrt{c'}}\right)^{1/\alpha} \quad \text{where } c' = \frac{1}{\alpha} + \frac{1}{\alpha}\left(\frac{n+1+\frac{p+1}{\alpha-1}}{p+1}\right)^{2\alpha-1} + \frac{1}{n}. \tag{29}
$$

By our assumption $p + 1 > n + 1 + \frac{p+1}{\alpha-1}$ and $n > 2\alpha$, we obtain that $\frac{5}{2\alpha} > c'$. Since $\left(\frac{\sqrt{x}}{1-\sqrt{x}}\right)^{1/\alpha}$ is increasing with $x$ when $0 \leq x \leq 1$, we have

$$\left(\frac{1}{\sqrt{\frac{2\alpha}{5}} - 1}\right)^{1/\alpha} \geq \left(\frac{\sqrt{c'}}{1 - \sqrt{c'}}\right)^{1/\alpha}.$$

Then, to ensure the existence of an interval of $i$'s satisfying the above inequality, we choose

$$p - (p+1)\frac{\alpha \sin(\pi/\alpha)}{\pi(\alpha-1)}\left(\frac{1}{\sqrt{\frac{2\alpha}{5}} - 1}\right)^{1/\alpha} \geq (n+1)\frac{\alpha \sin(\pi/\alpha)}{\pi}\left(\frac{1}{\sqrt{\frac{2\alpha}{5}} - 1}\right)^{1/\alpha}$$

$$\Longleftrightarrow p\frac{\pi\left(\sqrt{\frac{2\alpha}{5}} - 1\right)^{1/\alpha}}{\alpha \sin(\pi/\alpha)} - \frac{p+1}{\alpha-1} \geq n+1$$

which follows from our assumption on $n$. Thus, discarding the features $i$ provided in the interval (29) will strictly improve the test risk of the masked surrogate-to-target model over the standard target model. □

One can verify that our assumptions are coherent because they ensure a non-empty interval for $n$ when $\alpha > 4$ as the coefficients of $p$ are positive and $\frac{\alpha}{(\alpha-1)^2}$ is smaller compared to the other three coefficients of $p$ in the minimum function.

## C  PROOFS FOR SECTION 5

**Theorem 4** (Distributional characterization, Han & Xu (2023)). *Let $\kappa_s = p/m > 1$ and suppose that, for some $M > 1$, $1/M \leq \kappa_s, \sigma_s^2 \leq M$ and $\|\Sigma_s\|_{op}, \|\Sigma_s^{-1}\|_{op} \leq M$. Let $\tau_s \in \mathbb{R}$ be the unique solution of the following equation:*

$$\kappa_s^{-1} - \frac{\lambda_s}{\tau} = \frac{1}{p}tr\left((\Sigma_s + \tau_s I)^{-1}\Sigma_s\right). \tag{30}$$

*We define the random variable $X^s_{\kappa_s,\sigma_s^2}(\Sigma_s, \beta_\star, g_s)$ based on $g_s \sim \mathcal{N}(0, I)$ and the function $\gamma_s : \mathbb{R}^p \to \mathbb{R}$ as follows:*

$$X^s_{\kappa_s,\sigma_s^2}(\Sigma_s, \beta_\star, g_s) := (\Sigma_s + \tau_s I)^{-1}\Sigma_s\left[\beta_\star + \frac{\Sigma_s^{-1/2}\gamma_s(\beta_\star)g_s}{\sqrt{p}}\right]$$

$$\gamma_s^2(\beta_\star) := \kappa_s\left(\sigma_s^2 + \mathbb{E}_{g_s}[\|\Sigma_s^{1/2}(X^s_{\kappa_s,\sigma_s^2}(\Sigma_s, \beta_\star, g_s) - \beta_\star)\|_2^2]\right). \tag{31}$$

*Then, for any $L$-Lipschitz function $f : \mathbb{R}^p \to \mathbb{R}$ where $L < L(M)$, there exists a constant $C = C(M)$ such that for any $\varepsilon \in (0, 1/2]$, we have the following:*

$$\sup_{\beta_\star \in B(R)} \mathbb{P}(\sup_{\lambda_s \in [0,M]} \left|f(\beta^s) - \mathbb{E}_{g_s}[f(X^w_{\kappa_s,\sigma_s^2}(\Sigma_s, \beta_\star, g_s))]\right| \geq \varepsilon) \leq Cpe^{-p\varepsilon^4/C}, \tag{32}$$

*where $R < M$.*

**Definition 3.** *Recall the definition of $\tau_t$ and $\gamma_t$ in Theorem 1. Let $\kappa_s = p/m > 1$ and define $\tau_s \in \mathbb{R}$ similarly to $\tau_t$. We define the random variable $X^s_{\kappa_s,\sigma_s^2}$ based on $g_s \sim \mathcal{N}(0, I)$ and the function $\gamma_s : \mathbb{R}^p \to \mathbb{R}$ as follows:*

$$X^s_{\kappa_s,\sigma_s^2}(\Sigma_s, \beta_\star, g_s) := (\Sigma_s + \tau_s I)^{-1}\Sigma_s\left[\beta_\star + \frac{\Sigma_s^{-1/2}\gamma_s(\beta_\star)g_s}{\sqrt{p}}\right]$$

$$\gamma_s^2(\beta_\star) := \kappa_s\left(\sigma_s^2 + \mathbb{E}_{g_s}[\|\Sigma_s^{1/2}(X^s_{\kappa_s,\sigma_s^2}(\Sigma_s, \beta_\star, g_s) - \beta_\star)\|_2^2]\right).$$

Let $\dot{k} = (\kappa_s, \kappa_t)$, $\dot{\Sigma} = (\Sigma_s, \Sigma_t)$, and $\dot{\sigma} = (\sigma_s^2, \sigma_t^2)$. Then, we define the asymptotic risk estimate as

$$\bar{\mathcal{R}}_{\dot{k},\dot{\sigma}}(\dot{\Sigma}, \boldsymbol{\beta}_\star) = \|\Sigma_t^{1/2}\left(\boldsymbol{I} - (\Sigma_t + \tau_t\boldsymbol{I})^{-1}\Sigma_t(\Sigma_s + \tau_s\boldsymbol{I})^{-1}\Sigma_s\right)\boldsymbol{\beta}_\star\|_2^2 + \frac{\mathbb{E}_{\boldsymbol{\beta}^s \sim X_{\kappa_s,\sigma_s^2}^s}[\gamma_t^2(\boldsymbol{\beta}^s)]}{p}\,\mathrm{tr}\left(\Sigma_t^2(\Sigma_t + \tau_t\boldsymbol{I})^{-2}\right)$$
$$+ \frac{\gamma_s^2(\boldsymbol{\beta}_\star)}{p}\,\mathrm{tr}\left(\Sigma_s^{1/2}(\Sigma_s + \tau_s\boldsymbol{I})^{-1}\Sigma_t(\Sigma_t + \tau_t\boldsymbol{I})^{-1}\Sigma_t(\Sigma_t + \tau_t\boldsymbol{I})^{-1}\Sigma_t(\Sigma_s + \tau_s\boldsymbol{I})^{-1}\Sigma_s^{1/2}\right).$$

**Theorem 2.** *Suppose that, for some constant $M_t > 1$, we have $1/M_t \leq \kappa_s, \sigma_s^2, \kappa_t, \sigma_t^2 \leq M_t$ and $\|\Sigma_s\|_{op}, \left\|\Sigma_s^{-1}\right\|_{op}, \|\Sigma_t\|_{op}, \left\|\Sigma_s^{-1}\right\|_{op} \leq M_t$. Consider the surrogate-to-target model defined in Section 2, and let $\mathcal{R}(\boldsymbol{\beta}^{s2t})$ represent its risk when $\boldsymbol{\beta}_\star$ is given. Recall the definition of $\dot{\Sigma}, \dot{k}, \dot{\sigma}$ and $\bar{\mathcal{R}}_{\dot{k},\dot{\sigma}}$ in Definition 3. Then, there exists a constant $C = C(M_t)$ such that for any $\varepsilon \in (0, 1/2]$, the following holds when $R + 1 < M_t$:*

$$\sup_{\boldsymbol{\beta}_\star \in \boldsymbol{B}_p(R)} \mathbb{P}\bigl(\bigl|\mathcal{R}(\boldsymbol{\beta}^{s2t}) - \bar{\mathcal{R}}_{\dot{k},\dot{\sigma}}(\dot{\Sigma}, \boldsymbol{\beta}_\star)\bigr| \geq \varepsilon\bigr) \leq Cpe^{-p\varepsilon^4/C}.$$

*Proof.* Define a function $f_1 : \mathbb{R}^p \to \mathbb{R}$ as $f_1(\boldsymbol{x}) = \|\Sigma_t^{1/2}(\boldsymbol{x} - \boldsymbol{\beta}_\star)\|_2^2$. The gradient of this function is

$$\|\nabla f_1(\boldsymbol{x})\|_2 = \|2\Sigma_t(\boldsymbol{x} - \boldsymbol{\beta}_\star)\|_2 \leq 2\|\Sigma_t\|_{\mathrm{op}}\|\boldsymbol{x} - \boldsymbol{\beta}_\star\|_2.$$

Using Proposition 12, there exists an event $E$ with $\mathbb{P}(E^c) \leq C_t e^{-p/C_t}$ where $C_t = C_t(M_t, \frac{M_t-R}{2})$ with the definition of $M_t$ in Proposition 12, such that $f_1(\boldsymbol{\beta}^{s2t})$ is $2M_t^2$-Lipschitz if $\boldsymbol{\beta}_\star \in \boldsymbol{B}_p(R)$. Applying Theorem 4 on the target model, there exists a constant $\bar{C}_s = \bar{C}_s(M_t)$ such that for any $\varepsilon \in (0, 1/2]$, we obtain

$$\sup_{\boldsymbol{\beta}^s \in \boldsymbol{B}(\frac{M_t+R}{2})} \mathbb{P}\left(\left|f(\boldsymbol{\beta}^{s2t}) - \mathbb{E}_{\boldsymbol{g}_t}[f(X_{\kappa_t,\sigma_t^2}^t(\Sigma_t, \boldsymbol{\beta}^s, \boldsymbol{g}_t))]\right| \geq \varepsilon\right) \leq Cpe^{-p\varepsilon^4/C}, \tag{33}$$

where $f(\boldsymbol{\beta}^{s2t}) = \mathcal{R}(\boldsymbol{\beta}^{s2t})$ and

$$X_{\kappa_t,\sigma_t^2}^t(\Sigma_t, \boldsymbol{\beta}^s, \boldsymbol{g}_t) = (\Sigma_t + \tau_t\boldsymbol{I})^{-1}\Sigma_t\left[\boldsymbol{\beta}^s + \frac{\Sigma_t^{-1/2}\gamma_t(\boldsymbol{\beta}^s)\boldsymbol{g}_t}{\sqrt{p}}\right].$$

Furthermore,

$$\mathbb{E}_{\boldsymbol{g}_t}\left[f(X_{\kappa_t,\sigma_t^2}^s(\Sigma_t, \boldsymbol{\beta}^s, \boldsymbol{g}_t))\right] = \mathbb{E}_{\boldsymbol{g}_t}\left[\|\Sigma_t^{1/2}(\boldsymbol{\theta}_1(\boldsymbol{\beta}^s - \boldsymbol{\beta}_\star) - (\boldsymbol{I} - \boldsymbol{\theta}_1)\boldsymbol{\beta}_\star + \boldsymbol{\theta}_2\gamma_t(\boldsymbol{\beta}^s))\|_2^2\right]$$
$$= (\boldsymbol{\beta}^s - \boldsymbol{\beta}_\star)^\top\boldsymbol{\theta}_1^\top\Sigma_t\boldsymbol{\theta}_1(\boldsymbol{\beta}^s - \boldsymbol{\beta}_\star) + \gamma_t^2(\boldsymbol{\beta}^s)\mathbb{E}_{\boldsymbol{g}_t}[\boldsymbol{\theta}_2^\top\Sigma_t\boldsymbol{\theta}_2]$$
$$+ \boldsymbol{\beta}_\star^\top(\boldsymbol{I} - \boldsymbol{\theta}_1)^\top\Sigma_t(\boldsymbol{I} - \boldsymbol{\theta}_1)\boldsymbol{\beta}_\star - 2\boldsymbol{\beta}_\star^\top(\boldsymbol{I} - \boldsymbol{\theta}_1)^\top\Sigma_t\boldsymbol{\theta}_1(\boldsymbol{\beta}^s - \boldsymbol{\beta}_\star), \tag{34}$$

where $\boldsymbol{\theta}_1 := (\Sigma_t + \tau_t\boldsymbol{I})^{-1}\Sigma_t$ and $\boldsymbol{\theta}_2 := (\Sigma_t + \tau_t\boldsymbol{I})^{-1}\Sigma_t^{1/2}\frac{\boldsymbol{g}_t}{\sqrt{p}}$. Let $E(M_t, \frac{M_t-R}{2})$ be the event defined in Proposition 11. Let $f_2 : \mathbb{R}^p \to \mathbb{R}$ be defined as $f_2(\boldsymbol{x}) := (\boldsymbol{x} - \boldsymbol{\beta}_\star)^\top\boldsymbol{\theta}_1^\top\Sigma_t\boldsymbol{\theta}_1(\boldsymbol{x} - \boldsymbol{\beta}_\star)$. By Proposition 13, the function $f_2$ is $2M_t^2$-Lipschitz if $\boldsymbol{\beta}_\star \in \boldsymbol{B}_p(R)$ on the event $E(M_t, \frac{M_t-R}{2})$. Applying Theorem 4 on the surrogate model, there exists a constant $\bar{C}_{w,1} = \bar{C}_{w,1}(M_t)$ such that for any $\varepsilon \in (0, 1/2]$, we obtain

$$\sup_{\boldsymbol{\beta}_\star \in \boldsymbol{B}_p(R)} \mathbb{P}\left(\left|f_2(\boldsymbol{\beta}^s) - \boldsymbol{\beta}_\star^\top(\boldsymbol{I} - \boldsymbol{\Phi}_1)^\top\boldsymbol{\theta}_1^\top\Sigma_t\boldsymbol{\theta}_1(\boldsymbol{I} - \boldsymbol{\Phi}_1)\boldsymbol{\beta}_\star - \gamma_s^2(\boldsymbol{\beta}_\star)\mathbb{E}_{\boldsymbol{g}_s}[\boldsymbol{\Phi}_2^\top\boldsymbol{\theta}_1^\top\Sigma_t\boldsymbol{\theta}_1\boldsymbol{\Phi}_2]\right| > \varepsilon\right) \leq \bar{C}_{w,1}pe^{-p\varepsilon^4/\bar{C}_{w,1}}, \tag{35}$$

where $\boldsymbol{\Phi}_1 := (\Sigma_s + \tau_s\boldsymbol{I})^{-1}\Sigma_s$ and $\boldsymbol{\Phi}_2 := (\Sigma_s + \tau_s\boldsymbol{I})^{-1}\Sigma_s^{1/2}\frac{\boldsymbol{g}_s}{\sqrt{p}}$.

Let $f_3 : \mathbb{R}^p \to \mathbb{R}$ be defined as $f_3(\boldsymbol{x}) := \gamma_t^2(\boldsymbol{x})\boldsymbol{\theta}_2^\top\Sigma_t\boldsymbol{\theta}_2$. By Proposition 14 and Proposition 2.1 in Han & Xu (2023), the function $f_3$ is $4M_t^2$-Lipschitz if $\boldsymbol{\beta}_\star \in \boldsymbol{B}_p(R)$ on the event $E(M_t, \frac{M_t-R}{2})$. Applying Theorem 4 on the surrogate model, there exists a constant $\bar{C}_{w,2} = \bar{C}_{w,2}(M_t)$ such that for any $\varepsilon \in (0, 1/2]$, we obtain

$$\sup_{\boldsymbol{\beta}_\star \in \boldsymbol{B}_p(R)} \mathbb{P}\left(\left|f_3(\boldsymbol{\beta}^s) - \mathbb{E}_{\boldsymbol{\beta}^s \sim X^s}[\gamma_t^2(\boldsymbol{\beta}^s)]\mathbb{E}_{\boldsymbol{g}_t}[\boldsymbol{\theta}_2^\top\Sigma_t\boldsymbol{\theta}_2]\right| > \varepsilon\right) \leq \bar{C}_{w,2}pe^{-p\varepsilon^4/\bar{C}_{w,2}}. \tag{36}$$

Let $f_4 : \mathbb{R}^p \to \mathbb{R}$ as $f_4(\boldsymbol{x}) := -2\boldsymbol{\beta}_\star^\top (\boldsymbol{I} - \boldsymbol{\theta}_1)^\top \boldsymbol{\Sigma}_t \boldsymbol{\theta}_1 (\boldsymbol{x} - \boldsymbol{\beta}_\star)$. By Proposition 15 and Proposition 2.1 in Han & Xu (2023), the function $f_4$ is $2M_t^2$–Lipschitz if $\boldsymbol{\beta}_\star \in \boldsymbol{B}_p(R)$ on the event $E(M_t, \frac{M_t - R}{2})$. Applying Theorem 4 on the surrogate model, there exists a constant $\bar{C}_{w,3} = \bar{C}_{w,3}(M_t)$ such that for any $\varepsilon \in (0, 1/2]$, we obtain

$$\sup_{\boldsymbol{\beta}_\star \in \boldsymbol{B}_p(R)} \mathbb{P}\left( \left| f_4(\boldsymbol{\beta}^s) - 2\left[ \boldsymbol{\beta}_\star^\top (\boldsymbol{I} - \boldsymbol{\theta}_1)^\top \boldsymbol{\Sigma}_t \boldsymbol{\theta}_1 (\boldsymbol{\Phi}_1 - \boldsymbol{I}) \boldsymbol{\beta}_\star \right] \right| > \varepsilon \right) \le \bar{C}_{w,3} p e^{-p\varepsilon^4/\bar{C}_{w,3}}. \tag{37}$$

By the definition of these functions, we have

$$\mathbb{E}_{\boldsymbol{g}_t}\left[ f(X^s_{\kappa_t, \sigma_t^2}(\boldsymbol{\Sigma}_t, \boldsymbol{\beta}^s, \boldsymbol{g}_t)) \right] - \boldsymbol{\beta}_\star^\top (\boldsymbol{I} - \boldsymbol{\theta}_1)^\top \boldsymbol{\Sigma}_t (\boldsymbol{I} - \boldsymbol{\theta}_1) \boldsymbol{\beta}_\star = f_2(\boldsymbol{\beta}^s) + f_3(\boldsymbol{\beta}^s) - f_4(\boldsymbol{\beta}^s) \tag{38}$$

By the definition of $\boldsymbol{\theta}_1, \boldsymbol{\theta}_2, \boldsymbol{\Phi}_1$, and $\boldsymbol{\Phi}_2$, we have

$$\bar{\mathcal{R}}_{\kappa,\bar{\sigma}}(\dot{\boldsymbol{\Sigma}}, \boldsymbol{\beta}_\star) - \boldsymbol{\beta}_\star^\top (\boldsymbol{I} - \boldsymbol{\theta}_1)^\top \boldsymbol{\Sigma}_t (\boldsymbol{I} - \boldsymbol{\theta}_1) \boldsymbol{\beta}_\star = \boldsymbol{\beta}_\star^\top (\boldsymbol{I} - \boldsymbol{\Phi}_1)^\top \boldsymbol{\theta}_1^\top \boldsymbol{\Sigma}_t \boldsymbol{\theta}_1 (\boldsymbol{I} - \boldsymbol{\Phi}_1) \boldsymbol{\beta}_\star + \gamma_s^2(\boldsymbol{\beta}_\star) \mathbb{E}_{\boldsymbol{g}_s}[\boldsymbol{\Phi}_2^\top \boldsymbol{\theta}_1^\top \boldsymbol{\Sigma}_t \boldsymbol{\theta}_1 \boldsymbol{\Phi}_2]$$
$$+ \mathbb{E}_{\boldsymbol{\beta}^s \sim X^s}[\gamma_t^2(\boldsymbol{\beta}^s)] \mathbb{E}_{\boldsymbol{g}_t}[\boldsymbol{\theta}_2^\top \boldsymbol{\Sigma}_t \boldsymbol{\theta}_2] - 2\left[ \boldsymbol{\beta}_\star^\top (\boldsymbol{I} - \boldsymbol{\theta}_1)^\top \boldsymbol{\Sigma}_t \boldsymbol{\theta}_1 (\boldsymbol{\Phi}_1 - \boldsymbol{I}) \boldsymbol{\beta}_\star \right]. \tag{39}$$

Using (38)-(39) and applying a union bound on (33), (35), (36), and (37), we obtain the advertised claim. $\square$

**Proposition 11.** *Suppose that, for some $M_t > 1$, $1/M_t \le \kappa_s, \sigma_s^2 \le M_t$ and $\|\boldsymbol{\Sigma}_s\|_{op}, \left\|\boldsymbol{\Sigma}_s^{-1}\right\|_{op} \le M_t$. For every $c_s > 0$, there exists an event $E(M_t, c_s)$ with $\mathbb{P}((E(M_t, c_s))^c) \le C_s e^{-p/C_s}$ where $C_s = C_s(M_t, c_s)$ such that*

$$\|\boldsymbol{\beta}^s\|_2 \le \|\boldsymbol{\beta}_\star\|_2 + c_s \quad \text{and} \quad \|\boldsymbol{\beta}^s - \boldsymbol{\beta}_\star\|_2 \le \|\boldsymbol{\beta}_\star\|_2 + c_s.$$

*Proof.* By the definition of $\boldsymbol{\beta}^s$, we have

$$\begin{aligned} \boldsymbol{\beta}^s &= \tilde{X}^\top (\tilde{X}\tilde{X}^\top)^{-1} \tilde{\boldsymbol{y}} \\ &= \tilde{X}^\top (\tilde{X}\tilde{X}^\top)^{-1} \tilde{X}\boldsymbol{\beta}_\star + \tilde{X}^\top (\tilde{X}\tilde{X}^\top)^{-1} \tilde{\boldsymbol{z}}, \end{aligned} \tag{40}$$

where $\tilde{\boldsymbol{z}} \sim \mathcal{N}(\boldsymbol{0}, \sigma_s^2 \boldsymbol{I})$. By triangle inequality, we obtain

$$\begin{aligned} \|\boldsymbol{\beta}^s\|_2 &\le \|\tilde{X}^\top (\tilde{X}\tilde{X}^\top)^{-1} \tilde{X}\boldsymbol{\beta}_\star\|_2 + \|\tilde{X}^\top (\tilde{X}\tilde{X}^\top)^{-1} \tilde{\boldsymbol{z}}\|_2 \\ &\overset{(a)}{\le} \|\boldsymbol{\beta}_\star\|_2 + \|\tilde{X}^\top (\tilde{X}\tilde{X}^\top)^{-1} \tilde{\boldsymbol{z}}\|_2, \end{aligned} \tag{41}$$

where $(a)$ in above follows from the fact that $\tilde{X}^\top (\tilde{X}\tilde{X}^\top)^{-1} \tilde{X}$ is a projection matrix, and so all of its eigenvalues are either 0 or 1. Focusing on the second term of the RHS, we derive

$$\begin{aligned} \|\tilde{X}^\top (\tilde{X}\tilde{X}^\top)^{-1} \tilde{\boldsymbol{z}}\|_2^2 = \tilde{\boldsymbol{z}}^\top (\tilde{X}\tilde{X}^\top)^{-1} \tilde{\boldsymbol{z}} &= \frac{\tilde{\boldsymbol{z}}^\top}{\sqrt{p}} \left( \frac{\tilde{X}\tilde{X}^\top}{p} \right)^{-1} \frac{\tilde{\boldsymbol{z}}}{\sqrt{p}} \\ &\overset{(a)}{\le} \frac{\tilde{\boldsymbol{z}}^\top \tilde{\boldsymbol{z}}}{p} \left\| \left( \frac{\tilde{X}\tilde{X}^\top}{p} \right)^{-1} \right\|_{op}, \end{aligned} \tag{42}$$

where $(a)$ in the above inequality follows from Cauchy-Schwarz inequality. Using Bernstein's inequality, there exists an absolute constant $C_0 > 0$ that depends on $\sigma_s^2$ such that

$$\mathbb{P}\left( \frac{\tilde{\boldsymbol{z}}^\top \tilde{\boldsymbol{z}}}{p} - \sigma_s^2 > t \right) \le \exp\left\{ -c \min\left\{ \frac{pt^2}{4C_0^2}, \frac{pt}{2C_0} \right\} \right\}.$$

On the other hand, let $\tilde{\boldsymbol{Z}} = \tilde{X}\boldsymbol{\Sigma}_s^{-1/2}$, which means that the entries of $\tilde{\boldsymbol{Z}}$ are independent and normally distributed with zero mean and unit variance. Then,

$$\left\| \left( \frac{\tilde{X}\tilde{X}^\top}{p} \right)^{-1} \right\|_{op} = \left\| \left( \frac{\tilde{\boldsymbol{Z}}\boldsymbol{\Sigma}_s\tilde{\boldsymbol{Z}}^\top}{p} \right)^{-1} \right\|_{op} \le \left\| \boldsymbol{\Sigma}_s^{-1} \right\|_{op} \left\| \left( \frac{\tilde{\boldsymbol{Z}}\tilde{\boldsymbol{Z}}^\top}{p} \right)^{-1} \right\|_{op}. \tag{43}$$

Using Theorem 1.1 in Rudelson & Vershynin (2009), there exist absolute constants $C_1, C_2 > 0$ such that we have the following for every $\varepsilon > 0$

$$\mathbb{P}\left(\left\|\left(\frac{\tilde{Z}\tilde{Z}^\top}{p}\right)^{-1}\right\|_{\text{op}} \leq \varepsilon^2 \left(1 - \frac{1}{\kappa_s}\right)^2\right) \leq (C_1\varepsilon)^{p-m+1} + e^{-pC_2}. \tag{44}$$

By combining (42), (43), and (44), we obtain that

$$\mathbb{P}\left(\|\boldsymbol{\beta}^s\|_2 \leq \|\boldsymbol{\beta}_\star\|_2 + \varepsilon(1 - \frac{1}{\kappa_s})\sqrt{(t + \sigma_s^2)\left\|\boldsymbol{\Sigma}_s^{-1}\right\|_{\text{op}}}\right)$$

$$\leq (C_1\varepsilon)^{p-m+1} + e^{-pC_2} + e^{-c\min\left\{\frac{pt^2}{4C_0^2}, \frac{pt}{2C_0}\right\}}$$

The advertised claim for $\|\boldsymbol{\beta}^s\|_2$ follows when $\varepsilon$ is selected as $\varepsilon < \frac{1}{C_1e}$. For $\|\boldsymbol{\beta}^s - \boldsymbol{\beta}_\star\|_2$, using the definition of $\boldsymbol{\beta}^s$, we write as follows:

$$\|\boldsymbol{\beta}^s - \boldsymbol{\beta}_\star\|_2 = \|\tilde{X}^\top(\tilde{X}\tilde{X}^\top)^{-1}\tilde{X}\boldsymbol{\beta}_\star + \tilde{X}^\top(\tilde{X}\tilde{X}^\top)^{-1}\tilde{z} - \boldsymbol{\beta}_\star\|_2$$

$$\leq \|\tilde{X}^\top(\tilde{X}\tilde{X}^\top)^{-1}\tilde{X} - I\|_2\|\boldsymbol{\beta}_\star\|_2 + \|\tilde{X}^\top(\tilde{X}\tilde{X}^\top)^{-1}\tilde{z}\|_2$$

$$\overset{(a)}{\leq} \|\boldsymbol{\beta}_\star\|_2 + \|\tilde{X}^\top(\tilde{X}\tilde{X}^\top)^{-1}\tilde{z}\|_2, \tag{45}$$

where $(a)$ in the above inequalities follows from the fact that the eigenvalues of $\tilde{X}^\top(\tilde{X}\tilde{X}^\top)^{-1}\tilde{X} - I$ are either 1 or 0 as the eigenvalues of $\tilde{X}^\top(\tilde{X}\tilde{X}^\top)^{-1}\tilde{X}$ are either 1 or 0. The remaining part of this proof is identical to the previous part. □

**Corollary 2.** *Suppose that $\boldsymbol{\beta}^s \in \mathbb{R}^p$ is given, and for some $M_t > 1$, we have $1/M_t \leq \kappa_t, \sigma_t^2 \leq M_t$ and $\|\boldsymbol{\Sigma}_t\|_{op}, \left\|\boldsymbol{\Sigma}_t^{-1}\right\|_{op} \leq M_t$. For every $c_t > 0$, there exists an event $E(M_t, c_t)$ with $\mathbb{P}((E(M_t, c_t))^c) \leq C_t e^{-p/C_t}$ where $C_t = C_t(M_t, c_t)$ such that*

$$\|\boldsymbol{\beta}^{s2t}\|_2 \leq \|\boldsymbol{\beta}^s\|_2 + c_t \quad and \quad \|\boldsymbol{\beta}^{s2t} - \boldsymbol{\beta}^s\|_2 \leq \|\boldsymbol{\beta}^s\|_2 + c_t.$$

*Proof.* The result directly follows from the proof of Proposition 11. □

**Proposition 12.** *Suppose that, for some $M_t > 1$, $1/M_t \leq \kappa_t, \sigma_t^2 \leq M_t$ and $\|\boldsymbol{\Sigma}_t\|_{op}, \left\|\boldsymbol{\Sigma}_t^{-1}\right\|_{op} \leq M_t$. For every $c_t > 0$, there exists an event $E(M_t, c_t)$ with $\mathbb{P}((E(M_t, c_t))^c) \leq C_t e^{-p/C_t}$ where $C_t = C_t(M_t, c_t)$ such that we have the following on this event $E(M_t, c_t)$:*

$$\|\boldsymbol{\beta}^{s2t}\|_2 \leq \|\boldsymbol{\beta}_\star\|_2 + c_t \quad and \quad \|\boldsymbol{\beta}^{s2t} - \boldsymbol{\beta}_\star\|_2 \leq \|\boldsymbol{\beta}_\star\|_2 + c_t$$

*Proof.* By the definition of $\boldsymbol{\beta}^{s2t}$, we have the following:

$$\boldsymbol{\beta}^{s2t} = X(XX^\top)^{-1}X\boldsymbol{\beta}^s + X^\top(XX^\top)^{-1}z \tag{46}$$

where $z \sim \mathcal{N}(0, \sigma_t^2 I)$. Plugging (40) into (46), we obtain

$$\boldsymbol{\beta}^{s2t} = X(XX^\top)^{-1}X\left(\tilde{X}^\top(\tilde{X}\tilde{X}^\top)^{-1}\tilde{X}\boldsymbol{\beta}_\star + \tilde{X}^\top(\tilde{X}\tilde{X}^\top)^{-1}\tilde{z}\right) + X^\top(XX^\top)^{-1}z \tag{47}$$

Note that $X(XX^\top)^{-1}X$ and $\tilde{X}^\top(\tilde{X}\tilde{X}^\top)^{-1}\tilde{X}$ are projection matrices. Multiplication of two projection matrices results in a projection matrix. Using the fact that the eigenvalues of a projection matrix are either 1 or 0 in (47), we have

$$\|\boldsymbol{\beta}^{s2t}\|_2 \leq \|\boldsymbol{\beta}_\star\|_2 + \|\tilde{X}^\top(\tilde{X}\tilde{X}^\top)^{-1}\tilde{z}\|_2 + \|X^\top(XX^\top)^{-1}z\|_2 \tag{48}$$

By a similar reasoning used in (42),(43), and (44); there exist absolute constants $C_0, C_1, C_2, c > 0$ such that we have the following for every $\varepsilon, t > 0$:

$$\mathbb{P}\left(\|X^\top(XX^\top)^{-1}z\|_2 \leq \varepsilon(1 - \frac{1}{\kappa_t})\sqrt{(t + \sigma_t^2)\left\|\boldsymbol{\Sigma}_t^{-1}\right\|_{\text{op}}}\right)$$

$$\leq (C_1\varepsilon)^{p-n+1} + e^{-pC_2} + e^{-c\min\left\{\frac{pt^2}{4C_0^2}, \frac{pt}{2C_0}\right\}} \tag{49}$$

Similarly, for every $\tilde{\varepsilon} > 0$, there exist absolute constants $\tilde{C}_0, \tilde{C}_1, \tilde{C}_2, \tilde{c} > 0$ such that we have the following for every $\tilde{\varepsilon}, \tilde{t}$:

$$\mathbb{P}\left(\|\tilde{X}^\top(\tilde{X}\tilde{X}^\top)^{-1}\tilde{z}\|_2 \le \tilde{\varepsilon}(1 - \frac{1}{\kappa_s}) \sqrt{(\tilde{t} + \sigma_s^2) \left\|\Sigma_s^{-1}\right\|_{\text{op}}}\right)$$

$$\le (\tilde{C}_1\tilde{\varepsilon})^{p-m+1} + e^{-p\tilde{C}_2} + e^{-\tilde{c}\min\left\{\frac{p\tilde{t}^2}{4\tilde{c}_0^2}, \frac{p\tilde{t}}{2\tilde{c}_0}\right\}} \tag{50}$$

Note that $X, z, \tilde{X}$, and $\tilde{z}$ are independent of each other. Therefore, we can apply union bound on (49) and (50) with selecting $\varepsilon, t, \tilde{\varepsilon}$, and $\tilde{t}$ such that $\varepsilon < \frac{1}{C_1 e}$, $\frac{c_t}{2} < \varepsilon(1 - \frac{1}{\kappa_t}) \sqrt{(t + \sigma_t^2) \left\|\Sigma_t^{-1}\right\|_{\text{op}}}$, $\tilde{\varepsilon} < \frac{1}{\tilde{C}_1}$, and $\frac{c_t}{2} < \varepsilon(1 - \frac{1}{\kappa_s}) \sqrt{(\tilde{t} + \sigma_s^2) \left\|\Sigma_s^{-1}\right\|_{\text{op}}}$. As a result, there exists an event $E$ with $\mathbb{P}(E^c) \le C_t(M_t, c_t)$ such that

$$\|\beta^{s2t}\|_2 \le \|\beta_\star\|_2 + c_t.$$

Using a similar argument in (45), we derive the following on the same event $E_1$

$$\|\beta^{s2t} - \beta_\star\|_2 \le \|\beta_\star\|_2 + c_t.$$

This completes the proof.

$\square$

**Proposition 13.** *Let $g : \mathbb{R}^p \to \mathbb{R}$ be a function such that*

$$g(\beta^s) := \|\Sigma_t^{1/2}(\Sigma_t + \tau_t I)^{-1}\Sigma_t(\beta^s - \beta_\star)\|_2^2$$

*Then, on the same event $E(M_t, c_s)$ in Proposition 11, the function $g$ is $(\|\beta_\star\|_2 + c_s)\frac{2\lambda_1^3}{(\lambda_1 + \tau_t)^2}-$Lipschitz where $\lambda_1$ is the largest eigenvalue of $\Sigma_t$.*

*Proof.* We take the gradient of the function $g$:

$$\begin{aligned}
\|\nabla g(\beta^s)\|_2 &= 2\|\Sigma_t(\Sigma_t + \tau_t I)^{-1}\Sigma_t(\Sigma_t + \tau_t I)^{-1}\Sigma_t(\beta^s - \beta_\star)\|_2 \\
&\le \|\beta^s - \beta_\star\|_2 \max_i \frac{2\lambda_i^3}{(\lambda_i + \tau_t)^2} \\
&= \|\beta^s - \beta_\star\|_2 \max_i 2\lambda_i \left(1 - \frac{\tau_t}{\lambda_i + \tau_t}\right)^2 \\
&= \|\beta^s - \beta_\star\|_2 \frac{2\lambda_1^3}{(\lambda_1 + \tau_t)^2}.
\end{aligned}$$

Combining Proposition 11 on the event $E(M_t, c_s)$ with the above inequality provides the advertised claim. $\square$

**Proposition 14.** *Let $g : \mathbb{R}^p \to \mathbb{R}$ be a function such that*

$$g(\beta^s) := \frac{1}{p}\|\Sigma_t^{1/2}(\Sigma_t + \tau_t I)^{-1}\Sigma_t^{1/2}\gamma_t(\beta^s)\|_F^2$$

*Then, on the same event $E(M_t, c_s)$ in Proposition 11, the function $g$ is $L-$Lipschitz where $(\lambda_i)_{i=1}^p$ are the eigenvalues of $\Sigma_t$ with a descending order and*

$$L = \frac{4\tau_t^2}{m} \frac{\lambda_1^3}{(\lambda_1 + \tau_t)^4} \frac{\|\beta_\star\|_2 + c_s}{1 - \frac{1}{m}\sum_{i=1}^p \left(\frac{\lambda_i}{\lambda_i + \tau_t}\right)^2}.$$

*Proof.* We take the gradient of the function $g$:

$$\nabla g(\beta^s) = \frac{2}{p}\Sigma_t^{1/2}(\Sigma_t + \tau_t I)^{-1}\Sigma_t(\Sigma_t + \tau_t I)^{-1}\Sigma_t^{1/2}\nabla\gamma_t^2(\beta^s).$$

Note that

$$
\gamma_t^2(\boldsymbol{\beta}^s) = \kappa_s \left( \sigma_s^2 + \mathbb{E}_{\boldsymbol{g}_s}[\|\boldsymbol{\Sigma}_s^{1/2}(X_{\kappa_s,\sigma_s^2}^s(\boldsymbol{\Sigma}_s, \boldsymbol{\beta}_\star, \boldsymbol{g}_s) - \boldsymbol{\beta}_\star)\|_2^2] \right)
$$
$$
= \kappa_t \frac{\sigma_t^2 + \tau_t^2 \|(\boldsymbol{\Sigma}_t + \tau_t \boldsymbol{I})^{-1} \boldsymbol{\Sigma}_t^{1/2} \boldsymbol{\beta}^s\|_2^2}{1 - \frac{1}{m} \operatorname{tr}\left( (\boldsymbol{\Sigma}_t + \tau_t \boldsymbol{I})^{-2} \boldsymbol{\Sigma}_t^2 \right)}.
$$

Then, we have

$$
\nabla \gamma_t^2(\boldsymbol{\beta}^s) = 2\kappa_t \frac{\tau_t^2 \boldsymbol{\Sigma}_t^{1/2} (\boldsymbol{\Sigma}_t + \tau_t \boldsymbol{I})^{-2} \boldsymbol{\Sigma}_t^{1/2} \boldsymbol{\beta}^s}{1 - \frac{1}{m} \operatorname{tr}\left( (\boldsymbol{\Sigma}_t + \tau_t \boldsymbol{I})^{-2} \boldsymbol{\Sigma}_t^2 \right)}.
$$

Plugging $\nabla \gamma_t^2(\boldsymbol{\beta}^s)$ into $\nabla g(\boldsymbol{\beta}^s)$, we obtain that

$$
\|\nabla g(\boldsymbol{\beta}^s)\|_2 = \frac{4\tau_t^2}{m} \frac{\boldsymbol{\Sigma}_t^{1/2} (\boldsymbol{\Sigma}_t + \tau_t \boldsymbol{I})^{-1} \boldsymbol{\Sigma}_t (\boldsymbol{\Sigma}_t + \tau_t \boldsymbol{I})^{-2} \boldsymbol{\Sigma}_t (\boldsymbol{\Sigma}_t + \tau_t \boldsymbol{I})^{-1} \boldsymbol{\Sigma}_t^{1/2} \boldsymbol{\beta}^s}{1 - \frac{1}{m} \operatorname{tr}\left( (\boldsymbol{\Sigma}_t + \tau_t \boldsymbol{I})^{-2} \boldsymbol{\Sigma}_t^2 \right)}
$$
$$
\leq \frac{4\tau_t^2}{m} \frac{\lambda_1^3}{(\lambda_1 + \tau_t)^4} \frac{\|\boldsymbol{\beta}^s\|_2}{1 - \frac{1}{m} \sum_{i=1}^p \left( \frac{\lambda_i}{\lambda_i + \tau_t} \right)^2}.
$$

Combining Proposition 11 on the event $E(M_t, c_s)$ with the above inequality provides the advertised claim. □

**Proposition 15.** *Let* $g : \mathbb{R}^p \to \mathbb{R}$ *be a function such that*

$$
g(\boldsymbol{\beta}^s) := 2\boldsymbol{\beta}_\star^\top \left( \boldsymbol{I} - (\boldsymbol{\Sigma}_t + \tau_t \boldsymbol{I})^{-1} \boldsymbol{\Sigma}_t \right)^\top \boldsymbol{\Sigma}_t (\boldsymbol{\Sigma}_t + \tau_t \boldsymbol{I})^{-1} \boldsymbol{\Sigma}_t (\boldsymbol{\beta}^s - \boldsymbol{\beta}_\star).
$$

*Then, the function* $g$ *is* $2\|\boldsymbol{\beta}_\star\|_2 \tau_t \left( \frac{\lambda_1}{\lambda_1 + \tau_t} \right)^2$ *−Lipschitz where* $\lambda_1$ *is the largest eigenvalue of* $\boldsymbol{\Sigma}_t$.

*Proof.* We take the gradient of the function $g$:

$$
\|\nabla g(\boldsymbol{\beta}^s)\|_2 = 2\|\boldsymbol{\Sigma}_t (\boldsymbol{\Sigma}_t + \tau_t \boldsymbol{I})^{-1} \boldsymbol{\Sigma}_t \left( \boldsymbol{I} - (\boldsymbol{\Sigma}_t + \tau_t \boldsymbol{I})^{-1} \boldsymbol{\Sigma}_t \right) \boldsymbol{\beta}_\star\|_2
$$
$$
\leq 2\|\boldsymbol{\beta}_\star\|_2 \tau_t \max_i \left( 1 - \frac{\tau_t}{\lambda_i + \tau_t} \right)^2
$$
$$
= 2\|\boldsymbol{\beta}_\star\|_2 \tau_t \left( \frac{\lambda_1}{\lambda_1 + \tau_t} \right)^2,
$$

and the desired result readily follows. □

**Lemma 1.** *We have that*

$$
\mathbb{E}_{\boldsymbol{\beta}^s \sim X_{\kappa_s,\sigma_s^2}^s}[\gamma_t^2(\boldsymbol{\beta}^s)] = \kappa_t \frac{\sigma_t^2 + \tau_t^2 \|(\boldsymbol{\Sigma}_t + \tau_t \boldsymbol{I})^{-1} \boldsymbol{\Sigma}_t^{1/2}((\boldsymbol{\Sigma}_s + \tau_s \boldsymbol{I})^{-1} \boldsymbol{\Sigma}_s \boldsymbol{\beta}_\star\|_2^2}{1 - \frac{1}{n} \operatorname{tr}\left( (\boldsymbol{\Sigma}_t + \tau_t \boldsymbol{I})^{-2} \boldsymbol{\Sigma}_t^2 \right)}
$$
$$
+ \frac{\kappa_t \tau_t^2 \gamma_s^2(\boldsymbol{\beta}_\star)}{p} \frac{\operatorname{tr}\left( \boldsymbol{\Sigma}_s^{1/2} (\boldsymbol{\Sigma}_s + \tau_s \boldsymbol{I})^{-1} \boldsymbol{\Sigma}_t^{1/2} (\boldsymbol{\Sigma}_t + \tau_t \boldsymbol{I})^{-2} \boldsymbol{\Sigma}_t^{1/2} (\boldsymbol{\Sigma}_s + \tau_s \boldsymbol{I})^{-1} \boldsymbol{\Sigma}_s^{1/2} \right)}{1 - \frac{1}{n} \operatorname{tr}\left( (\boldsymbol{\Sigma}_t + \tau_t \boldsymbol{I})^{-2} \boldsymbol{\Sigma}_t^2 \right)}.
$$

*Proof.* The desired claim follows from the following manipulations using the definition of $X^s_{\kappa_s,\sigma_s^2}$ in (14):

$$
\mathbb{E}_{\beta^s \sim X^s_{\kappa_s,\sigma_s^2}}[\gamma_t^2(\beta^s)] = \mathbb{E}_{\beta^s \sim X^s_{\kappa_s,\sigma_s^2}}\left[\kappa_t \frac{\sigma_t^2 + \tau_t^2\|(\Sigma_t + \tau_t I)^{-1}\Sigma_t^{1/2}\beta^s\|_2^2}{1 - \frac{1}{n}\mathrm{tr}\left((\Sigma_t + \tau_t I)^{-2}\Sigma_t^2\right)}\right]
$$

$$
= \mathbb{E}_{g_s}\left[\kappa_t \frac{\sigma_t^2 + \tau_t^2\|(\Sigma_t + \tau_t I)^{-1}\Sigma_t^{1/2}\left((\Sigma_s + \tau_s I)^{-1}\Sigma_s\beta_\star + (\Sigma_s + \tau_s I)^{-1}\Sigma_s^{1/2}\gamma_s(\beta_\star)g_s/\sqrt{p}\right)\|_2^2}{1 - \frac{1}{n}\mathrm{tr}\left((\Sigma_t + \tau_t I)^{-2}\Sigma_t^2\right)}\right]
$$

$$
= \kappa_t \frac{\sigma_t^2 + \tau_t^2\|(\Sigma_t + \tau_t I)^{-1}\Sigma_t^{1/2}((\Sigma_s + \tau_s I)^{-1}\Sigma_s\beta_\star\|_2^2}{1 - \frac{1}{n}\mathrm{tr}\left((\Sigma_t + \tau_t I)^{-2}\Sigma_t^2\right)}
$$

$$
+ \frac{\kappa_t \tau_t^2 \gamma_s^2(\beta_\star)}{p} \frac{\mathrm{tr}\left(\Sigma_s^{1/2}(\Sigma_s + \tau_s I)^{-1}\Sigma_t^{1/2}(\Sigma_t + \tau_t I)^{-2}\Sigma_t^{1/2}(\Sigma_s + \tau_s I)^{-1}\Sigma_s^{1/2}\right)}{1 - \frac{1}{n}\mathrm{tr}\left((\Sigma_t + \tau_t I)^{-2}\Sigma_t^2\right)}.
$$

$\square$

## C.1 Analysis of Two-stage Model in Under-parametrized Region

**Proposition 16.** *Consider the two-stage model where the surrogate model is under-parameterized with $p_s \leq p$ features ($m < p_s$) and the target model is over-parameterized ($n > p$) with $p$ features. Then, the difference in the asymptotic risks of surrogate-to-target and standard target models is:*

$$
(\beta_\star)'^\top \theta_1^\top \Sigma_t \theta_1 (\beta_\star)' + \frac{\sigma_s^2}{\kappa_s^{-1} - 1}\mathrm{tr}\left(\theta_1^\top \Sigma_t \theta_1 \bar{\Sigma}_s^{-1}\right) + 2(\beta_\star)'^\top (I - \theta_1)^\top \Sigma_t \theta_1 (\beta_\star)'
$$

$$
+ \frac{\Omega}{1 - \Omega}\left(\frac{\sigma_s^2}{\kappa_s^{-1} - 1}\tau_t^2\mathrm{tr}\left((\Sigma_t + \tau_t I)^{-2}\Sigma_t\bar{\Sigma}_s^{-1}\right) - \tau_t^2\|(\Sigma_t + \tau_t I)^{-1}\Sigma_t^{1/2}(\beta_\star)'\|_2^2\right),
$$

*where $\theta_1 = (\Sigma_t + \tau_t I)^{-1}\Sigma_t$, $\Omega = \frac{1}{n}\mathrm{tr}\left(\Sigma_t^2(\Sigma_t + \tau_t I)^{-2}\right)$, and $(\beta_\star)' = \begin{bmatrix} 0 & 0 & \cdots & \beta_{p_s+1} & \beta_{p_s+2} & \cdots & \beta_p \end{bmatrix}^\top \in \mathbb{R}^p$.*

*Proof.* Let $\theta_2 := (\Sigma_t + \tau_t I)^{-1}\Sigma_t^{1/2}\frac{g_t}{\sqrt{p}}$ where $g_t \sim \mathcal{N}(0, I_p)$. Then, by Equation (10), the asymptotic risk estimate for the single-stage linear regression is given as:

$$
\bar{\mathcal{R}}^{s2t}_{\kappa_t,\sigma_t}(\Sigma_t, \beta_\star, \beta^s) := (\beta^s - \beta_\star)^\top \theta_1^\top \Sigma_t \theta_1 (\beta^s - \beta_\star) + \gamma_t^2(\beta^s)\mathbb{E}_{g_t}[\theta_2^\top \Sigma_t \theta_2]
$$
$$
+ \beta_\star^\top (I - \theta_1)^\top \Sigma_t (I - \theta_1)\beta_\star - 2\beta_\star^\top (I - \theta_1)^\top \Sigma_t \theta_1 (\beta^s - \beta_\star). \tag{51}
$$

For the ease of the analysis, define

$$
\bar{\Sigma}_s^{-1/2} = \begin{bmatrix} \Sigma_s^{-1/2} & 0_{p_s \times (p-p_s)} \\ 0_{(p-p_s)\times p_s} & 0_{(p-p_s)\times(p-p_s)} \end{bmatrix} \in \mathbb{R}^{p\times p}, \quad \bar{\beta}^s = \begin{bmatrix} \beta^s \\ 0_{p-p_s} \end{bmatrix} \in \mathbb{R}^p, \quad \bar{g}_s = \begin{bmatrix} g_s \\ 0_{p-p_s} \end{bmatrix} \in \mathbb{R}^p.
$$

Let $\bar{\beta}_\star = \beta_\star - (\beta_\star)'$ be obtained by setting the last $p - p_s$ entries of $\beta_\star$ to 0. Then, by Theorem 3 of Chang et al. (2021), since the surrogate model is under-parameterized, the following asymptotic estimate for the surrogate model obtained after this stage holds:

$$
\bar{\beta}^s = \bar{\beta}_\star + \sigma_s \frac{\bar{\Sigma}_s^{-1/2}\bar{g}_s}{\sqrt{p(\kappa_s^{-1} - 1)}},
$$

where $g_s \sim \mathcal{N}(0, I_p)$. Let $\dot{\kappa} = (\kappa_s, \kappa_t)$, $\dot{\Sigma} = (\Sigma_s, \Sigma_t)$, and $\dot{\sigma} = (\sigma_s^2, \sigma_t^2)$. Thus, plugging this surrogate parameter in the asymptotic risk estimate for the second stage gives:

$$
\bar{\mathcal{R}}_{\dot{\kappa},\dot{\sigma}}(\dot{\Sigma}, \beta_\star) = \mathbb{E}_{g_s}\left[\left((\beta_\star)' - \sigma_s\frac{\bar{\Sigma}_s^{-1/2}\bar{g}_s}{\sqrt{p(\kappa_s^{-1} - 1)}}\right)^\top \theta_1^\top \Sigma_t \theta_1 \left((\beta_\star)' - \sigma_s\frac{\bar{\Sigma}_s^{-1/2}\bar{g}_s}{\sqrt{p(\kappa_s^{-1} - 1)}}\right) + \gamma_t^2(\bar{\beta}^s)\frac{\mathrm{tr}\left(\Sigma_t^2(\Sigma_t + \tau_t I)^{-2}\right)}{p}\right]
$$

$$
+ \beta_\star^\top (I - \theta_1)^\top \Sigma_t (I - \theta_1)\beta_\star - \mathbb{E}_{g_s}\left[2\beta_\star^\top (I - \theta_1)^\top \Sigma_t \theta_1 \left(\sigma_s\frac{\bar{\Sigma}_s^{-1/2}\bar{g}_s}{\sqrt{p(\kappa_s^{-1} - 1)}} - (\beta_\star)'\right)\right],
$$

where

$$\gamma_t^2(\bar{\boldsymbol{\beta}}^s) = \kappa_t \frac{\sigma_t^2 + \tau_t^2 \|(\boldsymbol{\Sigma}_t + \tau_t \boldsymbol{I})^{-1}\boldsymbol{\Sigma}_t^{1/2}\bar{\boldsymbol{\beta}}^s\|_2^2}{1 - \frac{1}{n}\mathrm{tr}\left((\boldsymbol{\Sigma}_t + \tau_t \boldsymbol{I})^{-2}\boldsymbol{\Sigma}_t^2\right)} = \kappa_t \frac{\sigma_t^2 + \tau_t^2 \|(\boldsymbol{\Sigma}_t + \tau_t \boldsymbol{I})^{-1}\boldsymbol{\Sigma}_t^{1/2}(\bar{\boldsymbol{\beta}}_\star + \sigma_s \frac{\bar{\boldsymbol{\Sigma}}_s^{-1/2}\bar{\boldsymbol{g}}_s}{\sqrt{p(\kappa_s^{-1}-1)}}))\|_2^2}{1 - \Omega}.$$

Further modifications give:

$$\begin{aligned}
\bar{\mathcal{R}}_{\dot{\kappa},\dot{\sigma}}(\dot{\boldsymbol{\Sigma}},\boldsymbol{\beta}_\star) &= (\boldsymbol{\beta}_\star)'^\top \boldsymbol{\theta}_1^\top \boldsymbol{\Sigma}_t \boldsymbol{\theta}_1 (\boldsymbol{\beta}_\star)' + \frac{\sigma_s^2}{p(\kappa_s^{-1}-1)}\mathrm{tr}\left(\bar{\boldsymbol{\Sigma}}_s^{-1/2}\boldsymbol{\theta}_1^\top \boldsymbol{\Sigma}_t \boldsymbol{\theta}_1 \bar{\boldsymbol{\Sigma}}_s^{-1/2}\right) \\
&\quad + \frac{\Omega}{1-\Omega}\left(\sigma_t^2 + \mathbb{E}_{g_s}\left[\tau_t^2 \|(\boldsymbol{\Sigma}_t + \tau_t \boldsymbol{I})^{-1}\boldsymbol{\Sigma}_t^{1/2}(\bar{\boldsymbol{\beta}}_\star + \sigma_s \frac{\bar{\boldsymbol{\Sigma}}_s^{-1/2}\bar{\boldsymbol{g}}_s}{\sqrt{p(\kappa_s^{-1}-1)}})\|_2^2\right]\right) \\
&\quad + \boldsymbol{\beta}_\star^\top(\boldsymbol{I}-\boldsymbol{\theta}_1)^\top \boldsymbol{\Sigma}_t(\boldsymbol{I}-\boldsymbol{\theta}_1)\boldsymbol{\beta}_\star + 2\boldsymbol{\beta}_\star^\top(\boldsymbol{I}-\boldsymbol{\theta}_1)^\top \boldsymbol{\Sigma}_t \boldsymbol{\theta}_1 (\boldsymbol{\beta}_\star)' \\
&= (\boldsymbol{\beta}_\star)'^\top \boldsymbol{\theta}_1^\top \boldsymbol{\Sigma}_t \boldsymbol{\theta}_1 (\boldsymbol{\beta}_\star)' + \frac{\sigma_s^2}{p(\kappa_s^{-1}-1)}\mathrm{tr}\left(\boldsymbol{\theta}_1^\top \boldsymbol{\Sigma}_t \boldsymbol{\theta}_1 \bar{\boldsymbol{\Sigma}}_s^{-1}\right) \\
&\quad + \frac{\Omega}{1-\Omega}\left(\sigma_t^2 + \tau_t^2 \|(\boldsymbol{\Sigma}_t + \tau_t \boldsymbol{I})^{-1}\boldsymbol{\Sigma}_t^{1/2}\bar{\boldsymbol{\beta}}_\star\|_2^2 + \frac{\sigma_s^2}{p(\kappa_s^{-1}-1)}\tau_t^2\mathrm{tr}\left((\boldsymbol{\Sigma}_t + \tau_t \boldsymbol{I})^{-2}\boldsymbol{\Sigma}_t \bar{\boldsymbol{\Sigma}}_s^{-1}\right)\right) \\
&\quad + \boldsymbol{\beta}_\star^\top(\boldsymbol{I}-\boldsymbol{\theta}_1)^\top \boldsymbol{\Sigma}_t(\boldsymbol{I}-\boldsymbol{\theta}_1)\boldsymbol{\beta}_\star + 2(\boldsymbol{\beta}_\star)'^\top(\boldsymbol{I}-\boldsymbol{\theta}_1)^\top \boldsymbol{\Sigma}_t \boldsymbol{\theta}_1 (\boldsymbol{\beta}_\star)'.
\end{aligned}$$

Then, using the fact that since $(\boldsymbol{\beta}_\star)'$ is orthogonal to $\bar{\boldsymbol{\beta}}_\star$ (thus, their supports do not overlap), it follows that the difference between this surrogate-to-target risk and the standard target model risk is:

$$\begin{aligned}
\textbf{Difference} &= (\boldsymbol{\beta}_\star)'^\top \boldsymbol{\theta}_1^\top \boldsymbol{\Sigma}_t \boldsymbol{\theta}_1 (\boldsymbol{\beta}_\star)' + \frac{\sigma_s^2}{p(\kappa_s^{-1}-1)}\mathrm{tr}\left(\boldsymbol{\theta}_1^\top \boldsymbol{\Sigma}_t \boldsymbol{\theta}_1 \bar{\boldsymbol{\Sigma}}_s^{-1}\right) + 2(\boldsymbol{\beta}_\star)'^\top(\boldsymbol{I}-\boldsymbol{\theta}_1)^\top \boldsymbol{\Sigma}_t \boldsymbol{\theta}_1 (\boldsymbol{\beta}_\star)' \\
&\quad + \frac{\Omega}{1-\Omega}\left(\frac{\sigma_s^2}{p(\kappa_s^{-1}-1)}\tau_t^2\mathrm{tr}\left((\boldsymbol{\Sigma}_t + \tau_t \boldsymbol{I})^{-2}\boldsymbol{\Sigma}_t \bar{\boldsymbol{\Sigma}}_s^{-1}\right) - \tau_t^2 \|(\boldsymbol{\Sigma}_t + \tau_t \boldsymbol{I})^{-1}\boldsymbol{\Sigma}_t^{1/2}(\boldsymbol{\beta}_\star)'\|_2^2\right).
\end{aligned}$$

(52)

This completes the proof. $\qquad\square$

**Proposition 17.** *Consider the two-stage model where the surrogate model is under-parameterized with $p$ features ($m < p$) and the second stage is over-parameterized ($n > p$) with $p$ features. Under this setting, the surrogate-to-target model cannot outperform the standard target model in terms of asymptotic risk.*

*Proof.* Let $\boldsymbol{\theta}_1 := (\boldsymbol{\Sigma}_t + \tau_t \boldsymbol{I})^{-1}\boldsymbol{\Sigma}_t$ and $\boldsymbol{\theta}_2 := (\boldsymbol{\Sigma}_t + \tau_t \boldsymbol{I})^{-1}\boldsymbol{\Sigma}_t^{1/2}\frac{\boldsymbol{g}_t}{\sqrt{p}}$ where $\boldsymbol{g}_t \sim \mathcal{N}(\boldsymbol{0}, \boldsymbol{I}_p)$. Then, by Equation (10), the asymptotic risk estimate for the single-stage linear regression is given as:

$$\begin{aligned}
\bar{\mathcal{R}}_{\kappa_t,\sigma_t}^{s2t}(\boldsymbol{\Sigma}_t,\boldsymbol{\beta}_\star,\boldsymbol{\beta}^s) &:= (\boldsymbol{\beta}^s - \boldsymbol{\beta}_\star)^\top \boldsymbol{\theta}_1^\top \boldsymbol{\Sigma}_t \boldsymbol{\theta}_1 (\boldsymbol{\beta}^s - \boldsymbol{\beta}_\star) + \gamma_t^2(\boldsymbol{\beta}^s)\mathbb{E}_{g_t}[\boldsymbol{\theta}_2^\top \boldsymbol{\Sigma}_t \boldsymbol{\theta}_2] \\
&\quad + \boldsymbol{\beta}_\star^\top(\boldsymbol{I}-\boldsymbol{\theta}_1)^\top \boldsymbol{\Sigma}_t(\boldsymbol{I}-\boldsymbol{\theta}_1)\boldsymbol{\beta}_\star - 2\boldsymbol{\beta}_\star^\top(\boldsymbol{I}-\boldsymbol{\theta}_1)^\top \boldsymbol{\Sigma}_t \boldsymbol{\theta}_1(\boldsymbol{\beta}^s - \boldsymbol{\beta}_\star).
\end{aligned}$$

(53)

When the surrogate model is under-parameterized, we can use the following asymptotic estimate for the surrogate model obtained after this stage by Theorem 3 of Chang et al. (2021):

$$\boldsymbol{\beta}^s = \boldsymbol{\beta}_\star + \sigma_s \frac{\boldsymbol{\Sigma}_s^{-1/2}\boldsymbol{g}_s}{\sqrt{p(\kappa_s^{-1}-1)}},$$

where $\boldsymbol{g}_s \sim \mathcal{N}(\boldsymbol{0}, \boldsymbol{I}_p)$. Let $\dot{\kappa} = (\kappa_s, \kappa_t)$, $\dot{\boldsymbol{\Sigma}} = (\boldsymbol{\Sigma}_s, \boldsymbol{\Sigma}_t)$, and $\dot{\sigma} = (\sigma_s^2, \sigma_t^2)$. Then, by plugging this in Equation (53), the asymptotic risk estimate is the following:

$$\begin{aligned}
\bar{\mathcal{R}}_{\dot{\kappa},\dot{\sigma}}(\dot{\boldsymbol{\Sigma}},\boldsymbol{\beta}_\star) &= \mathbb{E}_{g_s}\left[\frac{\sigma_s^2}{p(\kappa_s^{-1}-1)}\boldsymbol{g}_s^\top \boldsymbol{\Sigma}_s^{-1/2}\boldsymbol{\theta}_1^\top \boldsymbol{\Sigma}_t \boldsymbol{\theta}_1 \boldsymbol{\Sigma}_s^{-1/2}\boldsymbol{g}_s + \gamma_t^2(\boldsymbol{\beta}^s)\frac{\mathrm{tr}\left(\boldsymbol{\Sigma}_t^2(\boldsymbol{\Sigma}_t + \tau_t \boldsymbol{I})^{-2}\right)}{p}\right] \\
&\quad + \boldsymbol{\beta}_\star^\top(\boldsymbol{I}-\boldsymbol{\theta}_1)^\top \boldsymbol{\Sigma}_t(\boldsymbol{I}-\boldsymbol{\theta}_1)\boldsymbol{\beta}_\star - \mathbb{E}_{g_s}\left[2\frac{\sigma_s}{\sqrt{p(\kappa_s^{-1}-1)}}\boldsymbol{\beta}_\star^\top(\boldsymbol{I}-\boldsymbol{\theta}_1)^\top \boldsymbol{\Sigma}_t \boldsymbol{\theta}_1 \boldsymbol{\Sigma}_s^{-1/2}\boldsymbol{g}_s\right],
\end{aligned}$$

where

$$\gamma_t^2(\boldsymbol{\beta}^s) = \kappa_t \frac{\sigma_t^2 + \tau_t^2 \|(\boldsymbol{\Sigma}_t + \tau_t \boldsymbol{I})^{-1} \boldsymbol{\Sigma}_t^{1/2} \boldsymbol{\beta}^s\|_2^2}{1 - \frac{1}{n} \text{tr}\left((\boldsymbol{\Sigma}_t + \tau_t \boldsymbol{I})^{-2} \boldsymbol{\Sigma}_t^2\right)} = \kappa_t \frac{\sigma_t^2 + \tau_t^2 \|(\boldsymbol{\Sigma}_t + \tau_t \boldsymbol{I})^{-1} \boldsymbol{\Sigma}_t^{1/2}(\boldsymbol{\beta}_\star + \sigma_s \frac{\boldsymbol{\Sigma}_s^{-1/2} \boldsymbol{g}_s}{\sqrt{p(\kappa_s^{-1}-1)}})\|_2^2}{1 - \Omega}.$$

Further simplifications give:

$$\begin{aligned}
\bar{\mathcal{R}}_{k,\acute{\sigma}}(\dot{\boldsymbol{\Sigma}}, \boldsymbol{\beta}_\star) &= \frac{\sigma_s^2}{p(\kappa_s^{-1}-1)} \text{tr}\left(\boldsymbol{\Sigma}_s^{-1/2} \boldsymbol{\theta}_1^\top \boldsymbol{\Sigma}_t \boldsymbol{\theta}_1 \boldsymbol{\Sigma}_s^{-1/2}\right) + \frac{\Omega}{1-\Omega}\left(\sigma_t^2 + \mathbb{E}_{\boldsymbol{g}_s}\left[\tau_t^2 \|(\boldsymbol{\Sigma}_t + \tau_t \boldsymbol{I})^{-1} \boldsymbol{\Sigma}_t^{1/2}(\boldsymbol{\beta}_\star + \sigma_s \frac{\boldsymbol{\Sigma}_s^{-1/2} \boldsymbol{g}_s}{\sqrt{p(\kappa_s^{-1}-1)}})\|_2^2\right]\right) \\
&\quad + \boldsymbol{\beta}_\star^\top (\boldsymbol{I} - \boldsymbol{\theta}_1)^\top \boldsymbol{\Sigma}_t (\boldsymbol{I} - \boldsymbol{\theta}_1) \boldsymbol{\beta}_\star \\
&= \frac{\sigma_s^2}{p(\kappa_s^{-1}-1)} \text{tr}\left(\boldsymbol{\Sigma}_s^{-1/2} \boldsymbol{\theta}_1^\top \boldsymbol{\Sigma}_t \boldsymbol{\theta}_1 \boldsymbol{\Sigma}_s^{-1/2}\right) + \frac{\Omega}{1-\Omega}\left(\sigma_t^2 + \sum_{i=1}^p \lambda_{t,i} \zeta_{t,i}^2 \left(\beta_i^2 + \frac{\sigma_s^2 \lambda_{s,i}^{-1}}{p(\kappa_s^{-1}-1)}\right)\right) + \sum_{i=1}^p \lambda_{t,i} \zeta_{t,i}^2 \beta_i^2 \\
&> \frac{\sigma_t^2 \Omega + \sum_{i=1}^p \lambda_{t,i} \zeta_{t,i}^2 \beta_i^2}{1-\Omega},
\end{aligned}$$

which is the asymptotic risk estimate of the standard target model (follows from Definition 2 or by simply plugging $\boldsymbol{\beta}^s = \boldsymbol{\beta}_\star$ in Equation (53)). Hence, we conclude that improvement over the excess test risk is not possible when the first stage is under-parameterized with $p$-dimensions. $\qquad\square$

**Corollary 3.** *Consider the setting of Proposition 3. Let $\boldsymbol{\Sigma}_t$ be the covariance matrix of the target model and let $\mathcal{M}$ be the mask such that the surrogate-to-target model with $\boldsymbol{\Sigma}_t$ and $\mathcal{M}$ outperforms the standard target model in the asymptotic risk. Let $\boldsymbol{\Sigma}_s = \mathbb{E}[\mathcal{M}(\boldsymbol{x})\mathcal{M}(\boldsymbol{x})^\top]$ where $\boldsymbol{x} \sim \mathcal{N}(\boldsymbol{0}, \boldsymbol{\Sigma}_t)$. Then, for any $\boldsymbol{\beta}_\star \in \mathbb{R}^p$, there exists $K \in \mathbb{N}$ such that the surrogate-to-target model with two stages given $\boldsymbol{\Sigma}_t, \boldsymbol{\Sigma}_s, \boldsymbol{\beta}_\star$, and $m > K$ outperforms the standard target model.*

*Proof.* We make the following three observations. First, the expression in (52) is continuous with respect to $m \in \mathbb{R}$ when $m > p$. Note that $m$ is a natural number in our setting, but we consider the expression in (52) as a function of $m$ where $m \in \mathbb{R}$. Second, the expression in (52) is monotone with respect to $m$ since $\text{tr}\left(\boldsymbol{\theta}_1^\top \boldsymbol{\Sigma}_t \boldsymbol{\theta}_1 \bar{\boldsymbol{\Sigma}}_s^{-1}\right) \geq 0$ and $\frac{\Omega}{1-\Omega} \tau_t^2 \text{tr}\left((\boldsymbol{\Sigma}_t + \tau_t \boldsymbol{I})^{-2} \boldsymbol{\Sigma}_t \bar{\boldsymbol{\Sigma}}_s^{-1}\right) \geq 0$. Third, by definition, we know that the standard target model with $\boldsymbol{\Sigma}_t$ and $\mathcal{M}$ outperforms the standard target model in the asymptotic setting. This means that the standard target model with two stages given $\boldsymbol{\Sigma}_t, \boldsymbol{\Sigma}_s$, and $\boldsymbol{\beta}_\star$ while $m$ approaches $\infty$ outperforms the standard target model. By combining these three observations, the desired result follows. $\qquad\square$

