## A  Proofs for Section 3

**Observation 1.** *The model shift in the surrogate-to-target model is equivalent to the covariance shift model (Mallinar et al., 2024). Formally, given $\boldsymbol{\beta}_\star \in \mathbb{R}^p$ and the covariance matrix $\boldsymbol{\Sigma}_t \in \mathbb{R}^{p \times p}$, there exists a unique $\boldsymbol{\beta}^s \in \mathbb{R}^p$ such that the risk of the surrogate-to-target problem $\mathcal{R}(\boldsymbol{\beta}^{s2t})$ with $(\boldsymbol{\beta}_\star, \boldsymbol{\beta}^s, \boldsymbol{\Sigma}_t)$ is equivalent to the risk of the covariance shift model $\mathcal{R}^{cs}(\hat{\boldsymbol{\beta}})$ with $(\boldsymbol{\beta}_\star, \boldsymbol{\Sigma}_s, \boldsymbol{\Sigma}_t)$ for any $\boldsymbol{\Sigma}_s \in \mathbb{R}^{p \times p}$ that is jointly diagonalizable with $\boldsymbol{\Sigma}_t$.*

*Proof.* By Observation 2, we assume that $\boldsymbol{\Sigma}_t$ and $\boldsymbol{\Sigma}_s$ are diagonal matrices. As $\boldsymbol{\Sigma}_t$ and $\boldsymbol{\Sigma}_s$ are jointly diagonalizable, there exists a unique diagonal matrix $A \in \mathbb{R}^{p \times p}$ such that

$$\boldsymbol{\Sigma}_s = A^\top \boldsymbol{\Sigma}_t A.$$

Then, consider the model shift discussed in Section 3. Take the case where $\boldsymbol{\beta}^s = A\boldsymbol{\beta}_\star$ and labels are generated as $y = \boldsymbol{x}^\top \boldsymbol{\beta}^s + z$, where $\boldsymbol{x} \sim \mathcal{N}(0, \boldsymbol{\Sigma}_t)$ and $z \sim \mathcal{N}(0, \sigma_t^2)$. This is equivalent to the case where $y = (\boldsymbol{x}^\top A)\boldsymbol{\beta}_\star + z = \bar{\boldsymbol{x}}^\top \boldsymbol{\beta}_\star + z$ such that $\boldsymbol{x} \sim \mathcal{N}(0, \boldsymbol{\Sigma}_s)$ and $z \sim \mathcal{N}(0, \sigma_t^2)$. Note that *(i)* the transformed inputs and the labels are identical in both scenarios, and *(ii)* the estimators are computed in the same way. Thus, it follows that the risks $\mathcal{R}(\boldsymbol{\beta}^{s2t})$ and $\mathcal{R}^{cs}(\hat{\boldsymbol{\beta}})$ are equivalent. The other way follows from an almost identical argument. □

**Observation 2.** *For any covariance matrix $\boldsymbol{\Sigma} \in \mathbb{R}^{p \times p}$, there exists an orthonormal matrix $U \in \mathbb{R}^{p \times p}$ such that the transformation of $\boldsymbol{x} \to U^\top \boldsymbol{x}$ and $\boldsymbol{\beta} \to U^\top \boldsymbol{\beta}$ does not affect the labels $\boldsymbol{y}$ but ensures that the covariance matrix is diagonal.*

*Proof.* Since the covariance matrix $\boldsymbol{\Sigma}$ is PSD, its unit-norm eigenvectors are orthogonal. Consider the matrix $U$ whose columns are the eigenvectors of $\boldsymbol{\Sigma}$. Then, $\boldsymbol{\Sigma}$ can be expressed as $\boldsymbol{\Sigma} = U\boldsymbol{\Lambda}U^\top$, where $\boldsymbol{\

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

}^{s2t}\right)^2\right] - \sigma_t^2 = \mathbb{E}\left[\left(\boldsymbol{\beta}^{s2t} - \boldsymbol{\beta}_\star\right)^\top \boldsymbol{\Sigma}\left(\boldsymbol{\beta}^{s2t} - \boldsymbol{\beta}_\star\right)\right].$$

Now, consider the zero-padded vector $\bar{\boldsymbol{\beta}}^s = \begin{bmatrix} \boldsymbol{\beta}^s \\ \boldsymbol{0}_{p-p_s} \end{bmatrix} \in \mathbb{R}^p$, and define $(\bar{\boldsymbol{\beta}}^s)' = \boldsymbol{\beta}_\star - \bar{\boldsymbol{\beta}}^s \in \mathbb{R}^p$ of which the first $p_s$ dimensions are zero. In this way, we can consider the labels in the second training phase as $y^s = \boldsymbol{x}^\top \bar{\boldsymbol{\beta}}^s + z$, where $z \sim \mathcal{N}(0, \sigma_t^2)$. Applying the test risk estimate in Definition 2, we obtain:

$$\mathbb{E}\left[\left(y^s - \boldsymbol{x}^\top \boldsymbol{\beta}^{s2t}\right)^2\right] = \mathbb{E}\left[\left(\boldsymbol{\beta}^{s2t} - \boldsymbol{\beta}^s\right)^\top \boldsymbol{\Sigma}\left(\boldsymbol{\beta}^{s2t} - \boldsymbol{\beta}^s\right)\right] = \frac{\mathcal{B}(\bar{\boldsymbol{\beta}}^s) + \sigma_t^2 \Omega}{1 - \Omega}.$$

We further derive

$$\begin{aligned}
\mathbb{E}\left[\left(y - \boldsymbol{x}^\top \boldsymbol{\beta}^{s2t}\right)^2\right] &= \mathbb{E}\left[\left(y^s - \boldsymbol{x}^\top \boldsymbol{\beta}^{s2t} + \boldsymbol{x}^\top (\bar{\boldsymbol{\beta}}^s)'\right)^2\right] \\
&= \mathbb{E}\left[\left(y^s - \boldsymbol{x}^\top \boldsymbol{\beta}^{s2t}\right)^2\right] - 2\mathbb{E}\left[\left(y^s - \boldsymbol{x}^\top \boldsymbol{\beta}^{s2t}\right)\left(\boldsymbol{x}^\top (\bar{\boldsymbol{\beta}}^s)'\right)\right] + \mathbb{E}\left[\left(\boldsymbol{x}^\top (\bar{\boldsymbol{\beta}}^s)'\right)^2\right] \\
&\overset{(a)}{=} \mathbb{E}\left[\left(y^s - \boldsymbol{x}^\top \boldsymbol{\beta}^{s2t}\right)^2\right] - 2\mathbb{E}\left[y^s - \boldsymbol{x}^\top \boldsymbol{\beta}^{s2t}\right] \underbrace{\mathbb{E}\left[\boldsymbol{x}^\top (\bar{\boldsymbol{\beta}}^s)'\right]}_{=0} + \mathbb{E}\left[\left(\boldsymbol{x}^\top (\bar{\boldsymbol{\beta}}^s)'\right)^2\right] \\
&= \mathbb{E}\left[\left(y^s - \boldsymbol{x}^\top \boldsymbol{\beta}^{s2t}\right)^2\right] + \mathbb{E}\left[\left(\boldsymbol{x}^\top (\bar{\boldsymbol{\beta}}^s)'\right)^2\right] \\
&= \mathbb{E}\left[\left(y^s - \boldsymbol{x}^\top \boldsymbol{\beta}^{s2t}\right)^2\right] + \sum_{i=p_s+1}^p \lambda_i \beta_i^2 \\
&= \frac{\mathcal{B}(\bar{\boldsymbol{\beta}}^s) + \sigma_t^2 \Omega}{1 - \Omega} + \sum_{i=p_s+1}^p \lambda_i \beta_i^2.
\end{aligned} \tag{16}$$

where in the above equality $(a)$ follows from the fact that the components $x_i$ are independent as the covariance matrix is diagonal. Thus, the risk difference between the target and surrogate-to-target models is

$$\begin{aligned}
\mathcal{R}(\boldsymbol{\beta}^t) - \mathcal{R}(\boldsymbol{\beta}^{s2t}) &= \frac{\mathcal{B}(\boldsymbol{\beta}_\star) - \mathcal{B}(\bar{\boldsymbol{\beta}}^s)}{1 - \Omega} - \sum_{i=p_s+1}^p \lambda_i \beta_i^2 \\
&= \frac{\sum_{i=p_s+1}^p \lambda_i \zeta_i^2 \beta_i^2}{1 - \Omega} - \sum_{i=p_s+1}^p \lambda_i \beta_i^2.
\end{aligned}$$

We observe that each dimension's contribution to the excess test risk can be analyzed individually. Therefore, if

$$\zeta_i^2 > 1 - \Omega, \tag{17}$$

excluding feature $i$ in the feature selection reduces the overall risk $\mathcal{R}(\boldsymbol{\beta}^{s2t})$. Along the same lines, the projection $\mathcal{M}$ that selects all the features $i$ that satisfy $\zeta_i^2 < 1 - \Omega$ minimizes the asymptotic excess test risk. □

## B  Proofs for Section 4

**Definition 2** (Omniscient test risk estimate). *Fix $p > n \geq 1$. Given a covariance $\boldsymbol{\Sigma} = \boldsymbol{U}\operatorname{diag}(\boldsymbol{\lambda})\boldsymbol{U}^\top$, $\boldsymbol{\beta}_\star$, and the noise term $\sigma$, set $\bar{\boldsymbol{\beta}} = \boldsymbol{U}^\top \boldsymbol{\beta}_\star$ and define $\tau \in \mathbb{R}$ as the unique non-negative solution of $n = \sum_{i=1}^p \frac{\lambda_i}{\lambda_i + \tau}$. Then, the excess test risk estimate is the following:*

$$\mathcal{R}(\hat{\boldsymbol{\beta}}) \approx \mathbb{E}_{\hat{\boldsymbol{\beta}} \sim D(\boldsymbol{\beta}_\star)}\left[(y - \boldsymbol{x}^\top \hat{\boldsymbol{\beta}})^2\right] - \sigma^2 = \frac{\sigma^2 \Omega + \mathcal{B}(\bar{\boldsymbol{\beta}})}{1 - \Omega}, \tag{11}$$

$$where \quad \zeta_i = \frac{\tau}{\lambda_i + \tau}, \quad \Omega = \frac{1}{n}\sum_{i=1}^p (1 - \zeta_i)^2, \quad \mathcal{B}(\bar{\boldsymbol{\beta}}) = \sum_{i=1}^p \lambda_i \zeta_i^2 \bar{\beta}_i^2.$$

In the following proof, we suppose that the empirical distributions of $\bar{\boldsymbol{\beta}}$ and $\boldsymbol{\lambda}$ converge as $p \to \infty$ having fixed the ratio $p/n = \kappa$. Then, we will prove that the omniscient risk converges to the asymptotic risk defined in (9).

*Proof for the proportional asymptotic case.* Using Theorem 2.3 of Han & Xu (2023), we can estimate $\hat{\boldsymbol{\beta}}$ as follows:

$$
\hat{\boldsymbol{\beta}} = (\boldsymbol{\Sigma} + \tau \boldsymbol{I})^{-1} \boldsymbol{\Sigma} \left( \boldsymbol{\beta}_\star + \frac{\boldsymbol{\Sigma}^{-1/2} \gamma(\boldsymbol{\beta}_\star) \boldsymbol{g}}{\sqrt{p}} \right),
$$

where

$$
\boldsymbol{g} \sim \mathcal{N}(0, \boldsymbol{I}_p), \quad \gamma(\boldsymbol{\beta}_\star)^2 = \kappa \frac{\sigma + \tau^2 \|(\boldsymbol{\Sigma} + \tau \boldsymbol{I})^{-1} \boldsymbol{\Sigma}^{1/2} \boldsymbol{\beta}_\star\|_2^2}{1 - \frac{1}{n} \mathrm{tr}\left((\boldsymbol{\Sigma} + \tau \boldsymbol{I})^{-2} \boldsymbol{\Sigma}^2\right)}, \quad \tau \text{ is the solution to } n = \sum_{i=1}^{p} \frac{\lambda_i}{\lambda_i + \tau}.
$$

Let

$$
\boldsymbol{X}_1 = (\boldsymbol{\Sigma} + \tau \boldsymbol{I})^{-1} \boldsymbol{\Sigma} \quad , \quad \boldsymbol{X}_2 = \frac{(\boldsymbol{\Sigma} + \tau \boldsymbol{I})^{-1} \boldsymbol{\Sigma}^{1/2} \gamma(\boldsymbol{\beta}_\star)}{\sqrt{p}}.
$$

Using this estimate, we can calculate the excess test risk as

$$
\begin{aligned}
\mathcal{R}(\hat{\boldsymbol{\beta}}) &= \mathbb{E}\left[ ((\boldsymbol{X}_1 - \boldsymbol{I})\boldsymbol{\beta}_\star + \boldsymbol{X}_2 \boldsymbol{g})^\top \boldsymbol{\Sigma} ((\boldsymbol{X}_1 - \boldsymbol{I})\boldsymbol{\beta}_\star + \boldsymbol{X}_2 \boldsymbol{g}) \right] \\
&= \boldsymbol{\beta}_\star^\top (\boldsymbol{X}_1 - \boldsymbol{I})^\top \boldsymbol{\Sigma} (\boldsymbol{X}_1 - \boldsymbol{I}) \boldsymbol{\beta}_\star + \mathbb{E}\left[ \boldsymbol{g}^\top \boldsymbol{X}_2^\top \boldsymbol{\Sigma} \boldsymbol{X}_2 \boldsymbol{g} \right] \\
&= \boldsymbol{\beta}_\star^\top (\boldsymbol{X}_1 - \boldsymbol{I})^\top \boldsymbol{\Sigma} (\boldsymbol{X}_1 - \boldsymbol{I}) \boldsymbol{\beta}_\star + \mathrm{tr}\left( \boldsymbol{X}_2^\top \boldsymbol{\Sigma} \boldsymbol{X}_2 \right).
\end{aligned} \tag{18}
$$

Then by recalling the eigendecomposition for the covariance matrix $\boldsymbol{\Sigma} = \boldsymbol{U} \boldsymbol{\Lambda} \boldsymbol{U}^\top$, we have

$$
\begin{aligned}
\boldsymbol{X}_1 &= (\boldsymbol{U} \boldsymbol{\Lambda} \boldsymbol{U}^\top + \tau \boldsymbol{U} \boldsymbol{U}^\top)^{-1} \boldsymbol{U} \boldsymbol{\Lambda} \boldsymbol{U}^\top \\
&= \boldsymbol{U} (\boldsymbol{\Lambda} + \tau \boldsymbol{I})^{-1} \boldsymbol{U}^\top \boldsymbol{U} \boldsymbol{\Lambda} \boldsymbol{U}^\top \\
&= \boldsymbol{U} \operatorname{diag}\left( \frac{\boldsymbol{\lambda}}{\boldsymbol{\lambda} + \tau} \right) \boldsymbol{U}^\top.
\end{aligned}
$$

Using the diagonalization of $\boldsymbol{I}$, $\boldsymbol{X}_1 - \boldsymbol{I}$ can now be computed as

$$
\boldsymbol{X}_1 - \boldsymbol{I} = \boldsymbol{U} \operatorname{diag}\left( \frac{-\tau}{\boldsymbol{\lambda} + \tau} \right) \boldsymbol{U}^\top.
$$

Let's now compute

$$
\begin{aligned}
\boldsymbol{\beta}_\star^\top (\boldsymbol{X}_1 - \boldsymbol{I})^\top \boldsymbol{\Sigma} (\boldsymbol{X}_1 - \boldsymbol{I}) \boldsymbol{\beta}_\star &= \boldsymbol{\beta}_\star^\top \boldsymbol{U} \operatorname{diag}\left( \frac{-\tau}{\boldsymbol{\lambda} + \tau} \right) \boldsymbol{U}^\top \boldsymbol{U} \boldsymbol{\Lambda} \boldsymbol{U}^\top \boldsymbol{U} \operatorname{diag}\left( \frac{-\tau}{\boldsymbol{\lambda} + \tau} \right) \boldsymbol{U}^\top \boldsymbol{\beta}_\star \\
&= \boldsymbol{\beta}_\star^\top \boldsymbol{U} \operatorname{diag}\left( \frac{\boldsymbol{\lambda} \tau^2}{(\boldsymbol{\lambda} + \tau)^2} \right) \boldsymbol{U}^\top \boldsymbol{\beta}_\star.
\end{aligned}
$$

As $\bar{\boldsymbol{\beta}} = \boldsymbol{U}^\top \boldsymbol{\beta}_\star$, we obtain that the RHS of the previous expression equals

$$
\sum_{i=1}^{p} \frac{\lambda_i \tau^2 \bar{\beta}_i^2}{(\lambda_i + \tau)^2} = \mathcal{B}(\bar{\boldsymbol{\beta}}).
$$

Next, we write more compactly the terms $\mathrm{tr}\left( \boldsymbol{X}_2^\top \boldsymbol{\Sigma} \boldsymbol{X}_2 \right)$ and $\gamma(\boldsymbol{\beta}_\star)^2$. By defining the short-hand notation $\Omega = \frac{1}{n} \mathrm{tr}\left( (\boldsymbol{\Sigma} + \tau \boldsymbol{I})^{-2} \boldsymbol{\Sigma}^2 \right) = \frac{1}{n} \sum_{i=1}^{p} (1 - \zeta_i)^2$, we have

$$
\mathrm{tr}\left( \boldsymbol{X}_2^\top \boldsymbol{\Sigma} \boldsymbol{X}_2 \right) = \frac{\gamma(\boldsymbol{\beta}_\star)^2}{p} \sum_{i=1}^{p} \left( \frac{\lambda_i}{\lambda_i + \tau} \right)^2 = \frac{\gamma(\boldsymbol{\beta}_\star)^2 n \Omega}{p}
$$

$$
\gamma(\boldsymbol{\beta}_\star)^2 = \kappa \frac{\sigma^2 + \tau^2 \|(\boldsymbol{\Sigma} + \tau \boldsymbol{I})^{-1} \boldsymbol{\

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

$$= n^{-\beta+1} \frac{k_1}{k_2} \left((\omega_n/n)^{-\beta+1} - (\phi_n/n)^{-\beta+1}\right)$$

$$= \Theta(n^{-\beta+1}).$$

Recalling $\omega_n/n = \Theta(1)$ and $\phi_n/n = \Theta(1)$, we obtain that $\mathcal{R}^*(\boldsymbol{\beta}^{s2t}) = \Omega(n^{-\beta+1})$, and thus,

$$\mathcal{R}(\boldsymbol{\beta}^{s2t}) \geq \mathcal{R}^*(\boldsymbol{\beta}^{s2t}) \implies \mathcal{R}(\boldsymbol{\beta}^{s2t}) = \Omega(n^{-\beta+1}).$$

**Case 2:** $\beta > 2\alpha + 1$

In this case, we have

$$\mathcal{R}^*(\boldsymbol{\beta}^{s2t}) \geq \sum_{i=1}^{p} \lambda_i \beta_i^2 \frac{(1-\zeta_i)^2 \zeta_i^2}{\left((1-\zeta_i)^2 + \frac{\Omega}{1-\Omega}\zeta_i^2\right)^2} = \sum_{i=1}^{p} \lambda_i \beta_i^2 \zeta_i^2 \frac{(1-\zeta_i)^2}{\left((1-\zeta_i)^2 + \frac{\Omega}{1-\Omega}\zeta_i^2\right)^2}$$

$$\geq \sum_{i:\zeta_i < 1-\Omega} \lambda_i \zeta_i^2 \beta_i^2 \frac{\Omega^2}{\left(1 + \frac{\Omega}{1-\Omega}\right)^2} = \sum_{i=1}^{\phi_n} \frac{i^{-\beta}}{(1 + \frac{1}{\tau_t} i^{-\alpha})^2} k_3,$$

where $k_3 = \frac{\Omega^2}{\left(1+\frac{\Omega}{1-\Omega}\right)^2} = \Theta(1)$. From Case 2 in Proposition 6, we already know that the same summation – with upper bound $\omega_n$ rather than $\phi_n$ – scales as $\Theta(n^{-2\alpha})$. Yet, since $\phi_n$ and $\omega_n$ have the same order $\Theta(n)$, the result remains. This suggests $\mathcal{R}^*(\boldsymbol{\beta}^{s2t}) = \Omega(n^{-2\alpha})$, which eventually yields

$$\mathcal{R}(\boldsymbol{\beta}^{s2t}) \geq \mathcal{R}^*(\boldsymbol{\beta}^{s2t}) \implies \mathcal{R}(\boldsymbol{\beta}^{s2t}) = \Omega(n^{-2\alpha}).$$

Hence, this allows us to say that the scaling law doesn't improve even with the freedom to choose any $\boldsymbol{\beta}^s$. $\quad\square$

**Proposition 7** (Non-asymptotic analysis of $\tau$). *Suppose that* $\Sigma \in \mathbb{R}^{p \times p}$ *is diagonal and* $\Sigma_{i,i} = \lambda_i = i^{-\alpha}$ *for* $1 < \alpha$. *Assume that* $n < pk$ *for* $k = \dfrac{3 + \frac{1}{2^\alpha}}{4 + \frac{1}{2^{\alpha-2}}}$. *If* $\xi$ *satisfies*

$$\sum_{i=1}^{p} \frac{\lambda_i}{\lambda_i + \frac{1}{\xi}} = n,$$

*then* $cn^\alpha \le \xi \le c\left(n + 1 + \frac{p+1}{\alpha-1}\right)^\alpha$ *for* $c = \left(\dfrac{\alpha \sin(\pi/\alpha)}{\pi}\right)^\alpha$. *Note that* $\xi$ *is defined for the sake of the analysis, and it corresponds to* $\frac{1}{\tau_t}$.

*Proof.* From Simon et al. (2024), we have:

$$n = \sum_{i=1}^{p} \frac{i^{-\alpha}}{i^{-\alpha} + \frac{1}{\xi}} \le \int_0^p \frac{x^{-\alpha}}{x^{-\alpha} + \frac{1}{\xi}} \, dx = \int_0^\infty \frac{x^{-\alpha}}{x^{-\alpha} + \frac{1}{\xi}} \, dx - \int_p^\infty \frac{x^{-\alpha}}{x^{-\alpha} + \frac{1}{\xi}} \, dx = \frac{\pi}{\alpha \sin(\pi/\alpha)} \xi^{1/\alpha} - \int_p^\infty \frac{x^{-\alpha}}{x^{-\alpha} + \frac{1}{\xi}} \, dx.$$

Using $1 + x^\alpha \le (1+x)^\alpha$ for $x \ge 0$,

$$\int_p^\infty \frac{x^{-\alpha}}{x^{-\alpha} + \frac{1}{\xi}} \, dx = \int_p^\infty \frac{1}{1 + \frac{x^\alpha}{\xi}} \, dx \ge \int_p^\infty \frac{1}{(1 + \frac{x}{\xi^{1/\alpha}})^\alpha} \, dx = \frac{\xi(\xi^{1/\alpha} + p)^{-\alpha+1}}{\alpha - 1},$$

which implies that

$$n \le \frac{\pi}{\alpha \sin(\pi/\alpha)} \xi^{1/\alpha} - \frac{\xi(\xi^{1/\alpha} + p)^{-\alpha+1}}{\alpha - 1}.$$

Since the summand is decreasing, we can bound the Riemann sum by an integral, thus:

$$n = \sum_{i=1}^{p} \frac{i^{-\alpha}}{i^{-\alpha} + \frac{1}{\xi}} \ge \int_1^{p+1} \frac{x^{-\alpha}}{x^{-\alpha} + \frac{1}{\xi}} \, dx$$

$$= \int_0^\infty \frac{x^{-\alpha}}{x^{-\alpha} + \frac{1}{\xi}} \, dx - \int_0^1 \frac{x^{-\alpha}}{x^{-\alpha} + \frac{1}{\xi}} \, dx - \int_{p+1}^\infty \frac{x^{-\alpha}}{x^{-\alpha} + \frac{1}{\xi}} \, dx$$

$$= \frac{\pi}{\alpha \sin(\pi/\alpha)} \xi^{1/\alpha} - \int_0^1 \frac{x^{-\alpha}}{x^{-\alpha} + \frac{1}{\xi}} \, dx - \int_{p+1}^\infty \frac{x^{-\alpha}}{x^{-\alpha} + \frac{1}{\xi}} \, dx$$

$$\ge \frac{\pi}{\alpha \sin(\pi/\alpha)} \xi^{1/\alpha} - 1 - \int_{p+1}^\infty \frac{1}{1 + \frac{1}{\xi} x^\alpha} \, dx$$

$$\ge \frac{\pi}{\alpha \sin(\pi/\alpha)} \xi^{1/\alpha} - 1 - \int_{p+1}^\infty \frac{1}{\frac{1}{\xi} x^\alpha} \, dx$$

$$= \frac{\pi}{\alpha \sin(\pi/\alpha)} \xi^{1/\alpha} - 1 - \left[ \frac{\xi x^{-\alpha+1}}{-\alpha + 1} \right]_{p+1}^\infty$$

$$= \frac{\pi}{\alpha \sin(\pi/\alpha)} \xi^{1/\alpha} - \frac{\xi(p+1)^{-\alpha+1}}{\alpha - 1} - 1.$$

Recalling that $\alpha > 1$ and assuming $\xi < p^\alpha$, we derive:

$$\frac{\pi}{\alpha \sin(\pi/\alpha)} \xi^{1/\alpha} - 1 - \frac{p+1}{\alpha - 1} \le n \le \frac{\pi}{\alpha \sin(\pi/\alpha)} \xi^{1/\alpha}$$

$$\iff \left( \frac{n\alpha \sin(\pi/\alpha)}{\pi} \right)^\alpha \le \xi \le \left( \frac{\left(n + 1 + \frac{p+1}{\alpha-1}\right) \alpha \sin(\pi/\alpha)}{\pi} \right)^\alpha.$$

We conclude by proving that $\xi < p^\alpha$. For the sake of contradiction, assume that $\xi \geq p^\alpha$. Then,

$$n = \sum_{i=1}^{p} \frac{1}{1 + \frac{i^\alpha}{\xi}} = \sum_{i=1}^{p/2} \frac{1}{1 + \frac{i^\alpha}{\xi}} + \sum_{i=p/2+1}^{p} \frac{1}{1 + \frac{i^\alpha}{\xi}}$$

$$\geq \sum_{i=1}^{p/2} \frac{1}{1 + \frac{1}{2^\alpha}} + \sum_{i=p/2+1}^{p} \frac{1}{1 + 1}$$

$$= p \left( \frac{3 + \frac{1}{2^\alpha}}{4 + \frac{1}{2^{\alpha-2}}} \right),$$

which contradicts our assumption that $n < pk$. $\qquad\square$

**Proposition 8** (Non-asymptotic analysis of $\Omega$)**.** *Suppose that $\Sigma \in \mathbb{R}^{p\times p}$ is diagonal and $\Sigma_{i,i} = \lambda_i = i^{-\alpha}$ for $1 < \alpha$. Let $\tau_t$ be defined as in Proposition 7 and $\Omega$ be the solution to*

$$n\Omega = \sum_{i=1}^{p} \left( \frac{\lambda_i}{\lambda_i + \frac{1}{\xi}} \right)^2.$$

*Then,*

$$\Omega > \frac{\alpha - 1}{\alpha} - \frac{1}{2\alpha - 1} \left( \frac{n + 1 + \frac{p+1}{\alpha-1}}{p + 1} \right)^{2\alpha-1} - \frac{1}{n}.$$

*Proof.* We have that

$$\sum_{i=1}^{p} \left( \frac{i^{-\alpha}}{i^{-\alpha} + \frac{1}{\xi}} \right)^2 \leq \int_0^\infty \left( \frac{x^{-\alpha}}{x^{-\alpha} + \frac{1}{\xi}} \right)^2 dx = \frac{\pi(\alpha - 1)}{\alpha^2 \sin(\pi/\alpha)} \xi^{1/\alpha}$$

Besides, since the summand is monotonically decreasing:

$$n\Omega = \sum_{i=1}^{p} \left( \frac{i^{-\alpha}}{i^{-\alpha} + \frac{1}{\xi}} \right)^2 \geq \int_1^{p+1} \left( \frac{x^{-\alpha}}{x^{-\alpha} + \frac{1}{\xi}} \right)^2 dx$$

$$= \int_0^\infty \left( \frac{x^{-\alpha}}{x^{-\alpha} + \frac{1}{\xi}} \right)^2 dx - \int_0^1 \left( \frac{x^{-\alpha}}{x^{-\alpha} + \frac{1}{\xi}} \right)^2 dx - \int_{p+1}^\infty \left( \frac{x^{-\alpha}}{x^{-\alpha} + \frac{1}{\xi}} \right)^2 dx$$

$$= \frac{\pi(\alpha - 1)}{\alpha^2 \sin(\pi/\alpha)} \xi^{1/\alpha} - \int_0^1 \left( \frac{x^{-\alpha}}{x^{-\alpha} + \frac{1}{\xi}} \right)^2 dx - \int_{p+1}^\infty \left( \frac{x^{-\alpha}}{x^{-\alpha} + \frac{1}{\xi}} \right)^2 dx$$

$$\geq \frac{\pi(\alpha - 1)}{\alpha^2 \sin(\pi/\alpha)} \xi^{1/\alpha} - 1 - \int_{p+1}^\infty \left( \frac{1}{1 + \frac{1}{\xi} x^\alpha} \right)^2 dx$$

$$\geq \frac{\pi(\alpha - 1)}{\alpha^2 \sin(\pi/\alpha)} \xi^{1/\alpha} - 1 - \int_{p+1}^\infty \frac{1}{(\frac{1}{\xi} x^\alpha)^2} dx$$

$$= \frac{\pi(\alpha - 1)}{\alpha^2 \sin(\pi/\alpha)} \xi^{1/\alpha} - 1 - \left[ \frac{\xi^2 x^{-2\alpha+1}}{-2\alpha + 1} \right]_{p+1}^\infty$$

$$= \frac{\pi(\alpha - 1)}{\alpha^2 \sin(\pi/\alpha)} \xi^{1/\alpha} - \frac{\xi^2 (p + 1)^{-2\alpha+1}}{2\alpha - 1} - 1.$$

Let's now utilize the upper and lower bounds for $\xi$ from Proposition 7. Then, we have

$$n\Omega \geq \frac{\pi(\alpha-1)}{\alpha^2 \sin(\pi/\alpha)} \frac{n\alpha \sin(\pi/\alpha)}{\pi} - \frac{\xi^2(p+1)^{-2\alpha+1}}{2\alpha-1} - 1$$

$$= \frac{n(\alpha-1)}{\alpha} - \frac{\xi^2(p+1)^{-2\alpha+1}}{2\alpha-1} - 1$$

$$\geq \frac{n(\alpha-1)}{\alpha} - \left(\frac{\left(n+1+\frac{p+1}{\alpha-1}\right)\alpha \sin(\pi/\alpha)}{\pi(p+1)}\right)^{2\alpha} \frac{p+1}{2\alpha-1} - 1$$

$$\implies \Omega > \frac{\alpha-1}{\alpha} - \left(\frac{\left(n+1+\frac{p+1}{\alpha-1}\right)\alpha \sin(\pi/\alpha)}{\pi(p+1)}\right)^{2\alpha} \frac{p+1}{n(2\alpha-1)} - \frac{1}{n}$$

$$> \frac{\alpha-1}{\alpha} - \frac{1}{2\alpha-1}\left(\frac{n+1+\frac{p+1}{\alpha-1}}{p+1}\right)^{2\alpha-1} - \frac{1}{n},$$

since $\dfrac{\alpha \sin(\pi/\alpha)}{\pi} < 1$ for $\alpha > 1$. $\qquad\square$

**Proposition 9.** *Under the assumption that* $n < \min\left((p+1)\dfrac{\alpha-2}{\alpha}, p\left(\dfrac{3+\frac{1}{2^\alpha}}{4+\frac{1}{2^{\alpha-2}}}\right), p\dfrac{\pi\left(\sqrt{\frac{\alpha}{2}}-1\right)^{1/\alpha}}{\alpha \sin(\pi/\alpha)} - \dfrac{p+1}{\alpha-1}\right) - 1$

*and* $\alpha > 3$, *we can find a masked surrogate-to-target setting that improves over the risk of the standard target model by selecting all features i such that* $\zeta_i^2 > 1 - \Omega$.

*Proof.* From Proposition 8, we have

$$\Omega > \frac{\alpha-1}{\alpha} - \frac{1}{2\alpha-1}\left(\frac{n+1+\frac{p+1}{\alpha-1}}{p+1}\right)^{2\alpha-1} - \frac{1}{n}.$$

It's then enough to show that we can find a set of $i$'s such that

$$\zeta_i^2 > \frac{1}{\alpha} + \frac{1}{2\alpha-1}\left(\frac{n+1+\frac{p+1}{\alpha-1}}{p+1}\right)^{2\alpha-1} + \frac{1}{n}.$$

From proof of Proposition 3, we know that

$$\zeta_i^2 > c' \iff i > \tau_t^{-1/\alpha}\left(\frac{\sqrt{c'}}{1-\sqrt{c'}}\right)^{1/\alpha}.$$

Hence, using the bound on $\frac{1}{\tau_t} = \xi$ from Proposition 7, it's enough to find indices $i$ such that

$$i > \frac{\alpha \sin(\pi/\alpha)}{\pi}\left(n+1+\frac{p+1}{\alpha-1}\right)\left(\frac{\sqrt{c'}}{1-\sqrt{c'}}\right)^{1/\alpha} \quad \text{where } c' = \frac{1}{\alpha} + \frac{1}{2\alpha-1}\left(\frac{n+1+\frac{p+1}{\alpha-1}}{p+1}\right)^{2\alpha-1} + \frac{1}{n}. \quad (21)$$

By our assumption $p+1 > n+1+\frac{p+1}{\alpha-1}$, we obtain that $\frac{2}{\alpha} > c'$. Since $\left(\frac{\sqrt{x}}{1-\sqrt{x}}\right)^{1/\alpha}$ in increasing with $x$ when $0 \leq x \leq 1$, we have

$$\left(\frac{1}{\sqrt{\frac{\alpha}{2}}-1}\right)^{1/\alpha} \geq \left(\frac{\sqrt{c'}}{1-\sqrt{c'}}\right)^{1/\alpha}.$$

Then, to ensure the existence of an interval of $i$'s satisfying the above inequality, we choose

$$p - (p+1)\frac{\alpha \sin(\pi/\alpha)}{\pi(\alpha-1)}\left(\frac{1}{\sqrt{\frac{\alpha}{2}}-1}\right)^{1/\alpha} \geq (n+1)\frac{\alpha \sin(\pi/\alpha)}{\pi}\left(\frac{1}{\sqrt{\frac{\alpha}{2}}-1}\right)^{1/\alpha}$$

$$\iff p\frac{\pi\left(\sqrt{\frac{\alpha}{2}}-1\right)^{1/\alpha}}{\alpha \sin(\pi/\alpha)} - \frac{p+1}{\alpha-1} \geq n+1$$

One can verify that the LHS expression is always positive when $\alpha > 3$. Thus, discarding the features $i$ provided in the interval (21) will strictly improve the test risk of the masked surrogate-to-target model over the standard target model. □

## C Proofs for Section 5

**Theorem 3** (Distributional characterization, Han & Xu (2023)). *Let $\kappa_s = p/m > 1$ and suppose that, for some $M > 1$, $1/M \le \kappa_s, \sigma_s^2 \le M$ and $\|\Sigma_s\|_{op}, \|\Sigma_s^{-1}\|_{op} \le M$. Let $\tau_s \in \mathbb{R}$ be the unique solution of the following equation:*

$$\kappa_s^{-1} = \frac{1}{p} \operatorname{tr}\left((\Sigma_s + \tau_s I)^{-1}\Sigma_s\right). \tag{22}$$

*We define the function $\gamma_s : \mathbb{R}^p \to \mathbb{R}$ and the random variable based on $g_s \sim \mathcal{N}(0, I)$ as follows:*

$$\gamma_s^2(\beta_\star) := \kappa_s \left(\sigma_s^2 + \mathbb{E}_{(x,y)\sim\mathcal{D}_s}[\|\Sigma_s^{1/2}(\beta^s - \beta_\star)\|_2^2]\right) = \kappa_s \frac{\sigma_s^2 + \tau_

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

$\qquad\square$