# OpenReview forum: "High-dimensional Analysis of Knowledge Distillation: Weak-to-Strong Generalization and Scaling Laws"
_ICLR.cc/2025/Conference — ICLR 2025 Spotlight_

### Official Review · Reviewer_vDyf · 2024-10-28

**Soundness:** 3
**Presentation:** 3
**Contribution:** 2
**Rating:** 8
**Confidence:** 3

**Summary:**

This paper considers the problem of knowledge distillation under the setting of linear ridge-less regression in the teacher student scenario.

The setting considered in this paper is the the proportional regime where $p,n\to\infty$ and their ratio $\kappa_t = p/n$ is kept fixed. They consider $\kappa_t > 1$ in the overparametrised regime.

The models considered in the paper are three:
* _The Surrogate-to-Target model_ where the data is generated from a dataset $\mathcal{D}$ with input $x\in\mathbb{R}^d$ and output $y = x^\top \beta_\star + z$ with $ \beta_\star$ a teacher vector. This data is used to estimate a min norm estimator called $\beta^s$ and generate a second data set $y^s = x^\top \beta^s + z$ and the final estimation is done as $\beta^{s2t}$ from $(x,y^s)$.
* _The Standard Target model_ where the model is evaluated on the generated data $(x, y)$
* _The Covariance Shift model_ here the dataset is generated with a certain choice of covariance and then the population risk evaluated on a different covariance model.

The first part of the paper is devoted to finding the performance conditioned on a specific teacher while the second to last section considers the full _Surrogate-to-Target_ setup.
The authors also consider the procedure of Masking for the surrogate model. In this case the surrogate model has been trained on a masked version of the data and the new labels are generated from the original inputs and the labels of the surrogate model.

The main technical results presented in the main are the characterisation of the population risk for the model conditioned on $\beta^s$ and then for the _Surrogate-to-Target_ model.

For the case conditioned on the target the authors are able to precisely derive the effect of the surrogate model on the final student, showing specific conditions (depending on the covariates and $\beta^\star$) under which a $\beta^{s2t}$ performs better than a _The Standard Target model_. The same is true for the masking.

**Strengths:**

This paper is mathematically sound and considers an interesting setting. The main strengths are
* The derivations of the different transition values for the covariates is sharp and to my knowledge is a novel finding.
* The theory match with simulations also work at finite dimension even if the result is high-dimensional.
* The model introduced and studied is expressive enough to show different behaviour that characterise performances.

**Weaknesses:**

* Right now the mathematical result is introduced in generality without explaining the idea behind the proof. The authors could briefly explain that to derive the results one should apply the theory from [Han&Xu2023] that relies on the use of of the convex gordon min max theorem.
* The authors provide some numerical simulations on ResNet-50 on a CIFAR10 classification showing a result that qualitatively differs from the theory. Either this limitation is explained in detail or I don't think it is necessary to be shown.
* Is there any reason why the authors consider in their technical results the ridgeless estimator instead of the ridge one? A long series of works (e.g. [Hastie2020, Louriero2022]) considers general loss and provides similar bounds.
* _(Minor)_ Section 4 is presented unclearly. The settings for the propositions are not well explained and need to be introduced more clearly.

[Hastie 2020] Surprises in High-Dimensional Ridgeless Least Squares Interpolation. Annals of Stats 2020

[Loureiro2022] Learning curves of generic features maps for realistic datasets with a teacher-student model. Neurips 2022

**Questions:**

* I would love to see Figure 1b and Figure 2b in LogLog Scale. One of the main points of the authors in Section 4 (Proposition 5) concerns the learning rates in the high-dimensional limit. It would be nice to see them in Figure 1b.
* Is it possible to generalise the result of Section 3.1 to the case where the chosen features are chosen with a matrix $A$ which has a non zero kernel? The masking seems a specific case of this.


* There is a broken citation on page 3.
* On line 334 is it "omniscent test risk estimate"?

---

> ### Author Response · Authors · 2024-11-22
>
> We thank the reviewer for the positive evaluation of our work and for the detailed comments. We address all of them below, and we have uploaded a revised version of the manuscript highlighting in red the main changes.
>
> >W1: Right now the mathematical result is introduced in generality without explaining the idea behind the proof. The authors could briefly explain that to derive the results one should apply the theory from [Han&Xu2023] that relies on the use of the convex gordon min max theorem.
>
> **Response:** Thanks for the feedback. We added a short proof sketch for Theorems 1 and 2 in the main text. In addition, we want to note that we give credit to the theory developed by Han & Xu (2023) in lines 237-239 and 477-479.
>
> >W2: The authors provide some numerical simulations on ResNet-50 on a CIFAR10 classification showing a result that qualitatively differs from the theory. Either this limitation is explained in detail or I don't think it is necessary to be shown.
>
> **Response:** Thank you for your insightful feedback. We improved our explanation of not being able to outperform the standard target model in practical settings, unlike our theoretical results. Here is our revised explanation:
>
> “The reason why the surrogate-to-target model underperforms the standard target model is that the surrogate model is not able to follow the feature selection mechanism characterized in Proposition 2 as there’s no notion of feature selection in neural networks, unlike linear models. This suggests that the feature selection mechanism is crucial for surpassing the performance of the target model.”
>
> We believe that this figure is valuable because the phenomenon of weak-to-strong generalization extends beyond large language models and can also be observed in neural network architectures.
>
>
> >W3: Is there any reason why the authors consider in their technical results the ridgeless estimator instead of the ridge one? A long series of works (e.g. [Hastie2020, Louriero2022]) considers general loss and provides similar bounds.
>
> **Response:** We appreciate the insightful question. Our result can be extended to ridge regression by defining $\tau$ based on the ridge parameter. This extension is achievable due to the characterization of the parameter $\hat{\beta}$ provided by Han & Xu (2023).  Indeed, we have made initial efforts in this direction, as discussed in Appendix A.1, where we apply the results of Section 3 to ridge regression. Our choice to perform the analysis for ridgeless regression was motivated by the desire to keep the model as simple as possible while conveying the important insights of the surrogate-to-target model.
>
> Furthermore, we want to highlight that the mentioned previous works (e.g. Hastie et al. (2020), Louriero et al. (2022)) that study the ridge regression provide only the asymptotic risk characterization and do not provide the characterization of the parameter $\hat{\beta}$. Therefore, our work is based on a relatively advanced technique developed by Han & Xu (2023).
>
>
> >W4: (Minor) Section 4 is presented unclearly. The settings for the propositions are not well explained and need to be introduced more clearly.
>
> **Response:** Thank you for your helpful feedback, we have revised the propositions in Section 4 to improve clarity.
>
> *Trevor Hastie, Andrea Montanari, Saharon Rosset, and Ryan J. Tibshirani. Surprises in high-dimensional ridgeless least squares interpolation, 2020.*
>
> *Qiyang Han and Xiaocong Xu. The distribution of ridgeless least squares interpolators, 2023.*
>
> *Bruno Loureiro, Cédric Gerbelot, Hugo Cui, Sebastian Goldt, Florent Krzakala, Marc Mézard, and Lenka Zdeborová. Learning curves of generic features maps for realistic datasets with a teacher-student model, 2022*

---

> > ### Author Response · Authors · 2024-11-22
> >
> > >Q1: I would love to see Figure 1b and Figure 2b in LogLog Scale. One of the main points of the authors in Section 4 (Proposition 5) concerns the learning rates in the high-dimensional limit. It would be nice to see them in Figure 1b.
> >
> > **Response:** Thank you for your thoughtful suggestion. In response to your question, we have added two new figures to Appendix B (cf. Figures 4 and 5). For ease of accessibility, here is also a [link](https://anonymous.4open.science/r/High-dimensional-Analysis-of-Knowledge-Distillation-Weak-to-Strong-Generalization-and-Scaling-Laws-87EC) to the plots. In our current Figure 1b, the sample size $n$ varies from small values up to values close to $p$. Even though a linear trend is observable for smaller values of $n$ when $p=500$ in the log-log scale, as $n$ approaches $p$, it’s less apparent due to finite-sample effects of $p$. We also note that the asymptotic approximations in Propositions 3 and 4 require $p$ to be significantly large compared to $n$.
> > Therefore, we conducted an additional experiment with a larger dimension ($p=5000$) and kept the same range of $n$ values to satisfy $n \ll p$. In this case, we observe a clear linear behavior in the log-log plot, which is consistent with the scaling law results presented in Proposition 5. Similarly, we also demonstrated the log-log scale for Figure 2b in the new Figure 5.
> >
> > >Q2: Is it possible to generalize the result of Section 3.1 to the case where the chosen features are chosen with a matrix A  which has a non zero kernel? The masking seems a specific case of this.
> >
> > **Response:** It seems to us that generalizing the results of Section 3.1 to the case where the features are chosen via a matrix A actually corresponds to the optimal surrogate given by Proposition 1. In fact, such an optimal surrogate is obtained by multiplying a certain matrix with the ground-truth parameters.
> >
> > In case we have misunderstood the point of the reviewer, we remain at disposal for additional clarifications.
> >
> >
> > >Q3: There is a broken citation on page 3.
> >
> > **Response:** Thank you for flagging this, we have fixed the broken citation.
> >
> >
> > >Q4: On line 334 is it "omniscent test risk estimate"?
> >
> > **Response:** Thank you for pointing this out, we updated that part accordingly.

---

> ### Comment · Reviewer_vDyf · 2024-11-22
>
> I want to thank the authors for the time taken to reply.
> I acknowledge the changes in the text to make the result more transparent in the text and the figures.
> I thank the authors for explaining the relationship of their work with the previous ones: it clarified my questions. I also don't think it is necessary to provide a full analysis of ridge regression as already interesting behaviour is found with this simple model.
>
> > It seems to us that generalising the results of Section 3.1 to the case where the features are chosen via a matrix A actually corresponds to the optimal surrogate given by Proposition 1. In fact, such an optimal surrogate is obtained by multiplying a certain matrix with the ground-truth parameters.
> > In case we have misunderstood the point of the reviewer, we remain at disposal for additional clarifications.
>
> At the cost of being repetitive, I would like to explain myself better. Considering a projection $A$, one can define in the same spirit the masked s2t model as $\tilde{y} = (A x)^T (A \beta_\star) + \tilde{z}$, and then one can choose to include or not the projection matrix in the $\mathcal{D}_t$, for example, one could consider $y = (A x)^T \beta_s+ z$. I don't see how these cases can be framed (with a change of variables) to the results in Proposition 1, and I am genuinely curious, also because $\beta_s^\star$ doesn't seem to depend on $\Sigma_s$.
>
> I have decided to increase my score.
>
> P.S. A minor typo in line 290 is the variance of the noise. I believe it should be $\sigma_t$ instead of $\sigma_s$.

---

> ### Author Response · Authors · 2024-11-23
>
> We thank the reviewer for their prompt response and positive feedback. We also appreciate the clarification for the projection matrix $A$.
>
> For the surrogate model stage, the labels $\tilde{y}$ are generated by the following linear equation: $\tilde{y} = (\mathbf{A \tilde{x}})^\top (\mathbf{A \beta_*})$. Then, the surrogate parameter $\mathbf{\beta^s}$ is estimated using the features $\mathbf{A \tilde{x}}$, as described in line 291. When the surrogate model has infinitely many data, the covariance matrix $\Sigma_s$ does not affect the surrogate parameter $\mathbf{\beta^s}$, so the surrogate parameter $\mathbf{\beta^s}$ is equal to $\mathbf{A \beta_*}$. For the analysis of this setting under finite data, please refer to Appendix C.1.
>
> For the target model stage, the labels $y$ are generated by the following linear equation: $y = (\mathbf{A x})^\top \mathbf{\beta^s}$. Then, the surrogate-to-target parameter $\mathbf{\beta^{s2t}}$ is estimated using the features $\mathbf{x}$, as described in line 292. The label-generation process reduces to the following:
>
> $$ \qquad y = (\mathbf{A x})^\top \mathbf{\beta^s} = \mathbf{x}^\top (\mathbf{A^\top \beta^s}) = \mathbf{x}^\top (\mathbf{A^\top A \beta_*}).$$
>
> Let’s define the matrix $\mathbf{K}$ as follows:
>
> $$ \qquad \mathbf{K} = ((\Sigma_t  + \tau_t \mathbf{I})^{-1} \Sigma_t + \frac{\Omega \tau_t^2}{1- \Omega} \Sigma_t^{-1} (\Sigma_t  + \tau_t \mathbf{I})^{-1})^{-1}.$$
>
> Now, if the projection matrix $\mathbf{A}$ is rank$-p$, then the optimal projection matrix $\mathbf{A}$ can be found by solving $\mathbf{ A^\top A} = \mathbf{K}$ following Proposition 1.
>
> If the projection matrix $\mathbf{A}$ is rank$-p_s$ where $p_s \leq p$, then we need to select the largest $p_s$ signal coefficients of the labels $y$. This corresponds to the the largest $p_s-$eigen-directions based on $\lambda_i^2 \beta_{*, i}$ where $\lambda_i$ are the eigenvalues of $\Sigma_t$. (Here, WLOG, we assume that $\Sigma_t$ is diagonal by Observation 2). Let $\mathbf{U} \in \mathbb{R}^{p \times p_s}$ be the matrix whose columns are the largest $p_s-$eigen-directions with unit norm. Since the optimal surrogate parameter in Proposition 1 can be decoupled for each dimension, then an optimal projection matrix can be $\mathbf{U^\top K^{1/2}}$ following Proposition 1.
>
> Kindly let us know if this post addresses your question. If the reviewer thinks this generalization is worth writing in the paper, we can incorporate it in the Appendix.
>
> > P.S. A minor typo in line 290 is the variance of the noise. I believe it should be $\sigma_t$ instead of $\sigma_s$.
>
> Thanks for the typo notice. We fixed the typo and uploaded the new version.

---

### Official Review · Reviewer_pg8p · 2024-11-02

**Soundness:** 3
**Presentation:** 2
**Contribution:** 3
**Rating:** 6
**Confidence:** 3

**Summary:**

The paper provides a sharp characterization for knowledge distillation in the high-dimensional regression setting, including both model shift and distribution shift, cases. Concretely, the paper characterizes the precise risk of the target model in both cases through non-asymptotic bounds in terms of sample size and data distribution under mild conditions. As a consequence, the paper identifies the form of the optimal surrogate model, which reveals the benefits and limitations of such processes. Finally, the paper validates the results by numerical experiments.

**Strengths:**

1. Knowledge distillation and weak-to-strong generalization are significant topics today, and their theory is very poor. Therefore, this is a meaningful paper for me.
2. The theory is complete and well-written.
3. The derived bounds seem tight because they are matched with empirical results.

**Weaknesses:**

1. The theory only focuses the high-dimensional linear regression setting, which is well-studied in the literature. Besides, the results can not be extended to neural networks directly.
2. A typo in line 134.

**Questions:**

1. Can you give me some insights to extend the theory to neural networks (even if two-layer neural network)? I think the authors should also discuss this in the refined version.

---

> ### Author Response · Authors · 2024-11-22
>
> We thank the reviewer for appreciating the strengths of our work and for the detailed comments. We reply to each of the points raised in the review below. We have also uploaded a revised version of the manuscript highlighting in red the main changes.
>
>
> >W1: The theory only focuses the high-dimensional linear regression setting, which is well-studied in the literature. Besides, the results can not be extended to neural networks directly.
>
> **Response:** Even if high-dimensional linear regression has been the subject of intense study in the literature, a precise characterization of knowledge distillation and weak-to-strong generalization was still lacking prior to our work. We see the extension to neural networks as an exciting future direction, and, following the suggestion of the reviewer, we now discuss such extension in Section 6 of the revision, see our response to Q1 below.
>
> >W2: A typo in line 134.
>
> **Response:** Thanks for noticing this, we have corrected the broken reference.
>
> >Q1: Can you give me some insights to extend the theory to neural networks (even if two-layer neural network)? I think the authors should also discuss this in the refined version.
>
> **Response:** Going beyond linear regression, the precise asymptotics of the test error of the ERM solution were provided by Mei & Montanari (2022) for the random features model and by Montanari & Zhong (2022) for two-layer neural networks in the NTK regime. However, a non-asymptotic characterization (similar to that given by Han & Xu (2023) for linear regression) remains an open problem. The resolution of this open problem, as well as the analysis of the phenomena of knowledge distillation and weak-to-strong generalization, represent exciting directions for future research.
>
> Thank you for this suggestion, we now discuss this point in the final section of the revision.
>
> *Song Mei and Andrea Montanari, “The generalization error of random features regression: Precise asymptotics and the double descent curve”, Communications on Pure and Applied Mathematics, 2022.*
>
> *Andrea Montanari and Yiqiao Zhong, “The interpolation phase transition in neural networks: Memorization and generalization under lazy training”, Annals of Statistics, 2022.*

---

> > ### Comment · Reviewer_pg8p · 2024-11-25
> >
> > Thank the authors for the detailed reply. Since some of my concerns have been addressed, I will maintain my positive score.

---

### Official Review · Reviewer_GbMC · 2024-11-02

**Soundness:** 2
**Presentation:** 2
**Contribution:** 2
**Rating:** 6
**Confidence:** 3

**Summary:**

This paper studies the problem of knowledge distillation under linear regression. In the first stage, data are collected from a surrogate model. In the second stage, a target model is trained using the data generated in the first stage. The authors characterize the non-asymptotic excess risk of the target model under "model shift" setting and "distribution shift" setting. Numerical results are provided, justifying their theory on ridgeless regression and on neural network architectures.

**Strengths:**

The authors provide comprehensive theoretical results for weak-to-strong generalization, giving a exact characterization of the excess risk of the weak-to-strong estimator. This knowledge distillation problem is important in modern machine learning, indicating the significance of this work.

**Weaknesses:**

The presentation of this paper is in general not very satisfactory, in the sense that this paper is lack of necessary intuition and explanation. For example, how to interpret the non-asymptotic bounds and what does each term stand for? Why it is possible that weak-to-strong estimator is even better than purely using the strong model?

**Questions:**

1. Can you provide intuition why the risk of the surrogate-to-target model under the optimal selection of the parameters scales the same as that of the target model (even though there is a strict improvement in the risk)? I am wondering why improvement is possible. Is it because, for example, if the tail components of the covariance is zero, then features on these components are essentially useless, therefore an surrogate that omits those components will be better?

2. your equation (8) involves $\beta^{s2t}$. Does it mean that your asymptotic risk estimate (9) also involves $\beta^{s2t}$ and thus can not be directly computed? I think in the final bound $\beta^{s2t}$ should not appear; otherwise I can just claim the definition of the excess risk of $\beta^{s2t}$ is already an exact characterization of itself.

3. In observation1, you assume jointly diagonalizability. Is there fundamental hardness to remove this assumption?

---

> ### Author Response · Authors · 2024-11-22
>
> We thank the reviewer for recognizing the significance of our work, and the comprehensiveness of our results, as well as for the detailed comments. We address all of them below, and we have uploaded a revised version of the manuscript highlighting in red the main changes.
>
> >W: The presentation of this paper is in general not very satisfactory, in the sense that this paper is lack of necessary intuition and explanation. For example, how to interpret the non-asymptotic bounds and what does each term stand for? Why it is possible that weak-to-strong estimator is even better than purely using the strong model?
>
> **Response:** Thank you for raising this issue, which has allowed us to improve the presentation of our results in the revision. We now discuss the interpretation of the terms appearing in the expression for the risk (see l. 227-230 and 240-244 of the revision), and we give the intuition behind improving the strong model with the weak-to-strong estimator (see l. 270-281 of the revision). These changes are discussed in detail in our response to Q1 below.
>
> >Q1: Can you provide intuition why the risk of the surrogate-to-target model under the optimal selection of the parameters scales the same as that of the target model (even though there is a strict improvement in the risk)? I am wondering why improvement is possible. Is it because, for example, if the tail components of the covariance is zero, then features on these components are essentially useless, therefore an surrogate that omits those components will be better?
>
> **Response:** The improvement in the surrogate-to-target model's risk arises because we effectively cut/shrink (depending on the optimal masked surrogate or optimal surrogate model) the tail components of the covariance matrix that contribute less useful information. This is illustrated in Figure 1-a.
>
> However, the scaling law of the risk remains the same because the significant/uncut components (with large corresponding eigenvalues and signal coefficients) of the covariance matrix dominate the behavior of the risk and determine the scaling law exponent. This essentially follows from using the expression discussed in line 430 as the lower bound.
>
> To improve the presentation, we split the risk in (10) into two parts, $a$ and $b$, providing intuition for each of them. Specifically, the terms $a$ and $b$ corresponds to the following:
>
> $$\qquad a = || \Sigma_t^{1/2} (\beta_*  -  \mathbf{\theta}_1 \beta^s   ) ||_2^2$$
>
> $$ \qquad b = \gamma_t^2(\beta^s) \mathbb{E}_{\mathbf{g}_t} [ \mathbf{\theta}_2^\top \Sigma_t \mathbf{\theta}_2 ] $$
>
> We have added the following explanation in the main text: “In the asymptotic risk, the term $(a)$ corresponds to a part of the bias risk caused by the model shift ($\beta^s$), and the implicit regularization term where the eigenvalues of $\mathbf{\theta}_1$ are less than 1. The term $(b)$ corresponds to the remaining part of the bias and variance risks.”
>
> To answer the question of why there is room for improvement by introducing the surrogate parameter, we have added the following paragraph: “The surrogate parameter $\beta^s$ that minimizes the individual term (a) in 10 is $\mathbf{\theta_1^{-1}}  \beta_*$. On the other hand, the surrogate parameter $\beta^s$ that minimizes the individual term $(b)$ is the zero vector, which follows from (16) in Appendix A. Now, we are going to jointly minimize the asymptotic risk in the next proposition. The optimal surrogate parameter is visualized as the green curve in Figure 1.”
>
> We have provided the intuition behind a better performance of the surrogate parameter at the end of Section 3 as follows: “The intuition behind improving the performance of the standard target model by utilizing a surrogate parameter $\beta^s$ different from $\beta_*$ is associated with the implicit regularization of the minimum norm interpolator in the over-parametrized region $(p > n)$. As long as the covariance matrix eigenvalues are not constant, there is a way to mitigate the bias risk caused by the implicit regularization. This implicit regularization term is specific to the over-parametrized region. Indeed, in the next proposition, we will show that the optimal surrogate parameter $\beta^s$ is $\beta_*$ when the target model is under-parametrized:
>
> **Proposition**  The optimal surrogate parameter $\beta^s$ that minimizes the asymptotic risk in the under-parametrized region ($n > p$) is equivalent to the ground truth parameter $\beta_*$. In other words, for any $\beta^s$, the surrogate-to-target model cannot outperform the standard target model in the asymptotic risk.
>
> In other words, the result above shows that the improvement in the surrogate-to-target model compared to the standard target model is special to the over-parameterized region.”

---

> > ### Author Response · Authors · 2024-11-22
> >
> > >Q2: your equation (8) involves $\beta^{s2t}$. Does it mean that your asymptotic risk estimate (9) also involves $\beta^{s2t}$ and thus can not be directly computed? I think in the final bound $\beta^{s2t}$ should not appear; otherwise I can just claim the definition of the excess risk of $\beta^{s2t}$ is already an exact characterization of itself.
> >
> > **Response:** Thanks for pointing out this issue. There is a notational typo in the definition of the function $\gamma_t$. The expression $\mathbb{E}_{(\mathbf{x}, y) \sim \mathcal{D}_t(\beta^s)}[||\Sigma_t^{½} (\beta^{s2t} - \beta^s||_2^2 ]$ represents the *non-asymptotic risk* of the surrogate-to-target model when the training and test data is generated with respect to $\beta^s$. However, the function $\gamma_t$ should be defined as the *asymptotic risk* of the surrogate-to-target model under the same conditions. While the definition contains a mistake, throughout the paper, we have treated the function $\gamma_t$​ correctly. Indeed, the empirical and theoretical risks in Figures 1b and 2b are perfectly aligned with each other and this error does not affect the proofs. Specifically, $\gamma_t(\beta^s)$ is provided in both submitted and revised versions as the following: $$\gamma_t^2(\beta^s)= \kappa_t  \frac{\sigma_t^2 + \tau_t^2 ||\Sigma_t^{½} (\Sigma_t + \tau_t \mathbf{I})^{-1} \beta^s||_2^2}{1 - \frac{1}{n} \textbf{tr}( (\Sigma_t + \tau_t \mathbf{I})^{-2} \Sigma_t^2)}.$$
> >
> > That is why we consider this mistake as a notational typo. In the revised version, the correct definition is the following:
> > $$\gamma_t^2(\beta^s) = \kappa_t (\sigma_t^2 + \bar{\mathcal{R}}_{\kappa_t, \sigma_t}^{s2t}(\Sigma_t, \beta^s, \beta^s)).$$
> >
> >
> > >Q3: In observation1, you assume jointly diagonalizability. Is there fundamental hardness to remove this assumption?
> >
> > **Response:** Thanks for the question. Observation 1 establishes a one-to-one mapping between the domains of $\beta^s$ and $\Sigma_s$​. The number of free dimensions in the domain of $\beta^s$ is $p$, while the number of free dimensions in the domain of $\Sigma_s$​ is also $p$ under the assumption that $\Sigma_s$​ and $\Sigma_t$​ are jointly diagonalizable. Thus, an equivalence between the two can be established when the covariance matrices are jointly diagonalizable.
> >
> > However, when the joint diagonalizability assumption is removed, the number of free dimensions in the domain of $\Sigma_s$​ increases to $p^2$. This higher dimensionality makes it impossible to establish an equivalence.
> >
> > We would like to clarify that the joint diagonalizability assumption originates from the covariance shift model proposed by Mallinar et al. (2024), and the same assumption is also utilized in other transfer learning settings (Song et al. 2024). However, throughout this paper, we do not make any assumptions related to joint diagonalizability. It is important to note that the assumption of $\Sigma_t$​ being diagonal is made without loss of generality, as discussed in Observation 2.
> >
> > *Yanke Song, Sohom Bhattacharya, and Pragya Sur. Generalization error of min-norm interpolators in transfer learning, 2024*

---

> > ### Comment · Reviewer_GbMC · 2024-11-25
> >
> > Thank the authors for your detailed reply. Some of my concerns are addressed, so I will raise my score to 6.

---

### Official Review · Reviewer_E9SJ · 2024-11-04

**Soundness:** 4
**Presentation:** 3
**Contribution:** 3
**Rating:** 8
**Confidence:** 3

**Summary:**

In this paper, the authors propose a precise characterization of the benefits of knowledge distillation, mostly in the context of gaussian linear regression. In particular, the main set-up considers the excess risk of linear regression on a given distribution, but with the learner only able to access pseudolabels generated by a surrogate model instead of the true labels. Notably, the authors show that under a covariance-shift model (i.e. the distribution of covariates $x$ may change between the surrogate and target stages, but the underlying predictor $\beta_\star$ remains the same in between), then the optimal surrogate predictor minimizing the (asymptotic) excess risk on the target distribution is a weighted version of the ground-truth predictor $\beta_\star$, which which amplifies entries corresponding to large eigenvalues of the (diagonal) covariance matrix above a certain threshold, and shrinks entries below a threshold. Furthermore, in a masked setting, where the surrogate model is restricted to selecting a subset of the full set of features, then similarly the optimal surrogate predictor selects predictor entries above a certain threshold of covariance eigenvalues. Lastly, the authors show that in a certain asymptotic regime, an optimal surrogate-to-target model (i.e. a model trained on target distribution covariates with surrogate model pseudolabels) has the same excess risk as the least-squares target model trained with the true labels, demonstrating that knowledge distillation in a sense cannot beat out ERM with access to true labels.

**Strengths:**

This paper is rather well-written and contains quite a few interesting theoretical insights. The results draw clear delineations on how knowledge distillation through a surrogate model can help. As someone who hasn't thought about overparameterized linear regression in a while, Proposition 1 and Corollary 1 were rather surprising results, demonstrating that for a dataset size proportional (but smaller) than the number of parameters, the optimal surrogate predictor to use for generating pseudolabels is actually not the ground truth predictor, and that there is (in theory) always room for benefit as long as the covariance is non-isotropic, which implies that a learner benefits from using something other than the actual distribution of labels.

In addition to the theory, the numerical results on CIFAR-10 also counterintuitively support that a learner only trained on surrogate pseudolabels on the target domain actually outperform the surrogate model itself, which has access to true labels (albeit with a different covariate distribution...?).

**Weaknesses:**

Though the theoretical results are interesting, there are a few aspects that are worth clarifying. Notably, even though it is demonstrated that there can exist a surrogate model that induces better risk on the target model than using the true labels, in general a surrogate model is typically not trained with foreknowledge of the task distribution. I believe this is what Section 5 is trying to convey, but it is not clear to me after reading that section how to interpret the result therein. In particular, it should be explained how this relates to, for example, a target model that is trained using strong labels to demonstrate the marginal gain (or loss).

In general, the paper accrues a lot of jargon and notation; it would be very helpful to either create a table containing the different set-ups/regimes considered and the summary conclusion of the relative gain/suboptimality of knowledge distillation and/or a notation table that summarizes what the various risks denote. This would help clarify how to place the scaling law (Proposition 5) and Section 5 with respect to the results of the prior sections.

**Questions:**

In addition to the points above that should be clarified, I have one following question: how is the CIFAR-10 experiment performed? Notably, how are the surrogate and target distributions generated? This is worth expanding in the paper.

---

> ### Author Response · Authors · 2024-11-22
>
> We thank the reviewer for their positive evaluation of our work and for the detailed comments. We reply to each of the points raised in the review below. We have also uploaded a revised version of the manuscript highlighting the main changes in red color.
>
> >W1: Though the theoretical results are interesting, there are a few aspects that are worth clarifying. Notably, even though it is demonstrated that there can exist a surrogate model that induces better risk on the target model than using the true labels, in general a surrogate model is typically not trained with foreknowledge of the task distribution. I believe this is what Section 5 is trying to convey, but it is not clear to me after reading that section how to interpret the result therein. In particular, it should be explained how this relates to, for example, a target model that is trained using strong labels to demonstrate the marginal gain (or loss).
>
> **Response:** The reviewer is correct that the optimal surrogate parameter $\beta^s$ does depend on the ground-truth parameter $\beta_*$. By characterizing the optimal surrogate, we derived its form in terms of $\beta_*$ and provided intuitions for the surrogate-to-target model, including the transition points. Crucially, this also revealed that the mapping from the ground truth parameter $\beta_*$ to the optimal surrogate parameter $\beta^{s}$ is purely in terms of the feature covariance and does not depend on the foreknowledge of the task distribution, as stated in Proposition 1. This means that, rather than pruning $\beta_*$, we can alternatively modify the feature covariance (e.g. by truncating tail eigendirections) during the two stage regression in Section 5, as the reviewer has also suggested. If the first stage has access to infinite data (population covariance), two stage regression with covariance truncation is identical to Section 3.1. If we have finite data, then we can use empirical covariance as an approximation.
>
> Indeed, the aim of Section 5 is to fill the gap between theoretical findings and practical applications because this section characterizes the risk of the surrogate-to-target model when the surrogate model has finite data. We have extended the analysis in Section 5 to provide a more comprehensive treatment of the findings from Section 3.1 in Appendix C.1., where we allow the surrogate model to be inside the under-parametrized region.
>
> >W2:In general, the paper accrues a lot of jargon and notation; it would be very helpful to either create a table containing the different set-ups/regimes considered and the summary conclusion of the relative gain/suboptimality of knowledge distillation and/or a notation table that summarizes what the various risks denote. This would help clarify how to place the scaling law (Proposition 5) and Section 5 with respect to the results of the prior sections.
>
> **Response:** We thank the reviewer for the question. We have created a notation table at the beginning of Appendix A to bring further clarification. In Section 3, we analyze the case where the surrogate parameter $\beta^s$ is given and optimize the surrogate parameter based on the asymptotic risk. In Section 4, we analyze the scaling performance of the surrogate-to-target model as $p \rightarrow \infty$ and find that the scaling law of the surrogate-to-target model and target model is the same. Finally, in Section 5, we analyze the case where the surrogate parameter $\beta^s$ is a solution to an empirical minimization problem. To further clarify our results in the main text, we have added an explanation of the settings in the statement of Proposition 6 (Proposition 5 in the submitted version). In Theorems 1 and 2, we had already explained the setting and its notation in the statement of the submitted version. We are open to any other suggestions to clarify our statements.

---

> > ### Author Response · Authors · 2024-11-22
> >
> > >Q1:In addition to the points above that should be clarified, I have one following question: how is the CIFAR-10 experiment performed? Notably, how are the surrogate and target distributions generated? This is worth expanding in the paper.
> >
> > **Response:** Thank you for your helpful question. In the CIFAR-10 experiment, we initially trained the surrogate models on the training portion of the CIFAR-10 dataset. While training the surrogate-to-target models, we used the predictions from the surrogate models, and when training the standard target model, we utilized the ground truth labels. During testing, all models were evaluated using the test portion of the CIFAR-10 dataset. We note that the data distributions for both surrogate and target models are identical since we trained and tested the models on the same training and testing sets.
> >
> > We employ three distinct surrogate model sizes: big, medium, and small. The big model contains 127,094 parameters, the medium model 58,342 parameters, and the small model 28,286 parameters. All three models are shallow, three-layer convolutional networks that follow the same architectural specifications. Additional architectural details, including parameter configurations and hyperparameters, can be found in Appendix A.2, where we have provided a detailed table for reference.

---

> > > ### Comment · Reviewer_E9SJ · 2024-11-24
> > >
> > > Thank you for the clarifications. My comments have been addressed.

---

### Author Response · Authors · 2024-11-22
**Main Response to All Reviewers**

Firstly, we would like to thank the reviewers for their thoughtful and insightful comments. We are encouraged by their recognition that our work provides a useful theoretical framework for knowledge distillation and weak-to-strong generalization, and we greatly appreciate their positive feedback on the theoretical novelty and insightful nature of our findings. In response to the reviewers' suggestions, we have uploaded a revised manuscript with changes shown in red. The revisions can be summarized as follows:

* We have added further intuitions for the terms in the asymptotic risk expression presented in Theorem 1. These explanations provide deeper insights into why the surrogate-to-target model can achieve better performance compared to the standard target model. The implicit regularization induced by the minimum norm interpolator in the over-parameterized regime is the primary factor driving this improved performance. As a justification for our claim, we have introduced a new proposition establishing a necessary condition: the target model has to be over-parameterized for the surrogate-to-target model to offer a performance advantage.

* In Appendix A.1., we have provided an initial effort to extend our analysis from ridgeless regression to ridge regression. Our results in ridgeless regression can be applied to ridge regression by defining the parameter $\tau_{s,t}$ based on the ridge parameter $\lambda_{s,t}$ as follows:

&nbsp;&nbsp;&nbsp;&nbsp;&nbsp;&nbsp; $\tau_t$ is the solution of the following fixed point equation:

$$ \qquad \qquad \kappa_t^{-1}   - \frac{\lambda_t}{\tau_t}  = \frac{1}{p} \textbf{tr}((\Sigma_t + \tau_t \mathbf{I})^{-1} \Sigma_t) $$


* In Section 5, we have provided an asymptotic analysis of the two-stage model, specifically considering the cases where the surrogate model operates in the under-parameterized regime. This analysis extends the applicability/results of Section 3.1 by incorporating scenarios where the surrogate model is trained on a finite dataset.

---

> ### Author Response · Authors · 2024-12-03
>
> We would like to thank all the reviewers for their constructive comments, which significantly helped improve both the clarity and content of the paper.

---

### Meta-Review · Area_Chair_o8DV · 2024-12-16

**Metareview:**

The work presents new results on knowledge distillation, with focus on the linear Gaussian setting. The work characterizes excess risk of such linear regression based on pseudo labels  from a surrogate model, based on covariate shift. The work presents a clear set of results, demonstrating how using a surrogate model can help with knowledge distillation.

All reviewers found the results novel and interesting, and the presentation to be mostly clear. The clarity and novelty of the results can form the basis of future work on the topic.

**Additional Comments On Reviewer Discussion:**

All reviewers engaged with the authors, acknowledged the author responses, and, in some cases, followed up with additional questions, which the authors satisfactorily responded to.

---

### Decision · Program_Chairs · 2025-01-22

Accept (Spotlight)